# PROVABLE AND PRACTICAL IN-CONTEXT POLICY OPTIMIZATION FOR SELF-IMPROVEMENT

**Tianrun Yu**[1*], **Yuxiao Yang**[2*], **Zhaoyang Wang**[2], **Kaixiang Zhao**[1], **Porter Jenkins**[1],
**Xuchao Zhang**[3], **Chetan Bansal**[3], **Huaxiu Yao**[2], **Weitong Zhang**[2]
[1]Brigham Young University, [2]University of North Carolina at Chapel Hill, [3]Microsoft
[*]Equal contribution.    `tianruny@byu.edu, {yxyang, weitongz}@unc.edu`

## ABSTRACT

We study test-time scaling, where a model improves its answer through multi-round self-reflection at inference. We introduce In-Context Policy Optimization (ICPO), in which an agent optimizes its response in context using self-assessed or externally observed rewards without modifying its parameters. To explain this ICPO process, we theoretically show that with sufficient pretraining under a novel Fisher-weighted logit-matching objective, a single-layer linear self-attention model can provably imitate policy-optimization algorithm for linear bandits. Building on this theory, we propose Minimum-Entropy ICPO (ME-ICPO), a practical algorithm that iteratively uses its response and self-assessed reward to refine its response in-context at inference time. By selecting the responses and their rewards with minimum entropy, ME-ICPO ensures the robustness of the self-assessed rewards via majority voting. Across standard mathematical reasoning tasks, ME-ICPO attains competitive, top-tier performance while keeping inference costs affordable compared with other inference-time algorithms. Overall, ICPO provides a principled understanding of self-reflection in LLMs and yields practical benefits for test-time scaling for mathematical reasoning.

## 1 INTRODUCTION

Recent years have witnessed a growing capacity for large language models (LLMs) with rising abilities in mathematical reasoning (Yang et al., 2024a; Wei et al., 2022), problem solving (Rein et al., 2024; Zhou et al., 2024) and tool use (Yao et al., 2023b). Among these new abilities, the emergence of test-time scaling has been playing an important role, where the LLMs progressively improve their response through multi-round self-reflection without parameter updates. This test-time scaling has demonstrated a strong ability to enable LLMs to perform post-training search (Yao et al., 2023a; Besta et al., 2024), self-reflection and self-rewarding (Madaan et al., 2023; Shinn et al., 2023; Lightman et al., 2023) and Chain-of-Thoughts (CoT, Wei et al. 2022). The key part of this process hinges on the model's ability to digest the in-context information to improve its response. Such in-context information can be the previous response with users' finetuning instructions, or the CoT process with self-assessed rewards. However, despite repeated empirical validation, the mechanism underlying such in-context self-improvement remains under-explored in the literature. Existing works (Park et al., 2024) usually assume this ability for conducting the posterior sampling or policy optimization *intrinsically* within LLMs without answering why this ability emerges during the pretraining process.

On the other hand, recent works have attempted to understand the in-context learning for supervised learning (e.g., linear regression Zhang et al. 2024b; Garg et al. 2022) and reinforcement learning (e.g., TD learning Wang et al. 2024) that shows that some carefully designed transformers can learn these algorithms with sufficient pretraining. Yet, most of these works consider empowering the LLMs to predict the output based on the input, while it is vacant in literature understanding how transformers learn to optimize its behavior $\mathbf{x}$ by optimizing its policy towards maximizing the response $y$. In addition, there is a huge gap between the current theoretical understanding of the in-context learning and the empirical implementation of the in-context test-time scaling. Witnessing these lacks of theoretical understanding of the in-context policy optimization and the missing of how to leverage these in-context information iteratively in the test-time scaling for reasoning tasks, we would like to ask:

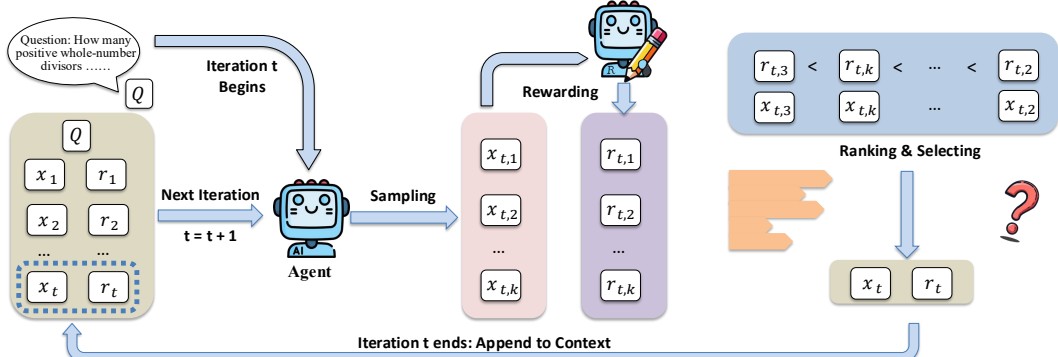

Figure 1: The In-Context Policy Optimization (ICPO) framework. At each round $t$, the agent leverages its history of past attempts with bandit feedback $\{(\mathbf{x}_1, r_1), \dots, (\mathbf{x}_t, r_t)\}$ to improve its response $\mathbf{x}_{t+1}$ in order to maximize the received reward $r_t$.

*Can we understand the self-reflection process of LLM from the in-context learning that inspires a test-time scaling for reasoning?*

In this paper, we answer this question affirmatively by providing the In-Context Policy Optimization (ICPO) framework which considers how LLMs leverage the in-context actions and response to improve their response $\mathbf{x}$ instead of predicting some certain outcomes. As illustrated in Figure 1, the ICPO process considers the transformer (LLMs) generating its response $\mathbf{x}_t$ and receives the reward given by user or self-assessment $y_t$ and then improve its response by generating $\mathbf{x}_{t+1}$. Theoretically, we show that with sufficient pretraining, a one-layer transformer is sufficient to execute a policy-optimization framework that gradually improves its response $\mathbf{x}$ using the observed rewards $r$. When applying the ICPO framework into the practical mathematical reasoning, we empirically show that the ICPO framework is robust enough to take the self-accessed reward into its policy optimization process and to gradually improve its response. Together with solid theoretical results on the ICPO process and a carefully designed practical algorithm ME-ICPO, we provide a provable and practical in-context learning framework for test-time scaling for mathematical reasoning. Together, our contributions are:

- We formulate the multi-round self-reflection mechanism as in-context policy optimization (ICPO) framework where the agent generates and improves its response with the received feedback. The ICPO framework extends the current in-context learning framework from supervised learning to policy optimization with bandit feedback. ICPO builds a theoretical foundation to understand the self-reflection and self-improvement for LLM reasoning.

- We prove that, under an explicit design of the Linear Self-Attention (LSA) transformer, when the LSA is sufficiently pre-trained on trajectories generated by a special policy-optimization framework, it provably mimics the underlying policy optimization even under previously unseen reward functions. To the best of our knowledge, this is the first directly derived mechanistic account of in-context policy optimization that provides detailed structural characterization.

- Empirically, we propose ME-ICPO, a practical algorithm grounded in our theory that yields substantial improvements over base models on mathematical reasoning tasks. ME-ICPO demonstrates that the ICPO framework can leverage self-assessed feedback, using entropy-regularized response selection to ensure robust policy updates.

Together, our work shows the first in-context optimization mechanism to help understand how LLMs can improve their response with self-reflection, with strong empirical performance in various mathematical reasoning tasks.

**Notation.** In this paper, we use plain letters such as $x$ to denote scalars, lowercase bold letters such as $\mathbf{x}$ to denote vectors, and uppercase bold letters such as $\mathbf{A}$ to denote matrices. Functions are denoted by bold symbols such as $\boldsymbol{f}$. Sets and classes are denoted by the calligraphic font such as $\mathcal{F}$. For a vector $\mathbf{x}$, $\|\mathbf{x}\|_2$ denotes its $\ell_2$-norm. For a matrix $\mathbf{A}$, $\|\mathbf{A}\|_{\mathrm{op}}$ denotes its operator (spectral) norm, i.e., $\|\mathbf{A}\|_{\mathrm{op}} := \sup_{\|\mathbf{x}\|_2=1} \|\mathbf{A}\mathbf{x}\|_2 = \sigma_{\max}(\mathbf{A})$. $\mathbf{F}(\mathbf{p}) := \mathrm{Diag}(\mathbf{p}) - \mathbf{p}\mathbf{p}^\top$. For a vector $\mathbf{x} \in \mathbb{R}^K$, $\mathrm{Diag}(\mathbf{x}) \in \mathbb{R}^{K \times K}$ denotes the diagonal matrix with $[\mathrm{Diag}(\mathbf{x})]_{ii} = x_i$ and off-diagonals zero. For a positive integer $N$, we use $[N]$ to denote $\{1, 2, \dots, N\}$.

## 2    RELATED WORK

We introduce the works on test-time scaling, self-reflection and in-context learning in this section.

**Test-Time Scaling.** Test-time scaling (or inference-time scaling) refers to the phenomenon where allocating more resources during test-time can improve the LLM's ability in reasoning and have been widely adopted in practice (Jaech et al., 2024; Guo et al., 2025). The earliest test-time scaling can be dated back to the few-shot Chain-of-Thought (Wei et al., 2022) where the provided few-shot demonstrations can improve the LLM reasoning ability. Other test-time scaling works focus on the post-training search algorithms, including the Monte-Carlo Tree Search (Zhang et al., 2024a), Best-of-N (Wang et al., 2022; Huang et al., 2025), Tree of Thoughts (Yao et al., 2023a). Following up with these works, TTRL (Zuo et al., 2025) provides a gradient-based update based on the self-assessment during the test-time and improves the LLM's response by updating its parameters.

**Self-reflection and self-assessment.** At the core of the test-time scaling lies the self-reflection and self-assessment where the LLM evaluates its own response via the self-rewarding (Madaan et al., 2023; Shinn et al., 2023). In particular, LLM-as-a-Judge and Majority-Judgment (MJ) provide inexpensive but noisy preference signals, and self-consistency can be converted into preferences or rankings (Prasad et al., 2024). However, self-evaluation suffers from prompt/position sensitivity and stylistic bias, calling for calibration (symmetric prompting, position shuffling, executability/format checks) (Wang et al., 2025). In parallel, process supervision (PRMs (Wang et al., 2023a; Chen et al., 2024b), step-wise verifiers (Lightman et al., 2023)) shifts supervision from outcomes to intermediate steps, reliably filtering errors across rounds (Lightman et al., 2023). Recent analysis of the *generation–verification (GV) gap* shows iterative improvement succeeds when verification is substantially easier than generation, motivating robust filtering and feedback (Song et al., 2024). Beyond these empirical works, recent theoretical works focus on the posterior sampling of LLM (Bai et al., 2023; Von Oswald et al., 2023) by directly assuming the LLM's ability for estimating the posterior distribution.

**In-Context Learning and In-Context Reinforcement Learning.** Besides the empirical advances, a line of theory clarifies regression as a core sandbox for ICL. For ridge linear regression, trained linear self-attention can implement preconditioned gradient descent in context, with model depth matching the number of implicit update steps and geometric convergence under standard assumptions (Von Oswald et al., 2023). From a training-dynamics perspective, prior work shows that, after training, a single-head linear attention layer effectively performs one step of gradient descent on the contextual least-squares objective (Zhang et al., 2024b). In sparse settings, multi-head constructions can recover sparse signals and carry out sparse linear regression in context (Chen et al., 2024a); recent work further identifies a layered mechanism in which first-layer heads preprocess the context and later layers carry out simple iterative optimization, together yielding excess risk guarantees that improve over naive gradient descent and ridge baselines (Chen et al., 2024a). Besides these works in understanding the regressions, more recent work has pushed forward the understanding of in-context learning to a meta-reinforcement learning process. In particular, (Lin et al., 2023) proved that a multi-layer transformer structure can imitate bandit/RL-style updates by pretraining on trajectories, and (Wang et al., 2024) shows that the linear regression for the in-context learning can be extended to the TD learning used in RL. Despite these, rare recent literature has covered the policy optimization which directly optimized the output $\mathbf{x}$ given the historical information.

## 3    PRELIMINARIES

We consider a multi-arm bandit abstraction which is a standard theoretical framework for sequential decision making. We consider a $K$-armed bandit and at each round $t \in [T]$, the agent selects an action $A_t \in [K]$ and plays the corresponding action written as the one-hot vector $\mathbf{x}_t = \mathbf{e}_{A_t} \in \mathbb{R}^K$. The agent then receives a scalar reward $r_t$ generated from a linear model with an unknown task vector $\mathbf{w} \in \mathbb{R}^K$ by $r_t = \langle \mathbf{w}, \mathbf{x}_t \rangle + \epsilon_t$, where $\epsilon_t$ is a zero-mean $\sigma_\xi$-sub-Gaussian random variable. The agent overall goal is to optimize its policy $\mathbf{x}_t$ by maximizing the expected return $\langle \mathbf{w}, \mathbf{x}_t \rangle$.

**Policy Optimization Framework.** We consider the pretrained dataset is generated from the policy optimization process in meta reinforcement learning. In particular, we start with the mirror descent that is similar to the Follow-the-Regularized Leader (FTRL, Shalev-Shwartz 2007; McMahan 2011a) framework given by

$$\mathbf{p}_{t+1} = \arg\max_{\mathbf{p} \in \Delta^K} \sum_{s=1}^{t} \left\langle \frac{r_s}{p_{s,A_s}} \mathbf{x}_s, \mathbf{p} \right\rangle - R(\mathbf{p}),$$

in the ICPO framework, we consider a practical solution in optimizing the log-likelihood of the policy defined by $\mathbf{s} \propto \log \mathbf{p}$. In the following of this paper, we consider the policy optimization written by

$$\mathbf{s}_{t+1} = \arg\max_{\mathbf{s} \in \mathbb{R}^K} \sum_{s=1}^{t} \langle r_s \mathbf{x}_s, \mathbf{s}_t \rangle - \lambda \sum_{s=1}^{t} \langle \mathbf{x}_s, \mathbf{s}_t \rangle - \frac{1}{2\eta_t} \mathbf{s}^\top \mathbf{H} \mathbf{s}, \qquad (3.1)$$

where we implement the entropy regularizer $R(\mathbf{p}) \approx \mathbf{s}^\top \mathbf{H} \mathbf{s}$ and replace the unbiased estimator $r_s/p_{s,A_s}$ used in FTRL with a Lagrange multiplier $\lambda \sum_{s=1}^{t} \langle \mathbf{x}_s, \mathbf{s}_t \rangle$ to penalize the frequently visited arms. The closed form solution for equation 3.1 yields a linear structure on $\mathbf{s}$ written by

$$\mathbf{s}_{t+1} = \eta_t(\mathbf{U}\mathbf{g}_t + \mathbf{V}\mathbf{n}_t), \text{ where } \mathbf{U} = \mathbf{H}^{-1}, \mathbf{V} = -\lambda\mathbf{H}^{-1}, \mathbf{g}_t = \sum_{s=1}^{t} r_s \mathbf{x}_s, \mathbf{n} = \sum_{s=1}^{t} \mathbf{x}_s \quad (3.2)$$

where we set $\eta_t = c/t$ and the optimized policy is then given by a softmax policy mixed with a $\gamma$-greedy random exploration

$$\mathbf{p}_{t+1} = \mathrm{softmax}(\mathbf{s}_{t+1}), \ \mathbf{p}_{t+1}^{\mathrm{PO}} = (1-\gamma)\mathbf{p}_{t+1} + \gamma\frac{\mathbf{1}}{K}, \ \gamma \in [0,1). \qquad (3.3)$$

**Supervised Pretraining Data Generation.** We generate a pretraining dataset by running the expert policy optimization algorithm. We sample $B$ independent trajectories. For each trajectory $\tau \in [B]$, a task vector $\mathbf{w}_\tau \sim \mathcal{N}(\mathbf{0}, \tau_w^2 \mathbf{I}_K)$ is sampled from the prior. The teacher is then executed for $N$ steps to generate a complete history of interactions $\mathcal{H}_{\tau,N} = \{(\mathbf{x}_{\tau,1}, r_{\tau,1}), \ldots, (\mathbf{x}_{\tau,N}, r_{\tau,N})\}$ and the corresponding sequence of expert logit vectors $\{\mathbf{s}_{\tau,t}^{\mathrm{PO}}\}_{t=1}^{N}$ is updated in equation 3.2. From these trajectories, we construct a supervised training dataset, $\mathcal{D}$ consisting of pairs of history prefixes and the teacher's next-step logits. The final dataset is the set of all such pairs $\mathcal{D} = \{(\mathcal{H}_{\tau,t}, \mathbf{s}_{\tau,t+1}^{\mathrm{PO}})\}_{\tau \in [B]}^{t \in [N-1]}$, where $\mathcal{H}_{\tau,t}$ is the history prefix of trajectory $\tau$ up to and including round $t$.

The following assumption is made on the data coverage on the supervised pretraining data $\mathcal{D}$, which is a standard *diversity* assumption in linear bandits (Papini et al., 2021; Hao et al., 2020; Wu et al., 2020).

**Assumption 3.1** (Data Coverage and Signal Dominance)**.** We assume that in the pretrained dataset, the coverage of the task $\tau_w^2$ and the FTRL exploration parameter $\gamma$ is wide enough to cover the reward noise. In particular, define the learning rate $\eta_t = c/t$, we assume the coverage rate is strictly positive

$$c_\lambda := \tau_w \gamma/K - (1-\gamma)c\|\mathbf{U}\|_{\mathrm{op}}\sigma_\xi^2/2 > 0.$$

**Linear Self-Attention (LSA).** The Linear Self-Attention (LSA) is a simplified variant of the standard self-attention mechanism, which has been established as a useful model for the theoretical analysis of transformers and in-context learning (Von Oswald et al., 2023; Zhang et al., 2024b). An LSA layer takes a sequence of input embeddings, represented as a matrix $\mathbf{E} \in \mathbb{R}^{d \times N}$, where $d$ is the embedding dimension and $N$ is the sequence length. It produces an output matrix of the same dimension through the following computational form:

$$f_{\mathrm{lsa}}(\mathbf{E}; \boldsymbol{\theta}) = \mathbf{E} + \mathbf{W}^{PV}\mathbf{E} \cdot \left(\mathbf{E}^\top \mathbf{W}^{KQ}\mathbf{E}/\rho\right), \qquad (3.4)$$

where $\boldsymbol{\theta} = (\mathbf{W}^{KQ}, \mathbf{W}^{PV})$ are learnable parameters (matrices) and $\rho$ is a normalization factor for the attention matrix. This operation allows for interactions between all elements in the input sequence, mediated by the Gram matrix term $\mathbf{E}^\top \mathbf{W}^{KQ}\mathbf{E}$, to update the initial embeddings.

# 4 THEORETICAL FRAMEWORK FOR ICPO

In this section, we provide a theoretical justification for in-context policy optimization based on an inspirational analysis in a Linear Self-Attention (LSA) network. Through this minimal LSA model, we theoretically demonstrate that a pretrained LSA can imitate an expert policy optimization algorithm using in-context data. We then present our main theoretical guarantees, which establish that this learning is not only possible in principle but also efficient with a finite amount of data, and robust to perturbations at test time.

**The ICPO Forward Pass.** We start with introducing the forward pass of ICPO framework. The LSA model parameterized by $\boldsymbol{\theta}$ starts with an empty embedding $\mathbf{E}^{(0)} = (\mathbf{q}_x, q_r)^\top$ where $\mathbf{q}_x = \mathbf{1}_K, q_r = 0$ are the placeholder for next-token generation. For each time step $t \in [T]$, the LSA model updates its policy according to the logits updates from the next-token generation of LSA described by

$$\widehat{\mathbf{s}}_t := \left[f_{\mathrm{LSA}}(\mathbf{E}^{(t-1)}; \boldsymbol{\theta})\right]_{1:K,\,t}, \qquad \mathbf{p}_t = (1-\gamma)\,\mathrm{softmax}(\widehat{\mathbf{s}}_t) + \frac{\gamma}{K}\mathbf{1},$$

where $\left[f_{\mathrm{LSA}}(\mathbf{E}^{(t-1)}; \boldsymbol{\theta}^*)\right]_{1:K,t}$ stands for the corresponding $K$ dimensions of the newly generated token indexed with $t$. $\gamma$ is the same exploration factor in the implementation of FTRL. With this policy $\mathbf{p}_t \in \mathbb{R}^K$, the LSA model selects its action $A_t$ and receives the reward by

$$A_t \sim \mathbf{p}_t, \quad \mathbf{x}_t = \mathbf{e}_{A_t}, \quad r_t = \langle \mathbf{w}, \mathbf{x}_t \rangle + \epsilon_t.$$

Finally the sequence of token is updated by inserting the observed reward $r_t$ and embedding $\mathbf{x}_t$ by

$$\mathbf{E}^{(t)} = \begin{pmatrix} \mathbf{x}_1 & \cdots & \mathbf{x}_t & \boldsymbol{q}_x \\ r_1 & \cdots & r_t & q_r \end{pmatrix}$$

and update the normalizing factor in LSA as $\rho = t$ to ensure the attention matrix is upper bounded by 1. The forward pass of ICPO then move to the next round $t \leftarrow t + 1$.

**Training Objective.** The supervised pretraining is conducted on the dataset $\mathcal{D}$ by matching the logits from the model output $\widehat{\mathbf{s}}_t = f_{\mathrm{LSA}}(\mathbf{E}^{(t-1)}, \boldsymbol{\theta})$ with the logits from policy optimization $\mathbf{s}_t^{\mathrm{PO}}$ by minimizing the projected weighted loss by

$$\mathcal{L}(\boldsymbol{\theta}) = \tfrac{1}{2}\mathbb{E}_{\tau \in \mathcal{D}}\left[\sum_{t=1}^{N-1} \left\|\mathrm{Proj}(\widehat{\mathbf{s}}_{\tau,t+1} - \mathbf{s}_{\tau,t+1}^{\mathrm{PO}})\right\|_{\boldsymbol{\Gamma}}^2\right], \tag{4.1}$$

where projection Proj is defined as $\boldsymbol{\Pi}_\perp := \mathbf{I} - \tfrac{1}{K}\mathbf{1}\mathbf{1}^\top$ which removes the constant bias $\mathbf{1}^\top \mathbf{s}$ from the logits, since such shifts do not affect the policy $\mathbf{p} \propto \exp(\mathbf{s})$. The expected matrix $\boldsymbol{\Gamma}$ is inspired by the design of Natural Policy Optimization (Kakade, 2001) defined by

$$\boldsymbol{\Gamma} = \tfrac{1}{N-1}\mathbb{E}_{\tau \in \mathcal{D}}\left[\sum_{t=1}^{N-1}\mathrm{Diag}(\mathbf{p}_{\tau,t}) - \mathbf{p}_{\tau,t}\mathbf{p}_{\tau,t}^\top\right].$$

The Fisher-weighted loss provides a new loss for the supervised pretraining. We show by the following theorem that the common KL loss between the pretrained data $\mathbf{p}_{t+1}^{\mathrm{PO}}$ and the LSA's output $\widehat{\mathbf{p}}_{t+1}$ is sandwiched by the loss $\mathcal{L}(\boldsymbol{\theta})$ up to constants.

**Theorem 4.1** (mixed-policy KL is controlled by the Fisher-projected quadratic loss). Assume both teacher and student use $\gamma$-mixture exploration with $\gamma \in (0, 1)$ as described in equation 3.3, and let $N$ denote the trajectory length of the sample inside the expectation. Then,

$$\frac{(1-\gamma)^2}{4}\mathcal{L}(\boldsymbol{\theta}) \le \mathbb{E}\left[\tfrac{1}{N-1}\sum_{t=1}^{N-1}D_{\mathrm{KL}}\big(\mathbf{p}_{t+1}^{\mathrm{PO}} \,\|\, \widehat{\mathbf{p}}_{t+1}\big)\right] \le \frac{K}{4\gamma}\mathcal{L}(\boldsymbol{\theta}).$$

Theorem 4.1 suggests that the widely used KL loss is a good surrogate to the Fisher-weighted loss and explains that even in a linear self-attention layer, using the KL loss enables the transformers to learn self-reflection and improve it's response.

## 4.1 THEORETICAL GUARANTEES FOR ICPO

We now present our theoretical results considering the empirical Fisher-weighted loss defined by

$$\widehat{\mathcal{L}}(\boldsymbol{\theta}) := \tfrac{1}{2B(N-1)}\sum_{\tau \in \mathcal{D}}\sum_{t \in [T]}\left\|\mathrm{Proj}\big(\widehat{\mathbf{s}}_{\tau,t+1} - \mathbf{s}_{\tau,t+1}^{\mathrm{PO}}\big)\right\|_{\widehat{\boldsymbol{\Gamma}}}^2, \quad \widehat{\boldsymbol{\Gamma}} := \tfrac{1}{B(N-1)}\sum_{\tau \in \mathcal{D}}\sum_{t \in [T]}\big(\mathrm{Diag}(\mathbf{p}_{\tau,t}) - \mathbf{p}_{\tau,t}\mathbf{p}_{\tau,t}^\top\big).$$

We denote the *population* optimizer as $\boldsymbol{\theta}^* = \arg\min_{\boldsymbol{\theta}} \mathcal{L}(\boldsymbol{\theta})$ and it's *empirical* solution $\widehat{\boldsymbol{\theta}} = \arg\min \widehat{\mathcal{L}}(\boldsymbol{\theta})$. Then the first theorem suggests that the population optimizer $\boldsymbol{\theta}^*$ is exactly imitating the policy optimization algorithm we described in equation 3.3.

**Theorem 4.2** (Population Equivalence). Under Assumption 3.1, consider a one-layer LSA with parameter $\boldsymbol{\theta}^*$ minimizing the population loss $\mathcal{L}(\boldsymbol{\theta})$ will imitate the policy optimization behavior for all possible history trajectory $\mathcal{H}_t$:, i.e. $\widehat{\mathbf{p}}_{t+1}(\mathcal{H}_t; \boldsymbol{\theta}^\star) = \mathbf{p}_{t+1}^{\mathrm{PO}}(\mathcal{H}_t)$.

Theorem 4.2 suggests that the population loss $\mathcal{L}(\boldsymbol{\theta})$ is precise and informational enough to guarantee that the parameter $\boldsymbol{\theta}^*$ will drive the LSA exact imitate the policy optimization framework leveraging any in-context data. Then a simple concentration analysis suggests that the empirical estimation will also yield a similar result with high probability:

**Theorem 4.3** (Finite sample result). Under Assumption 3.1, let the one-layer LSA be trained on $B$ i.i.d. trajectories $\{\mathcal{H}_{\tau,N}\}_{\tau=1}^B$ generated by the policy optimization process in equation 3.3. Define $M := B(N-1)$. Using all prefixes $t \in \{1, \ldots, N-1\}$ from each trajectory (allowing within-trajectory dependence), form the empirical Fisher–weighted loss $\widehat{\mathcal{L}}(\boldsymbol{\theta})$ and let the global optimizer be $\widehat{\boldsymbol{\theta}} = \arg\min_{\boldsymbol{\theta}} \widehat{\mathcal{L}}(\boldsymbol{\theta})$. For any $\delta \in (0, 1)$, with probability at least $1-\delta$, if $M \gtrsim t^2\left(2K+\log(1/\delta)\right)/c_\lambda^2$,

then for any fixed test history $\mathcal{H}_t$ we have $\widehat{\mathbf{p}}_{t+1}(\mathcal{H}_t; \widehat{\boldsymbol{\theta}}) = \mathbf{p}_{t+1}^{\mathrm{PO}}(\mathcal{H}_t)$. In addition, the expected behavioral mismatch is bounded by

$$\mathbb{E}_{\mathrm{train}}\Big[\ \big\|\widehat{\mathbf{p}}_{t+1}(\mathcal{H}_t; \widehat{\boldsymbol{\theta}}) - \mathbf{p}_{t+1}^{\mathrm{PO}}(\mathcal{H}_t)\big\|_2^2\ \Big] \ \leq\ 2(1-\gamma)^2\delta,$$

where $\mathbb{E}_{\mathrm{train}}$ is taken over the randomness of the $B$ trajectories.

We would like to summarize the theoretical results by the following remark.

**Remark 4.4.** Theorem 4.3 suggests an $\widetilde{\mathcal{O}}(N^2 K/c_\lambda^2)$ sample complexity for the supervised pretrained data to guarantee the LSA is well imitating the pretrained policy optimization trajectory. Such an constant sample complexity is because of the Assumption 3.1 which is similar with the diversity assumption used in Papini et al. (2021); Hao et al. (2020); Wu et al. (2020) which suggests that $\gamma$-greedy exploration suffices for a constant regret in linear bandits.

Our analysis and framework share commons and significant difficulties compared with Lin et al. (2023); Park et al. (2024)

**Remark 4.5.** Park et al. (2024) analyze the regret of the LLMs assuming the LLM can conduct the posterior sampling without the structural analysis. In contrast, our analysis is built on a slightly modified policy optimization framework inspired by inserting an Lagrangian to solve the FTRL mirror descent. With a newly designed *supervised* learning loss, we show that an linear self attention transformer can structurally imitate the policy optimization framework. Compared with the unsupervised loss proposed in Park et al. (2024), we show by Theorem 4.1 that the newly proposed Fisher-weighted loss is a nice surrogate of the practical KL loss, which better justices that the transformers can learn policy optimization with sufficient pretraining.

**Remark 4.6.** We note that Lin et al. (2023) studies algorithm distillation and proves that $\widetilde{\mathcal{O}}(\sqrt{T})$ layers of ReLU-activated transformers can mimic (soft) Linear UCB and Linear Thompson Sampling. In contrast, we target understanding policy optimization instead of its value-based counterpart, which is more suitable for analyzing the behavior of LLMs in improving their responses. In addition, our analysis is built for a one-layer linear self-attention framework so that the network does not need to change as the context grows larger, and is more aligned with practical long-context scenarios, whereas Lin et al. (2023) requires the number of network layers to grow on the order of $\sqrt{T}$, where $T$ is the length of the in-context sequence.

In addition, as frequently discussed in previous literature (McMahan, 2011b; Shani et al., 2020), policy optimization methods such as FTRL are known to be robust to adversarial or misspecified rewards, which highlights their applicability to ICPO with self-assessed, noisy, or perturbed rewards. A crucial property for practical self-refinement is that the learned policy is also stable. We analyze the robustness of the learned ICPO loop at test time by examining its response to a single-shot reward shock, which we present in the next theorem. We will start with the definition of the CRN-coupled setup (Glasserman & Yao, 1992).

**Definition 4.7** (*s*-CRN coupled trajectories). We say two trajectories are *s*-CRN coupled when they share a *common random number* (CRN) and they are identical up to round $s-1$ in trajectory $\mathcal{H}_t$ and $\widetilde{\mathcal{H}}_t$. We denote $\mathcal{F}_s$ as the filtration that includes all events happen before observing the reward at round $s$. At $s$-th round, the reward is shifted by $\delta_r$. We define the drift cause by this reward shift by

$$\Delta\mathbf{p}_{t+1}^s(\boldsymbol{\theta}) := \widehat{\mathbf{p}}_{t+1}(\widetilde{\mathcal{H}}_t; \boldsymbol{\theta}^*) - \widehat{\mathbf{p}}_{t+1}(\mathcal{H}_t; \boldsymbol{\theta}^*) \in \mathbb{R}^K.$$

Then we are ready to present the following theorem indicating that the impact from any one-time reward perturbations will be decreasing with a well-designed learning rate $\eta_t = c/t$.

**Theorem 4.8** (Stability to One-step Reward Perturbations). Under Assumption 3.1, assume the test-time ICPO loop runs with fixed, population-optimal parameters $\boldsymbol{\theta}^\star$ in the $s$-CRN-coupled trajectories with task $\mathbf{w}$, define

$$a := \frac{c(1-\gamma)}{2}\|\mathbf{U}\|_{\mathrm{op}}, \quad b := \frac{c(1-\gamma)}{2}\sqrt{\frac{K}{2}}\Big(\|\mathbf{V} + \mathbf{U}\,\mathrm{Diag}(\mathbf{w})\|_{\mathrm{op}} + \sqrt{\tfrac{2}{\pi}}\sigma_\xi\|\mathbf{U}\|_{\mathrm{op}}\Big),$$

that does not grow with $s$ or $t$, let $C_b = f(b)$ as another absolute constant, for any $1 \leq s \leq t$,

$$\mathbb{E}\left[\|\Delta\widehat{\mathbf{p}}_{t+1}^s\|_2 | \mathcal{F}_{s-1}\right] \leq \frac{a(1+C_b)}{s}\left(\frac{t}{s}\right)^{b-1}|\delta_r|.$$

In particular, let the learning rate $\eta_t = c/t$ be sufficiently small such that $b < 1$, the one-time reward shift in the $s$-CRN-coupled trajectories is decreasing to 0, i.e., $\lim_{t\to\infty}\mathbb{E}\big[\|\Delta\widehat{\mathbf{p}}_{t+1}\|_2 \mid \mathcal{F}_{s-1}\big] = 0$.

---

**Algorithm 1** ME-ICPO: Minimum-Entropy In-Context Policy Optimization

---

**Input:** Question $Q$; number of rounds $N$; candidates per round $k$;
**Input:** System prompt SysPrompt; Summarizer (prompt) Summ
1: $\mathcal{H}_0 \leftarrow \{$SysPrompt, $Q\}$
2: **for** $t = 1$ **to** $n$ **do**
3:      Sample response $A_{1:k}^{(t)} \sim p_t(\cdot \mid \mathcal{H}_{t-1})$ with their answer $a_j^{(t)}$ for all $j \in [k]$
4:      Assess $\widehat{a}_t \leftarrow$ MajorityVote$\{a_j^{(k)}\}_{j=1}^K$, let $r_j^{(t)} \leftarrow \mathbb{1}[a_j^{(t)} = \widehat{a}_t]$ for all $j \in [k]$
5:      Summarize $x_j^{(t)} =$ Summ$(A_j^{(}t))$ and $\widetilde{\mathcal{H}}_j^{(t)} = \mathcal{H}_{t-1} \cup (x_j^{(t)}, r_j^{(t)})$ for all $j \in [k]$
6:      Select the minimum entropy response $j^* \leftarrow \arg\min_{j \in [k]} H(\widetilde{\mathcal{H}}_j^{(t)})$
7:      Update in-context list $\mathcal{H}_t \leftarrow \mathcal{H}_{t-1} \cup (x_{j^*}^{(t)}, r_{j^*}^{(t)})$
8: **end for**
**Output:** Response sampled from $p_{n+1}(\cdot \mid H_n)$

---

## 5 MINIMUM-ENTROPY IN-CONTEXT POLICY OPTIMIZATION

Based on the theoretical understanding of the ICPO framework, we now present a practical algorithm leveraging the in-context policy optimization and the self-accessed reward to improve its reasoning ability via test-time scaling. To begin with, we adopt the ICPO notation presented in our theoretical analysis. The agent starts with the historical prompt $\mathcal{H}_0 = \{Q\}$ that only contains the question $\mathbf{Q}$. For each time $t \in [T]$, the model outputs it response $\mathbf{x}_t \sim p(\cdot|\mathcal{H}_{t-1})$ and then observe the reward $r_t$ via self-accessed rewards. Then the agent updates its history $\mathcal{H}_t = \{\mathbf{Q}, (\mathbf{x}_1, r_1), \cdots, (\mathbf{x}_t, r_t)\}$ and move on to the next time step $t \leftarrow t + 1$. However, directly applying the ICPO framework presents two significant challenges. The first challenge is related to the length of the contexts preventing the agents from cumulate and conduct effective reasoning process based on a prolonged context through the in-context policy optimization process. The second is the trustworthiness of the self-assessed rewards, since the self-evaluation can be noisy and inaccurate. To tackle these two challenges, we present our test-time in-context scaling algorithm: Minimum-Entropy In-Context Policy Optimization (ME-ICPO). The ME-ICPO algorithm works in the following three major procedures.

**Response Generation and Self-Assessment.** In Line 3, the agent samples $k$ responses $A_k^{(t)}$ using the historical in-context data $\mathcal{H}_{t-1}$ with their final answer $a_k^{(t)}$ appearing in the final boxed{} block (Hendrycks et al., 2021). Then the majority voting as conducted in Wang et al. (2022) is conducted over the answers in Line 4 for assessing the accuracy of the responses.

**Chain-of-Thought Summarization.** In order to compress the information from the output response and condense the in-context texts, we summarize the responses according to their Chain-of-Thoughts (CoTs) $x_j^{(t)}$ and ignore the detailed problem-solving process in Line 5. The motivation is that the detailed numerical processing is expected to be easier than the global CoTs.

**Minimum Entropy Response Selection.** Similar to the tree-search-style algorithms Yao et al. (2023a), we select a response $x_{j^*}^{(t)}$ and put it and its response into the in-context history $\mathcal{H}_{t+1}$ in Line 7. Specifically, unlike the tree-search Yao et al. (2023a), or Best-of-N Huang et al. (2025) algorithms that are designed for multi-step reasoning where $x_t$ is the one-step response, we consider the $x_t$ to be a CoT description of solving the whole problem. Therefore, instead of selecting the $x_t$ with the highest reward, we instead follow a "pessimism" in offline reinforcement learning by selecting the $x_t$ that leads to the *minimum entropy* in the future response, i.e., $j^* \leftarrow \arg\min_{j \in [k]} H(\widetilde{\mathcal{H}}_j^{(t)})$ as conducted in Line 6. Intuitively, this *minimum entropy* selection will avoid the agent selecting the corrupted response $x$ that may drive the agent to produce a random answer. In addition, this procedure will also encourage the agent to select the diversified responses $x_{j^*}$ that would help further reduce the entropy.

It is important to distinguish our test-time approach from methods that train a student model on trajectories from a fixed teacher algorithm. Since ME-ICPO performs no parameter updates at test time, we do not claim that the deployed LLM is uniquely equivalent to any specific policy-optimization algorithm. Rather, our theoretical analysis shows that the LSA architecture possesses a strong inductive bias for performing such updates. ME-ICPO is designed to provide a usable interface-via reward-aware prompting and principled feedback selection-that effectively leverages this

inherent computational capability of the model at test time without requiring gradients (Madaan et al., 2023; Shinn et al., 2023). For complexity derivations and prefactor discussions, see Appendix C.

## 6 EXPERIMENTS

We present the experiment results to validate our theoretical claims in Section 4 and the performance of ME-ICPO in this section.

### 6.1 VALIDATION EXPERIMENTS

To verify the Theorem 4.2 and Theorem 4.8 results, we run two controlled checks on a *single* LSA. Figure 2 shows, respectively, the teacher-student *policy matching* error (Top) and the *stability* of the mixed policy under a one-time reward shock together with the instantiated analytical bound (Bottom).

**Teacher-student policy matching.** We fix the meta configuration $K = 10$, $N = 30$, $B = 100$, $\gamma = 0.2$, stepsize $\eta_t = c/t$ with $c = 1.0$, task prior $\mathbf{w} \sim \mathcal{N}(\mathbf{0}, \tau_w^2 I)$ with $\tau_w = 1.0$, and reward noise $\epsilon_t \sim \mathcal{N}(0, \sigma_\xi^2)$ with $\sigma_\xi = 0.5$. At test time we draw $B_{\text{test}} = 64$ fresh tasks from the same generative process. At each round we hold the realized history $\mathcal{H}_t$ fixed, compute the teacher mixed policy $\mathbf{p}_{t+1}^{\text{PO}}$ and the model's mixed policy $\widehat{\mathbf{p}}_{t+1}$ as defined in Section 4, and then continue the closed loop using the model's policy. Aggregating over tasks, we report the mean and one-standard-deviation band of $\mathbb{E}\|\widehat{\mathbf{p}}_t - \mathbf{p}_t^{\text{PO}}\|_2$. As shown in Figure 2, the gap rapidly drops to numerical precision and decreases with $t$, consistent with the population equivalence and finite-sample guarantees.

**Stability under a single reward shock.** We use an LSA trained via supervised imitation on PO rollouts with $K = 5$, $N = 5$, $\gamma = 0.8$, $c = 0.5$, $\tau_w = 0.5$, and $\sigma_\xi = 0.1$. At test time we keep the data model and $\gamma$ unchanged, extend the horizon to $N = 10$, and evaluate on $B_{\text{test}} = 256$ tasks. For each task we run two $s$-CRN-coupled trajectories: a baseline and a perturbed path that injects a single reward shock $\delta r_s = 1.0$ at $s = 2$. Following Def. 4.7, we record $\Delta \widehat{\mathbf{p}}_t^s$ and plot the mean and

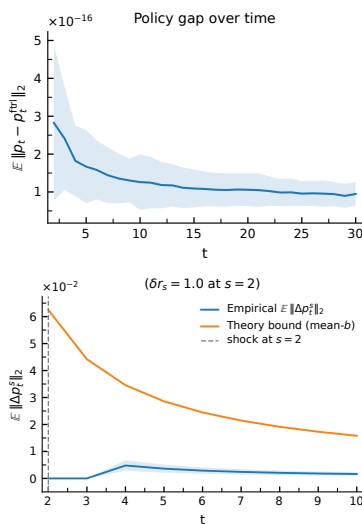

Figure 2: Validation of ICPO theory. (Top): Policy Matching. (Bottom): Reward-Shock Stability.

one-standard-deviation band of $\mathbb{E}\|\Delta \widehat{\mathbf{p}}_t^s\|_2$. We can compute $b \in [0.1236, 0.2127]$ and thus the non-amplification condition. Figure 2 shows a brief post-shock bump followed by a steady decline without amplification over time, with the analytical curve providing a conservative upper bound.

### 6.2 LLM EXPERIMENTAL SETUP

We evaluate ME-ICPO on standard mathematical QA benchmarks (**AIME 2024**, **AMC**, and **MATH-500**, split into five difficulty levels following TTRL) (aop, 2024; Mathematical Association of America, 2025; Li et al., 2024; Hendrycks et al., 2021) using representative backbones (**Qwen2.5-Math-1.5B** and **Qwen2.5-Math-7B**) (Yang et al., 2025). For reference, we also report the performance of **Llama-3.1-8B-Instruct** (Grattafiori et al., 2024) and **DeepSeek-R1-Distill-Llama-8B** (Guo et al., 2025) on **AIME 2024**. Full hyperparameters, baseline specifications, hardware/software environment, and prompt details are provided in Appendices B.2, B.3, and D. Following Guo et al. (2025), we generate $k=16$ responses per question ($T=0.6$, top-$p=0.95$). We report *Mean@k* $= \frac{1}{|\mathcal{D}|} \sum_{q \in \mathcal{D}} \frac{1}{k} \sum_{i=1}^k c_i(q)$, where $c_i(q) = \mathbf{1}[a_i(q) = a^\star(q)]$, and *Accuracy*, the probability of answering correctly with one attempt.

### 6.3 MAIN RESULTS

Our main results are presented in Table 1, reporting the mean and standard deviation (over 5 seeds) of *Mean@16* (%) and *Accuracy* (%) across all tasks and models. The results show that ME-ICPO consistently improves performance through its gradient-free, in-context optimization process. These improvements hold for both the larger model (**Qwen2.5-Math-7B**) and the smaller model (**Qwen2.5-Math-1.5B**), demonstrating that ME-ICPO is effective across different model scales. Qualitatively curated prompt examples are provided in Appendix D. We also evaluate an oracle-reward variant, with results summarized in Table 2.

We further report the performance of additional models **Llama-3.1-8B-Instruct** (Grattafiori et al., 2024) and **DeepSeek-R1-Distill-Llama-8B** (Guo et al., 2025) on **AIME 2024** in Figure 3.

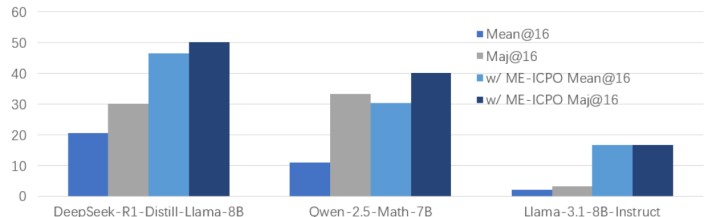

Figure 3: Performance comparison of backbone models before and after ME-ICPO.

Table 1: Performance comparison on different models with *Mean@16* and standard *Accuracy* (%).

| Model \ Benchmark | AIME 2024 | AMC | MATH-L1 | MATH-L2 | MATH-L3 | MATH-L4 | MATH-L5 |
|---|---|---|---|---|---|---|---|
| | | | *Mean@16 (%)* | | | | |
| Qwen2.5-Math-7B | $11.04 \pm 1.65$ | $41.42 \pm 0.99$ | $52.62 \pm 1.38$ | $50.76 \pm 0.95$ | $49.88 \pm 0.88$ | $43.60 \pm 0.80$ | $30.58 \pm 0.78$ |
| w/ ME-ICPO | $30.42 \pm 1.81$ | $47.06 \pm 0.84$ | $62.35 \pm 1.27$ | $64.31 \pm 1.23$ | $58.87 \pm 0.78$ | $49.31 \pm 0.97$ | $38.71 \pm 0.86$ |
| $\Delta$ | $+19.38$ | $+5.64$ | $+9.73$ | $+13.55$ | $+8.99$ | $+5.71$ | $+8.13$ |
| Qwen2.5-Math-1.5B | $6.46 \pm 0.96$ | $30.42 \pm 0.58$ | $49.27 \pm 0.81$ | $48.54 \pm 0.67$ | $45.42 \pm 0.86$ | $40.28 \pm 0.57$ | $25.23 \pm 0.45$ |
| w/ ME-ICPO | $9.79 \pm 1.11$ | $33.58 \pm 0.63$ | $61.19 \pm 0.77$ | $54.93 \pm 0.71$ | $52.08 \pm 0.69$ | $46.44 \pm 0.64$ | $29.85 \pm 0.70$ |
| $\Delta$ | $+3.33$ | $+3.16$ | $+11.92$ | $+6.39$ | $+6.66$ | $+6.16$ | $+4.62$ |
| | | | *Accuracy (%)* | | | | |
| Qwen2.5-Math-7B | $11.13 \pm 3.27$ | $41.33 \pm 1.97$ | $46.98 \pm 2.71$ | $42.67 \pm 1.86$ | $43.71 \pm 1.75$ | $37.79 \pm 1.65$ | $26.13 \pm 1.54$ |
| w/ ME-ICPO | $30.05 \pm 3.02$ | $47.20 \pm 2.26$ | $57.32 \pm 2.37$ | $54.74 \pm 2.04$ | $51.90 \pm 1.67$ | $40.84 \pm 1.78$ | $31.71 \pm 1.40$ |
| $\Delta$ | $+18.92$ | $+5.87$ | $+10.34$ | $+12.07$ | $+8.19$ | $+3.05$ | $+5.58$ |
| Qwen2.5-Math-1.5B | $6.51 \pm 1.94$ | $30.25 \pm 1.07$ | $44.68 \pm 1.83$ | $39.95 \pm 1.32$ | $39.77 \pm 0.99$ | $34.97 \pm 0.68$ | $20.89 \pm 1.35$ |
| w/ ME-ICPO | $9.82 \pm 2.15$ | $33.73 \pm 0.96$ | $57.06 \pm 1.70$ | $47.60 \pm 1.00$ | $47.81 \pm 1.17$ | $41.55 \pm 0.72$ | $24.83 \pm 1.29$ |
| $\Delta$ | $+3.31$ | $+3.48$ | $+12.38$ | $+7.65$ | $+8.04$ | $+6.58$ | $+3.94$ |

We also consider the *Maj@k* metric (Wang et al., 2023b), which aggregates $k$ responses by majority vote $\widehat{a}(q) = \text{mode}(a_1(q), \ldots, a_k(q))$ (ties broken uniformly) and computes *Maj@k* $= \frac{1}{|\mathcal{D}|} \sum_{q \in \mathcal{D}} \mathbf{1}\{\widehat{a}(q) = a^\star(q)\}$. As shown in Figure 3, our experiments indicate that ME-ICPO's average performance (*Mean@16*) can surpass the anticipated upper bound given by the base model's majority voting, similar to observations in TTRL. Furthermore, applying majority voting on top of ME-ICPO's outputs leads to additional gains.

## 6.4 ANALYSIS AND ABLATION STUDIES

We further analyze the AIME 2024 benchmark using **Qwen2.5-Math-7B** to assess the contributions of ME-ICPO's core components and its sensitivity to hyperparameters, as detailed in Appendix B.1, along with the computational cost analysis in Appendix C.

**Ablation Study.** To isolate the contributions of our core components—entropy-based selection and explicit reward signals—we conduct an ablation study, with results presented in Table 2. The results clearly demonstrate that the minimum-entropy selection criterion is the

Table 2: Ablation study of ME-ICPO. (Oracle results use groundtruth labels for reward and are shown for reference only.)

| Method | Accuracy (%) | Mean@16 (%) |
|---|---|---|
| *w/o Reward* | 19.30 | 19.17 |
| *w/o Entropy* | 5.77 | 5.83 |
| *w/o Entropy & Reward* | 6.21 | 6.46 |
| **ME-ICPO *(full)*** | **30.05** | **30.42** |
| ME-ICPO Oracle | 38.19 | 38.12 |

most critical component of our algorithm. Removing this greedy selection mechanism (*w/o Entropy*) causes a dramatic performance collapse. The explicit reward signal, made legible by our system prompt, also plays a crucial role. While keeping entropy selection active, removing the reward tags (*w/o Reward*) still results in a significant drop.

## 7 CONCLUSION

In this work, we studied the test-time scaling and self-reflection phenomenology in LLM reasoning by introducing a theoretically grounded In-Context Policy Optimization (ICPO) framework. We have shown that, under the ICPO framework, a single-layer linear self-attention transformer can provably imitate a policy-optimization algorithm, which provides a theoretical proof-of-concept for how self-reflection can be implemented within LLMs. Based on the ICPO framework, we propose a practical and effective algorithm, Minimum Entropy In-Context Policy Optimization (ME-ICPO), which provides a test time scaling pipeline with self-assessed rewards and in-context response selection. Extensive empirical studies demonstrate that our improved performance across diverse benchmarks.

REPRODUCIBILITY STATEMENT

We have made significant efforts to ensure the reproducibility of our results. We detail all datasets, model configurations, hyperparameters, and training procedures in Appendix B. To enable faithful reproduction, we release our full codebase, experiment scripts, and configuration files at https://github.com/UNCSciML/ICPO. The repository includes exact seeds, environment specifications. Our source code is provided to facilitate faithful reproduction of our experiments in the supplementary materials.

ETHICS STATEMENT

We have carefully reviewed the Code of Ethics and find that our work does not raise any significant ethical concerns. Our research does not involve human subjects, sensitive data, or potentially harmful applications. We believe our methodology and contributions align with principles of fairness, transparency, and research integrity.

ACKNOWLEDGMENTS

We thank the anonymous reviewers for their helpful comments. Part of this research is conducted during ZW's internship at Microsoft Research. This research was also supported by WZ's startup funding provided by the School of Data Science and Society at UNC Chapel Hill.

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

APPENDIX

## A   PROOFS AND TECHNICAL DETAILS FOR SECTION 4

### A.1   PRELIMINARIES AND KEY LEMMAS

In this subsection, we establish the minimal set of tools and lemmas required for the subsequent proofs. We adopt all notation and settings from the main text.

**Lemma A.1** (Mixture curvature and Fisher bounds on $\mathbf{1}^{\perp}$). *If the teacher uses mixture exploration* $\mathbf{p} = (1 - \gamma)\,\widetilde{\mathbf{p}} + \gamma\,\mathbf{1}/K$, *then the Fisher matrix* $\mathbf{F}(\mathbf{p}) := \mathrm{Diag}(\mathbf{p}) - \mathbf{p}\mathbf{p}^{\top}$ *is bounded on* $\mathbf{1}^{\perp}$:

$$\frac{\gamma}{K}\mathbf{I} \;\preceq\; \mathbf{F}(\mathbf{p})\Big|_{\mathbf{1}^{\perp}} \;\preceq\; \frac{1}{2}\mathbf{I}.$$

*Proof.* Using linearity of $\mathrm{Diag}(\cdot)$ and expanding $\mathbf{p}\mathbf{p}^{\top}$,

$$\mathbf{F}(\mathbf{p}) = \mathrm{Diag}\big((1-\gamma)\widetilde{\mathbf{p}} + \gamma\tfrac{1}{K}\big) - \big((1-\gamma)\widetilde{\mathbf{p}} + \gamma\tfrac{1}{K}\big)\big((1-\gamma)\widetilde{\mathbf{p}} + \gamma\tfrac{1}{K}\big)^{\top}$$
$$= (1-\gamma)\mathbf{F}(\widetilde{\mathbf{p}}) \;+\; \gamma\,\mathbf{F}(\tfrac{1}{K}) \;+\; \gamma(1-\gamma)\,(\widetilde{\mathbf{p}} - \tfrac{1}{K})(\widetilde{\mathbf{p}} - \tfrac{1}{K})^{\top}$$

The last rank-one term and $\mathbf{F}(\widetilde{\mathbf{p}})$ are PSD, hence

$$\mathbf{F}(\mathbf{p}) \;\succeq\; \gamma\,\mathbf{F}(\tfrac{1}{K}).$$

On $\mathbf{1}^{\perp}$, $\mathbf{F}(\tfrac{1}{k}) = \mathrm{Diag}(\tfrac{1}{k}) - (\tfrac{1}{k})(\tfrac{1}{k})^{\top}$ acts as $(\tfrac{1}{k})\mathbf{I}$, so the lower bound follows:

$$\mathbf{F}(\mathbf{p})\Big|_{\mathbf{1}^{\perp}} \;\succeq\; \frac{\gamma}{K}\mathbf{I}.$$

For the upper bound, for any $\mathbf{x} \in \mathbf{1}^{\perp}$ with $\|\mathbf{x}\|_2 = 1$,

$$\mathbf{x}^{\top}\mathbf{F}(\mathbf{p})\,\mathbf{x} \;=\; \mathrm{Var}_{i\sim\mathbf{p}}[x_i] \;\leq\; \frac{(\max_i x_i - \min_i x_i)^2}{4} \;\leq\; \frac{\big(\sqrt{2}\big)^2}{4} \;=\; \frac{1}{2},$$

where we used Popoviciu's inequality and that among vectors with $\mathbf{x}^{\top}\mathbf{1} = 0$ and $\|\mathbf{x}\|_2 = 1$, the range is maximized by placing mass $\pm 1/\sqrt{2}$ on two coordinates. Taking the supremum over such $\mathbf{x}$ yields $\mathbf{F}(\mathbf{p})\big|_{\mathbf{1}^{\perp}} \;\preceq\; \frac{1}{2}\mathbf{I}$. $\qquad\square$

**Lemma A.2** (Softmax is $1/2$-Lipschitz on $\mathbf{1}^{\perp}$). For any $\mathbf{u}, \mathbf{v} \in \mathbb{R}^K$ with $\mathbf{u} - \mathbf{v} \in \mathbf{1}^{\perp}$,

$$\|\mathrm{softmax}(\mathbf{u}) - \mathrm{softmax}(\mathbf{v})\|_2 \;\leq\; \tfrac{1}{2} \|\mathbf{u} - \mathbf{v}\|_2.$$

*Proof.* By the mean value theorem for vector maps, there exists $\xi$ on the segment $[v, u]$ such that

$$\mathrm{softmax}(\mathbf{u}) - \mathrm{softmax}(\boldsymbol{b}) \;=\; J(\xi)\,(\mathbf{u} - \mathbf{v}),$$

where $J(\xi)$ is the Jacobian of softmax. It is well known that $J(\xi) = \mathbf{F}(\mathbf{p})$ with $\mathbf{p} = \mathrm{softmax}(\xi)$, and $J(\xi)\mathbf{1} = \mathbf{0}$. Since $\mathbf{u} - \mathbf{v} \in \mathbf{1}^{\perp}$, we can restrict to $\mathbf{1}^{\perp}$ and apply Lemma A.1:

$$\|\mathrm{softmax}(\mathbf{u}) - \mathrm{softmax}(\mathbf{v})\|_2 \;=\; \|J(\xi)(\mathbf{u} - \mathbf{v})\|_2 \;\leq\; \|J(\xi)|_{\mathbf{1}^{\perp}}\|_{\mathrm{op}} \|\mathbf{u} - \mathbf{v}\|_2 \;\leq\; \tfrac{1}{2}\|\mathbf{u} - \mathbf{v}\|_2.$$

$\square$

**Lemma A.3** (Query-column closed form for one-layer LSA). The next-step logit **vector** admits the closed form

$$\widehat{\mathbf{s}}_{t+1} \;=\; \mathbf{q}_x \;+\; \frac{1}{t}\mathbf{R}\,\mathbf{G}_t\,\boldsymbol{b}, \tag{A.1}$$

where $\mathbf{R} := [\mathbf{W}^{PV}]_{1:K,:}$ is the row-selector **matrix**, $\boldsymbol{b} := \mathbf{W}^{KQ}\boldsymbol{q}$ is the transformed **query vector**, $\boldsymbol{q} := \begin{pmatrix} \mathbf{q}_x \\ q_r \end{pmatrix} \in \mathbb{R}^{K+1}$ with $\mathbf{q}_x \in \mathbb{R}^K$ and $q_r \in \mathbb{R}$, and $\mathbf{G}_t := \mathbf{E}^{(t)}(\mathbf{E}^{(t)})^{\top}$ is the history **Gram matrix**.

*Proof.* By the LSA definition in equation 3.4,

$$\begin{aligned}
\widehat{\mathbf{s}}_{t+1} &:= \left[ f_{\mathsf{lsa}}(\mathbf{E}^{(t)}; \boldsymbol{\theta}) \right]_{1:K,\, t+1} \\
&= \left[ \mathbf{E}^{(t)} + \mathbf{W}^{PV}\mathbf{E}^{(t)} \cdot \frac{(\mathbf{E}^{(t)})^{\top}\mathbf{W}^{KQ}\mathbf{E}^{(t)}}{t} \right]_{1:K,\, t+1} \\
&= \left[ \mathbf{E}^{(t)} \right]_{1:K,\, t+1} \;+\; \frac{1}{t}\left[ \mathbf{W}^{PV}\mathbf{E}^{(t)}\big((\mathbf{E}^{(t)})^{\top}\mathbf{W}^{KQ}\mathbf{E}^{(t)}\big) \right]_{1:K,\, t+1} \\
&\overset{(a)}{=} \mathbf{q}_x \;+\; \frac{1}{t}\left[ \mathbf{W}^{PV}\mathbf{E}^{(t)}\big((\mathbf{E}^{(t)})^{\top}\mathbf{W}^{KQ}\mathbf{E}^{(t)}\mathbf{e}_{t+1}\big) \right]_{1:K} \\
&\overset{(b)}{=} \mathbf{q}_x \;+\; \frac{1}{t}\left[ \mathbf{W}^{PV}\mathbf{E}^{(t)}\big((\mathbf{E}^{(t)})^{\top}\mathbf{W}^{KQ}(\mathbf{E}^{(t)}\mathbf{e}_{t+1})\big) \right]_{1:K} \\
&\overset{(c)}{=} \mathbf{q}_x \;+\; \frac{1}{t}\left[ \mathbf{W}^{PV}\big(\mathbf{E}^{(t)}(\mathbf{E}^{(t)})^{\top}\big)\big(\mathbf{W}^{KQ}\boldsymbol{q}\big) \right]_{1:K} \\
&\overset{(d)}{=} \mathbf{q}_x \;+\; \frac{1}{t}\,\mathbf{R}\,\mathbf{G}_t\,\boldsymbol{b}.
\end{aligned}$$

Justifications: (a) column slicing equals right-multiplying by the standard basis $\mathbf{e}_{t+1} \in \mathbb{R}^{t+1}$, and $[\mathbf{E}^{(t)}]_{1:K,\, t+1} = \mathbf{q}_x$; (b) associativity isolates $\mathbf{E}^{(t)}\mathbf{e}_{t+1}$; (c) substitute $\mathbf{E}^{(t)}\mathbf{e}_{t+1} = \boldsymbol{q} = (\mathbf{q}_x^{\top}, q_r)^{\top}$ and regroup $\mathbf{E}^{(t)}(\mathbf{E}^{(t)})^{\top}$; (d) apply the concise definitions $\mathbf{R} = [\mathbf{W}^{PV}]_{1:K,:}$, $\mathbf{G}_t = \mathbf{E}^{(t)}(\mathbf{E}^{(t)})^{\top}$, and $\boldsymbol{b} = \mathbf{W}^{KQ}\boldsymbol{q}$. $\square$

**Lemma A.4** (Two-channel projected logits: exact equality). Assume the architectural normal form: $q_r = 0$, $\mathbf{q}_x \in \mathrm{span}\{\mathbf{1}\}$, and the final column of the action–logit projection is parallel to $\mathbf{1}$ (i.e., $\mathbf{w}_{12}^{PV} \parallel \mathbf{1}$). Let the historical statistics up to round $t$ be the **count vector**

$$\mathbf{n}_t := \sum_{s=1}^{t} \mathbf{x}_s \;=\; \sum_{s=1}^{t} \mathbf{e}_{A_s},$$

and the **reward vector**

$$\mathbf{g}_t := \sum_{s=1}^{t} r_s \mathbf{x}_s \;=\; \sum_{s=1}^{t} r_s\, \mathbf{e}_{A_s}.$$

Let $\mathbf{W}^{PV}, \mathbf{W}^{KQ} \in \mathbb{R}^{(K+1)\times(K+1)}$ be block-partitioned as

$$\mathbf{W}^{PV} = \begin{pmatrix} \mathbf{W}_{11}^{PV} & \mathbf{w}_{12}^{PV} \\ (\mathbf{w}_{21}^{PV})^{\top} & w_{22}^{PV} \end{pmatrix}, \qquad \mathbf{W}^{KQ} = \begin{pmatrix} \mathbf{W}_{11}^{KQ} & \mathbf{w}_{12}^{KQ} \\ (\mathbf{w}_{21}^{KQ})^{\top} & w_{22}^{KQ} \end{pmatrix},$$

with $\mathbf{W}_{11}^{PV}, \mathbf{W}_{11}^{KQ} \in \mathbb{R}^{K \times K}$ and $\mathbf{w}_{12}^{PV}, \mathbf{w}_{21}^{PV}, \mathbf{w}_{12}^{KQ}, \mathbf{w}_{21}^{KQ} \in \mathbb{R}^K$. For the transformed query

$$\boldsymbol{b} := \mathbf{W}^{KQ}\boldsymbol{q} = \begin{pmatrix} \boldsymbol{\phi}_1 \\ \phi_2 \end{pmatrix}, \qquad \boldsymbol{\phi}_1 \in \mathbb{R}^K, \ \phi_2 \in \mathbb{R},$$

the projected LSA logits satisfy, for any nonzero $\boldsymbol{\phi}_1, \phi_2$,

$$\mathrm{Proj}\,(\widehat{\mathbf{s}}_{t+1}) = \frac{1}{t}\Big(\mathbf{W}_n\,\mathbf{n}_t + \mathbf{W}_g\,\mathbf{g}_t\Big), \tag{A.2}$$

with effective operators

$$\mathbf{W}_n := \mathrm{Proj}\big(\mathbf{W}_{11}^{PV}\,\mathrm{Diag}(\boldsymbol{\phi}_1)\big), \qquad \mathbf{W}_g := \mathrm{Proj}\big(\phi_2\,\mathbf{W}_{11}^{PV}\big).$$

*Proof.* Using Lemma A.3 we have $\widehat{\mathbf{s}}_{t+1} = \mathbf{q}_x + \frac{1}{t}\,\mathbf{R}\,\mathbf{G}_t\,\boldsymbol{b}$ with $\mathbf{R} = [\mathbf{W}^{PV}]_{1:K,:} = [\,\mathbf{W}_{11}^{PV} \ \mathbf{w}_{12}^{PV}\,]$ and

$$\mathbf{G}_t = \mathbf{E}^{(t)}(\mathbf{E}^{(t)})^\top = \begin{pmatrix} \mathbf{C}_t + \mathbf{q}_x\mathbf{q}_x^\top & \mathbf{g}_t \\ \mathbf{g}_t^\top & r^\top r \end{pmatrix}, \qquad \mathbf{C}_t := \mathrm{Diag}(\mathbf{n}_t), \quad r := (r_1, \ldots, r_t)^\top.$$

Projecting onto $\mathbf{1}^\perp$ and expanding block multiplications,

$$\begin{aligned}
\mathrm{Proj}(\widehat{\mathbf{s}}_{t+1}) &\overset{(a)}{=} \mathrm{Proj}\Big(\mathbf{q}_x + \frac{1}{t}\,\mathbf{R}\,\mathbf{G}_t\,\boldsymbol{b}\Big) \\
&\overset{(b)}{=} \frac{1}{t}\,\mathrm{Proj}(\mathbf{R}\,\mathbf{G}_t\,\boldsymbol{b}) \\
&\overset{(c)}{=} \frac{1}{t}\,\mathrm{Proj}\Big(\big[\,\mathbf{W}_{11}^{PV} \ \mathbf{w}_{12}^{PV}\,\big]\begin{bmatrix} \mathbf{C}_t + \mathbf{q}_x\mathbf{q}_x^\top & \mathbf{g}_t \\ \mathbf{g}_t^\top & r^\top r \end{bmatrix}\begin{bmatrix} \boldsymbol{\phi}_1 \\ \phi_2 \end{bmatrix}\Big) \\
&\overset{(d)}{=} \frac{1}{t}\,\mathrm{Proj}\big(\mathbf{W}_{11}^{PV}\big((\mathbf{C}_t + \mathbf{q}_x\mathbf{q}_x^\top)\boldsymbol{\phi}_1 + \mathbf{g}_t\,\phi_2\big) + \mathbf{w}_{12}^{PV}\big(\mathbf{g}_t^\top\boldsymbol{\phi}_1 + (r^\top r)\phi_2\big)\big) \\
&\overset{(e)}{=} \frac{1}{t}\,\mathrm{Proj}\big(\mathbf{W}_{11}^{PV}\,\mathbf{C}_t\,\boldsymbol{\phi}_1 + \mathbf{W}_{11}^{PV}\,\mathbf{g}_t\,\phi_2\big) \\
&\overset{(f)}{=} \frac{1}{t}\,\big(\mathrm{Proj}\big(\mathbf{W}_{11}^{PV}\,\mathrm{Diag}(\boldsymbol{\phi}_1)\big)\,\mathbf{n}_t + \mathrm{Proj}\big(\phi_2\,\mathbf{W}_{11}^{PV}\big)\,\mathbf{g}_t\big) \\
&\overset{(g)}{=} \frac{1}{t}\Big(\mathbf{W}_n\,\mathbf{n}_t + \mathbf{W}_g\,\mathbf{g}_t\Big).
\end{aligned}$$

Explanation of steps: (a) apply Proj to the closed form; (b) $\mathrm{Proj}\,\mathbf{q}_x = 0$ since $\mathbf{q}_x \in \mathrm{span}\{\mathbf{1}\}$; (c) block multiplication with $\mathbf{R}$, $\mathbf{G}_t$, $\boldsymbol{b}$; (d) evaluate the product explicitly; (e) both $\mathrm{Proj}(\mathbf{W}_{11}^{PV}\mathbf{q}_x\mathbf{q}_x^\top\boldsymbol{\phi}_1)$ and $\mathrm{Proj}(\mathbf{w}_{12}^{PV})$ vanish because $\mathbf{q}_x, \mathbf{w}_{12}^{PV} \parallel \mathbf{1}$; (f) use $(A\,\mathrm{Diag}(u))v = (A\,\mathrm{Diag}(v))u$ with $u = \mathbf{n}_t$, $v = \boldsymbol{\phi}_1$; (g) identify $\mathbf{W}_n, \mathbf{W}_g$ as stated. $\qquad \square$

**Lemma A.5** (Fisher–weighted quadratic in the two-channel operator). Let the normalized historical statistics be $\bar{\mathbf{n}}_t := \mathbf{n}_t/t$ and $\bar{\mathbf{g}}_t := \mathbf{g}_t/t$. Define the concatenated operator $\bar{\mathbf{W}} := [\,\mathbf{W}_n \ \mathbf{W}_g\,] \in \mathbb{R}^{K \times 2K}$ and the concatenated normalized statistic $\bar{\mathbf{z}}_t := (\bar{\mathbf{n}}_t^\top, \bar{\mathbf{g}}_t^\top)^\top \in \mathbb{R}^{2K}$. Let the population second-moment matrices be

$$\bar{\boldsymbol{\Sigma}} := \mathbb{E}[\bar{\mathbf{z}}_t\bar{\mathbf{z}}_t^\top] = \begin{pmatrix} \boldsymbol{\Sigma}_{nn} & \boldsymbol{\Sigma}_{ng} \\ \boldsymbol{\Sigma}_{gn} & \boldsymbol{\Sigma}_{gg} \end{pmatrix}, \qquad \boldsymbol{\Sigma}_{y\bar{z}} := \mathbb{E}[\mathbf{y}_{t+1}\bar{\mathbf{z}}_t^\top], \ \text{ with } \ \mathbf{y}_{t+1} := \mathrm{Proj}\,(\mathbf{s}_{t+1}^{\mathrm{PO}}).$$

With $\boldsymbol{\Gamma} := \mathbb{E}[\mathbf{F}(\mathbf{p}_{t+1}^{\mathrm{PO}})]$, the Fisher-weighted loss *from Eq. equation 4.1* admits the quadratic form

$$\mathcal{L}(\boldsymbol{\theta}) = \frac{1}{2}\,\mathrm{tr}(\boldsymbol{\Gamma}\,\bar{\mathbf{W}}\,\bar{\boldsymbol{\Sigma}}\,\bar{\mathbf{W}}^\top) - \mathrm{tr}(\boldsymbol{\Gamma}\,\boldsymbol{\Sigma}_{y\bar{z}}\,\bar{\mathbf{W}}^\top) + \frac{1}{2}\,\mathrm{tr}(\boldsymbol{\Gamma}\,\boldsymbol{\Sigma}_{yy}),$$

where $\boldsymbol{\Sigma}_{yy} := \mathbb{E}[\mathbf{y}_{t+1}\mathbf{y}_{t+1}^\top]$.

*Proof.* By Eq. equation 4.1, averaging uniformly over training pairs $(\tau, t)$,

$$\mathcal{L}(\boldsymbol{\theta}) = \frac{1}{2}\,\mathbb{E}\Big[\big\|\mathrm{Proj}\,(\widehat{\mathbf{s}}_{t+1} - \mathbf{s}_{t+1}^{\mathrm{PO}})\big\|_{\boldsymbol{\Gamma}}^2\Big].$$

By Lemma A.4, the projected student logits admit the two-channel form

$$\mathrm{Proj}\,(\widehat{\mathbf{s}}_{t+1}) = \frac{1}{t}\big(\mathbf{W}_n\mathbf{n}_t + \mathbf{W}_g\mathbf{g}_t\big) = \mathbf{W}_n\bar{\mathbf{n}}_t + \mathbf{W}_g\bar{\mathbf{g}}_t = \bar{\mathbf{W}}\,\bar{\mathbf{z}}_t.$$

Let $\mathbf{y}_{t+1} := \text{Proj}\left(\mathbf{s}_{t+1}^{\text{PO}}\right)$. Using $\|\boldsymbol{b}\|_{\boldsymbol{\Gamma}}^2 = \boldsymbol{b}^\top \boldsymbol{\Gamma} \boldsymbol{b}$, linearity of expectation, and $\text{tr}(ABC) = \text{tr}(CAB)$, we obtain

$$
\begin{aligned}
\mathcal{L}(\boldsymbol{\theta}) &= \frac{1}{2}\,\mathbb{E}\Big[\big\|\bar{\mathbf{W}}\bar{\mathbf{z}}_t - \mathbf{y}_{t+1}\big\|_{\boldsymbol{\Gamma}}^2\Big] \\
&= \frac{1}{2}\,\mathbb{E}\big[(\bar{\mathbf{W}}\bar{\mathbf{z}}_t - \mathbf{y}_{t+1})^\top \boldsymbol{\Gamma}(\bar{\mathbf{W}}\bar{\mathbf{z}}_t - \mathbf{y}_{t+1})\big] \\
&= \frac{1}{2}\,\mathbb{E}\big[\bar{\mathbf{z}}_t^\top \bar{\mathbf{W}}^\top \boldsymbol{\Gamma} \bar{\mathbf{W}}\bar{\mathbf{z}}_t\big] \;-\; \mathbb{E}\big[\bar{\mathbf{z}}_t^\top \bar{\mathbf{W}}^\top \boldsymbol{\Gamma} \mathbf{y}_{t+1}\big] \;+\; \frac{1}{2}\,\mathbb{E}\big[\mathbf{y}_{t+1}^\top \boldsymbol{\Gamma} \mathbf{y}_{t+1}\big] \\
&= \frac{1}{2}\,\text{tr}\big(\boldsymbol{\Gamma}\,\bar{\mathbf{W}}\,\mathbb{E}[\bar{\mathbf{z}}_t \bar{\mathbf{z}}_t^\top]\,\bar{\mathbf{W}}^\top\big) \;-\; \text{tr}\big(\boldsymbol{\Gamma}\,\mathbb{E}[\mathbf{y}_{t+1}\bar{\mathbf{z}}_t^\top]\,\bar{\mathbf{W}}^\top\big) \;+\; \frac{1}{2}\,\text{tr}\big(\boldsymbol{\Gamma}\,\mathbb{E}[\mathbf{y}_{t+1}\mathbf{y}_{t+1}^\top]\big) \\
&= \frac{1}{2}\,\text{tr}(\boldsymbol{\Gamma}\,\bar{\mathbf{W}}\,\bar{\boldsymbol{\Sigma}}\,\bar{\mathbf{W}}^\top) \;-\; \text{tr}(\boldsymbol{\Gamma}\,\boldsymbol{\Sigma}_{y\bar{z}}\,\bar{\mathbf{W}}^\top) \;+\; \frac{1}{2}\,\text{tr}(\boldsymbol{\Gamma}\,\boldsymbol{\Sigma}_{yy}).
\end{aligned}
$$

$\square$

**Lemma A.6** (Empirical Fisher–weighted quadratic form). Let $M := B(N-1)$ be the number of training pairs and index them by $(\tau, t)$ with $\tau \in [B]$ and $t \in [N-1]$. Define the empirical Fisher weight and empirical second moments by

$$
\widehat{\boldsymbol{\Gamma}} \;:=\; \frac{1}{M}\sum_{\tau,t}\Big(\text{Diag}(\mathbf{p}_{\tau,t}^{\text{PO}}) - \mathbf{p}_{\tau,t}^{\text{PO}}(\mathbf{p}_{\tau,t}^{\text{PO}})^\top\Big), \quad \widehat{\bar{\boldsymbol{\Sigma}}} \;:=\; \frac{1}{M}\sum_{\tau,t}\bar{\mathbf{z}}_{\tau,t}\bar{\mathbf{z}}_{\tau,t}^\top,
$$

$$
\widehat{\boldsymbol{\Sigma}}_{y\bar{z}} \;:=\; \frac{1}{M}\sum_{\tau,t}\mathbf{y}_{\tau,t+1}\bar{\mathbf{z}}_{\tau,t}^\top, \qquad \widehat{\boldsymbol{\Sigma}}_{yy} \;:=\; \frac{1}{M}\sum_{\tau,t}\mathbf{y}_{\tau,t+1}\mathbf{y}_{\tau,t+1}^\top,
$$

where $\bar{\mathbf{z}}_{\tau,t} := (\bar{\mathbf{n}}_{\tau,t}^\top, \bar{\mathbf{g}}_{\tau,t}^\top)^\top$, $\bar{\mathbf{n}}_{\tau,t} = \mathbf{n}_{\tau,t}/t$, $\bar{\mathbf{g}}_{\tau,t} = \mathbf{g}_{\tau,t}/t$, and $\mathbf{y}_{\tau,t+1} := \text{Proj}(\mathbf{s}_{\tau,t+1}^{\text{PO}})$. Let $\bar{\mathbf{W}} = [\,\mathbf{W}_n \;\; \mathbf{W}_g\,]$ be as in Lemma A.5. Then the empirical Fisher-weighted loss from Eq. equation 4.1 admits the quadratic form

$$
\widehat{\mathcal{L}}(\boldsymbol{\theta}) \;=\; \frac{1}{2}\,\text{tr}\big(\widehat{\boldsymbol{\Gamma}}\,\bar{\mathbf{W}}\,\widehat{\bar{\boldsymbol{\Sigma}}}\,\bar{\mathbf{W}}^\top\big) \;-\; \text{tr}\big(\widehat{\boldsymbol{\Gamma}}\,\widehat{\boldsymbol{\Sigma}}_{y\bar{z}}\,\bar{\mathbf{W}}^\top\big) \;+\; \frac{1}{2}\,\text{tr}\big(\widehat{\boldsymbol{\Gamma}}\,\widehat{\boldsymbol{\Sigma}}_{yy}\big),
$$

and in particular the last term is independent of $\boldsymbol{\theta}$.

*Proof.* By the definition in the main text,

$$
\widehat{\mathcal{L}}(\boldsymbol{\theta}) = \frac{1}{2M}\sum_{\tau,t}\Big\|\text{Proj}\big(\widehat{\mathbf{s}}_{\tau,t+1} - \mathbf{s}_{\tau,t+1}^{\text{PO}}\big)\Big\|_{\widehat{\boldsymbol{\Gamma}}}^2.
$$

By Lemma A.4, $\text{Proj}(\widehat{\mathbf{s}}_{\tau,t+1}) = \bar{\mathbf{W}}\,\bar{\mathbf{z}}_{\tau,t}$, hence

$$
\widehat{\mathcal{L}}(\boldsymbol{\theta}) = \frac{1}{2M}\sum_{\tau,t}\big(\bar{\mathbf{W}}\bar{\mathbf{z}}_{\tau,t} - \mathbf{y}_{\tau,t+1}\big)^\top \widehat{\boldsymbol{\Gamma}}\big(\bar{\mathbf{W}}\bar{\mathbf{z}}_{\tau,t} - \mathbf{y}_{\tau,t+1}\big).
$$

Expanding the quadratic and using linearity of trace with $\boldsymbol{b}^\top A \boldsymbol{b} = \text{tr}(A\boldsymbol{b}\boldsymbol{b}^\top)$,

$$
\begin{aligned}
\widehat{\mathcal{L}}(\boldsymbol{\theta}) =& \frac{1}{2}\,\text{tr}\Big(\widehat{\boldsymbol{\Gamma}}\,\bar{\mathbf{W}}\,\frac{1}{M}\sum_{\tau,t}\bar{\mathbf{z}}_{\tau,t}\bar{\mathbf{z}}_{\tau,t}^\top\,\bar{\mathbf{W}}^\top\Big) \;-\; \text{tr}\Big(\widehat{\boldsymbol{\Gamma}}\,\frac{1}{M}\sum_{\tau,t}\mathbf{y}_{\tau,t+1}\bar{\mathbf{z}}_{\tau,t}^\top\,\bar{\mathbf{W}}^\top\Big) \\
&+ \frac{1}{2}\,\text{tr}\Big(\widehat{\boldsymbol{\Gamma}}\,\frac{1}{M}\sum_{\tau,t}\mathbf{y}_{\tau,t+1}\mathbf{y}_{\tau,t+1}^\top\Big) \\
=& \frac{1}{2}\,\text{tr}\big(\widehat{\boldsymbol{\Gamma}}\,\bar{\mathbf{W}}\,\widehat{\bar{\boldsymbol{\Sigma}}}\,\bar{\mathbf{W}}^\top\big) \;-\; \text{tr}\big(\widehat{\boldsymbol{\Gamma}}\,\widehat{\boldsymbol{\Sigma}}_{y\bar{z}}\,\bar{\mathbf{W}}^\top\big) \;+\; \frac{1}{2}\,\text{tr}\big(\widehat{\boldsymbol{\Gamma}}\,\widehat{\boldsymbol{\Sigma}}_{yy}\big).
\end{aligned}
$$

$\square$

**Lemma A.7** (Population second moment is positive definite on $S$). Let $\bar{\mathbf{z}}_t := \binom{\bar{\mathbf{n}}_t}{\bar{\mathbf{g}}_t} \in \mathbb{R}^{2K}$ with $\bar{\mathbf{n}}_t = \frac{1}{t}\sum_{s=1}^t \mathbf{x}_s$ and $\bar{\mathbf{g}}_t = \frac{1}{t}\sum_{s=1}^t r_s\mathbf{x}_s$, where $r_s = \mathbf{w}^\top \mathbf{x}_s + \epsilon_s$. Assume $\mathbf{w} \sim \mathcal{N}(0, \tau_w^2 I_K)$ and is independent of $\{\mathbf{x}_s\}$ and $\{\epsilon_s\}$, and $\{\epsilon_s\}$ is a conditionally zero-mean $\sigma_\epsilon$-sub-Gaussian martingale difference sequence. The agent uses $\gamma$-mixture exploration. Define the population second-moment

$$
\bar{\boldsymbol{\Sigma}} := \mathbb{E}[\bar{\mathbf{z}}_t \bar{\mathbf{z}}_t^\top].
$$

Suppose the teacher's parameters (step size $c_0$, reward preconditioner $\mathbf{U}$) and problem parameters satisfy

$$\frac{(1-\gamma)}{2} c_0 \|\mathbf{U}\|_{\mathrm{op}} \sigma_\epsilon^2 < \tau_w \frac{\gamma}{K}. \tag{A.3}$$

Then for any $t \geq 2$, the restriction $\bar{\boldsymbol{\Sigma}}\big|_S$ is strictly positive definite on $S := \mathbf{1}^\perp \oplus \mathbf{1}^\perp$; in particular,

$$\lambda_{\min}(\bar{\boldsymbol{\Sigma}}\big|_S) > 0,$$

and $\bar{\boldsymbol{\Sigma}}\big|_S$ is invertible on $S$.

*Proof.* We work on $S := \mathbf{1}^\perp \oplus \mathbf{1}^\perp$. Write

$$\bar{\boldsymbol{\Sigma}} = \mathbb{E}[\bar{\mathbf{z}}_t \bar{\mathbf{z}}_t^\top] = \begin{pmatrix} \boldsymbol{\Sigma}_{nn} & \boldsymbol{\Sigma}_{ng} \\ \boldsymbol{\Sigma}_{gn} & \boldsymbol{\Sigma}_{gg} \end{pmatrix}.$$

Set

$$a := \lambda_{\min}(\boldsymbol{\Sigma}_{nn}\big|_{\mathbf{1}^\perp}), \qquad c := \lambda_{\min}(\boldsymbol{\Sigma}_{gg}\big|_{\mathbf{1}^\perp}), \qquad b := \big\|\boldsymbol{\Sigma}_{ng}\big|_{\mathbf{1}^\perp}\big\|_{\mathrm{op}}.$$

Decompose $\mathbf{x}_s = \mathbf{p}_s + \mathbf{e}_s$ with $\mathbf{p}_s := \mathbb{E}[\mathbf{x}_s \mid \mathcal{F}_{s-1}]$ and $\mathbb{E}[\mathbf{e}_s \mathbf{e}_s^\top \mid \mathcal{F}_{s-1}] = \mathbf{F}(\mathbf{p}_s)$. For any $u \in \mathbf{1}^\perp$,

$$\mathbb{E}\big[(u^\top \bar{\mathbf{n}}_t)^2\big] \geq \frac{1}{t^2} \sum_{s=1}^t \mathbb{E}\big[u^\top \mathbf{F}(\mathbf{p}_s)u\big].$$

By Lemma A.1 and mixture exploration, $\mathbf{F}(\mathbf{p}_s)\big|_{\mathbf{1}^\perp} \succeq (\gamma/K)\,I$, hence

$$a \geq \frac{1}{t^2} \sum_{s=1}^t \frac{\gamma}{K} = \frac{\gamma}{K\,t}. \tag{A.4}$$

Split $\bar{\mathbf{g}}_t = \bar{\mathbf{g}}_t^{(\mathrm{sig})} + \bar{\mathbf{g}}_t^{(\mathrm{noise})}$ with $\bar{\mathbf{g}}_t^{(\mathrm{sig})} = \frac{1}{t} \sum_{s=1}^t (\mathbf{w}^\top \mathbf{x}_s)\mathbf{x}_s = \mathrm{Diag}(\bar{\mathbf{n}}_t)\,\mathbf{w}$. For any $v \in \mathbf{1}^\perp$,

$$\mathbb{E}\big[(v^\top \bar{\mathbf{g}}_t^{(\mathrm{sig})})^2\big] = \tau_w^2\, \mathbb{E}\big[\|\mathrm{Diag}(\bar{\mathbf{n}}_t)v\|_2^2\big] = \tau_w^2 \sum_{i=1}^K \mathbb{E}[\bar{n}_{t,i}^2]\, v_i^2.$$

Let $N_{t,i} := \sum_{s=1}^t \mathbf{1}\{A_s = i\}$. Since $N_{t,i}^2 \geq N_{t,i}$, $\mathbb{E}[\bar{n}_{t,i}^2] = \mathbb{E}[N_{t,i}^2]/t^2 \geq \mathbb{E}[N_{t,i}]/t^2 = \mathbb{E}[\bar{n}_{t,i}]/t$. Mixture exploration yields $\mathbb{E}[\bar{n}_{t,i}] \geq \gamma/K$, so

$$c \geq \tau_w^2 \frac{\gamma}{K\,t}. \tag{A.5}$$

Adding the PSD noise covariance only increases $\boldsymbol{\Sigma}_{gg}$, thus equation A.5 holds.

We have

$$\boldsymbol{\Sigma}_{ng} = \mathbb{E}[\bar{\mathbf{n}}_t \bar{\mathbf{g}}_t^\top] = \underbrace{\mathbb{E}\big[\bar{\mathbf{n}}_t(\bar{\mathbf{g}}_t^{(\mathrm{sig})})^\top\big]}_{=0} + \mathbb{E}\big[\bar{\mathbf{n}}_t(\bar{\mathbf{g}}_t^{(\mathrm{noise})})^\top\big].$$

The first term vanishes because $\mathbf{w}$ is independent of $(\bar{\mathbf{n}}_t, \{\epsilon_s\})$ and $\mathbb{E}[\mathbf{w}] = 0$. For the second term,

$$\mathbb{E}\big[\bar{\mathbf{n}}_t(\bar{\mathbf{g}}_t^{(\mathrm{noise})})^\top\big] = \frac{1}{t^2} \sum_{u=1}^t \sum_{s=1}^t \mathbb{E}[\mathbf{x}_u\, \epsilon_s\, \mathbf{x}_s^\top] = \frac{1}{t^2} \sum_{s=1}^{t-1} \sum_{u=s+1}^t \mathbb{E}[\mathbf{x}_u\, \epsilon_s\, \mathbf{x}_s^\top],$$

where we used $\mathbb{E}[\epsilon_s \mid \mathcal{F}_{s-1}] = 0$ to eliminate $u \leq s$. Fix $s < u$. Consider "world 0" where $\epsilon_s$ is set to 0, and "world 1" the true world. Then

$$\mathbb{E}[\mathbf{x}_u\, \epsilon_s\, \mathbf{x}_s^\top] = \mathbb{E}\big[(\mathbf{p}_u^{(1)} - \mathbf{p}_u^{(0)})\, \epsilon_s\, \mathbf{x}_s^\top\big],$$

since $\mathbb{E}[\mathbf{p}_u^{(0)}\, \epsilon_s\, \mathbf{x}_s^\top] = 0$. With $\eta_u = c_0/u$, the projected logits satisfy (one-step normal form)

$$\mathbf{y}_u = \frac{c_0}{u-1}\big(\mathbf{V}\,\mathbf{n}_{u-1} + \mathbf{U}\,\mathbf{g}_{u-1}\big) \quad \Rightarrow \quad \Delta\mathbf{y}_u = \frac{c_0}{u-1}\,\mathbf{U}\,(\epsilon_s\,\mathbf{x}_s)$$

when only $\epsilon_s$ is perturbed. By Lemma A.2 and the $(1-\gamma)$ factor from equation 3.3,

$$\|\mathbf{p}_u^{(1)} - \mathbf{p}_u^{(0)}\|_2 \leq (1-\gamma) \cdot \tfrac{1}{2}\, \|\Delta\mathbf{y}_u\|_2 \leq (1-\gamma) \cdot \tfrac{1}{2} \cdot \frac{c_0}{u-1}\, \|\mathbf{U}\|_{\mathrm{op}}\, |\epsilon_s|.$$

Hence

$$\left\|\mathbb{E}[\mathbf{x}_u\,\epsilon_s\,\mathbf{x}_s^\top]\right\|_{\mathrm{op}} \le \mathbb{E}\big[\|\mathbf{p}_u^{(1)} - \mathbf{p}_u^{(0)}\|_2\,|\epsilon_s|\big]$$

$$\le \frac{(1-\gamma)}{2}\,\frac{c_0}{u-1}\,\|\mathbf{U}\|_{\mathrm{op}}\,\mathbb{E}[\epsilon_s^2]$$

$$\le \frac{(1-\gamma)}{2}\,\frac{c_0}{u-1}\,\|\mathbf{U}\|_{\mathrm{op}}\,\sigma_\epsilon^2.$$

Summing $u = s+1, \dots, t$ and $s = 1, \dots, t-1$ gives

$$b \;=\; \left\|\boldsymbol{\Sigma}_{ng}\big|_{\mathbf{1}^\perp}\right\|_{\mathrm{op}} \le \frac{1}{t^2}\sum_{s=1}^{t-1}\sum_{u=s+1}^{t} \frac{(1-\gamma)}{2}\,\frac{c_0}{u-1}\,\|\mathbf{U}\|_{\mathrm{op}}\,\sigma_\epsilon^2$$

$$\le \frac{(1-\gamma)}{2}\,c_0\,\|\mathbf{U}\|_{\mathrm{op}}\,\sigma_\epsilon^2 \cdot \frac{t-1}{t^2}$$

$$\le \frac{C_b}{t}, \tag{A.6}$$

with $C_b := \frac{(1-\gamma)}{2}\,c_0\,\|\mathbf{U}\|_{\mathrm{op}}\,\sigma_\epsilon^2$.

For any unit $(u,v) \in S$,

$$\begin{pmatrix} u \\ v \end{pmatrix}^\top \bar{\boldsymbol{\Sigma}}\Big|_S \begin{pmatrix} u \\ v \end{pmatrix} \;\ge\; a\|u\|^2 - 2b\|u\|\,\|v\| + c\|v\|^2 \;\ge\; \lambda_{\min}\begin{pmatrix} a & -b \\ -b & c \end{pmatrix}.$$

Thus

$$\lambda_{\min}\big(\bar{\boldsymbol{\Sigma}}\big|_S\big) \;\ge\; \frac{a + c - \sqrt{(a-c)^2 + 4b^2}}{2}. \tag{A.7}$$

By equation A.4, equation A.5, and equation A.6,

$$\sqrt{ac} \;=\; \tau_w\,\frac{\gamma}{K\,t}, \qquad b \;\le\; \frac{C_b}{t}.$$

The small cross-block condition equation A.3 implies $b < \sqrt{ac}$ (for all $t \ge 2$), hence the $2 \times 2$ matrix $\left(\begin{smallmatrix} a & -b \\ -b & c \end{smallmatrix}\right)$ is positive definite and the right-hand side of equation A.7 is strictly positive. Therefore $\bar{\boldsymbol{\Sigma}}\big|_S \succ 0$. $\qquad\square$

**Lemma A.8** (Sample second-moment concentration for $\bar{\mathbf{z}}_t$). Under Assumption 3.1 and $\mathbf{w} \sim \mathcal{N}(\mathbf{0}, \tau_w^2 \mathbf{I}_K)$, the normalized statistic $\bar{\mathbf{z}}_t := (\bar{\mathbf{n}}_t^\top, \bar{\mathbf{g}}_t^\top)^\top \in \mathbb{R}^{2K}$ is sub-Gaussian with a dimension-free $\psi_2$–norm $L := \|\bar{\mathbf{z}}_t\|_{\psi_2}$ depending only on $(\tau_w, \sigma_\epsilon, \gamma)$ (and not on $K$). Let

$$\widehat{\bar{\boldsymbol{\Sigma}}} \;:=\; \frac{1}{M}\sum_{m=1}^{M} \bar{\mathbf{z}}_t^{(m)}\bar{\mathbf{z}}_t^{(m)\top}, \qquad \bar{\boldsymbol{\Sigma}} \;:=\; \mathbb{E}[\bar{\mathbf{z}}_t\bar{\mathbf{z}}_t^\top],$$

where $\{\bar{\mathbf{z}}_t^{(m)}\}_{m=1}^{M}$ are i.i.d. copies of $\bar{\mathbf{z}}_t$ generated by independent rollouts of the same population process (fixed $\gamma$). Let $\underline{\sigma} > 0$ be the population lower bound from Lemma A.7, i.e., $\lambda_{\min}(\bar{\boldsymbol{\Sigma}}\big|_S) \ge \underline{\sigma}$ on $S := \mathbf{1}^\perp \oplus \mathbf{1}^\perp$. Then there is a universal constant $C > 0$ such that, for any $\delta \in (0,1)$, with probability at least $1 - \delta$,

$$\left\|\widehat{\bar{\boldsymbol{\Sigma}}} - \bar{\boldsymbol{\Sigma}}\right\|_{\mathrm{op}} \;\le\; C\,L^2\left(\sqrt{\frac{2K + \log(1/\delta)}{M}} + \frac{2K + \log(1/\delta)}{M}\right). \tag{A.8}$$

In particular, if

$$M \;\ge\; C_{\mathrm{mc}}\,\frac{2K + \log(1/\delta)}{\underline{\sigma}^2} \qquad \text{with} \qquad C_{\mathrm{mc}} := 16\,C^2\,L^4,$$

then $\left\|\widehat{\bar{\boldsymbol{\Sigma}}} - \bar{\boldsymbol{\Sigma}}\right\|_{\mathrm{op}} \le \frac{1}{2}\,\underline{\sigma}$, and consequently

$$\lambda_{\min}\big(\widehat{\bar{\boldsymbol{\Sigma}}}\big|_S\big) \;\ge\; \tfrac{1}{2}\,\underline{\sigma}. \tag{A.9}$$

*Proof.* The sub-Gaussianity of $\bar{\mathbf{z}}_t$ follows from: (i) $\bar{\mathbf{n}}_t = \frac{1}{t} \sum_{s=1}^{t} \mathbf{x}_s$ is an average of one-hot vectors from a $\gamma$-mixture policy, hence coordinate-wise sub-Gaussian with $\psi_2$–norm bounded by a constant depending only on $\gamma$; (ii) $\bar{\mathbf{g}}_t = \frac{1}{t} \sum_{s=1}^{t} r_s \mathbf{x}_s$ with $r_s = \mathbf{w}^\top \mathbf{x}_s + \epsilon_s$, where $\mathbf{w}$ is Gaussian independent of $\{\mathbf{x}_s\}$ and $\{\epsilon_s\}$, and $\{\epsilon_s\}$ is conditionally sub-Gaussian. Thus $\bar{\mathbf{g}}_t$ is a sum of sub-Gaussian vectors with $\psi_2$–norm controlled by $(\tau_w, \sigma_\epsilon, \gamma)$; concatenation preserves sub-Gaussianity with $L = \|\bar{\mathbf{z}}_t\|_{\psi_2} = O(1)$ in $(\tau_w, \sigma_\epsilon, \gamma)$.

For i.i.d. sub-Gaussian samples in $\mathbb{R}^d$ with $d = 2K$, the standard sample covariance operator-norm deviation bound (e.g., matrix Bernstein / sub-Gaussian covariance concentration) yields equation A.8:

$$\left\| \widehat{\bar{\mathbf{\Sigma}}} - \bar{\mathbf{\Sigma}} \right\|_{\mathrm{op}} \;\leq\; C\,L^2 \left( \sqrt{\frac{d + \log(1/\delta)}{M}} \;+\; \frac{d + \log(1/\delta)}{M} \right).$$

If $M \geq C_{\mathrm{mc}}(2K + \log(1/\delta))/\underline{\sigma}^2$, then $\|\widehat{\bar{\mathbf{\Sigma}}} - \bar{\mathbf{\Sigma}}\|_{\mathrm{op}} \leq \underline{\sigma}/2$. By Weyl's inequality on the restriction to $S$,

$$\lambda_{\min}\!\left(\widehat{\bar{\mathbf{\Sigma}}}\Big|_S\right) \;\geq\; \lambda_{\min}\!\left(\bar{\mathbf{\Sigma}}\Big|_S\right) \;-\; \left\| \widehat{\bar{\mathbf{\Sigma}}} - \bar{\mathbf{\Sigma}} \right\|_{\mathrm{op}} \;\geq\; \underline{\sigma} - \tfrac{1}{2}\underline{\sigma} \;=\; \tfrac{1}{2}\underline{\sigma},$$

which is equation A.9. $\qquad\square$

**Lemma A.9** (Fisher two-channel quadratic: gradient and Hessian)**.** Consider the *population* Fisher-weighted quadratic from Eq. equation 4.1 written in the two-channel parameterization

$$\mathcal{L}(\bar{\mathbf{W}}) \;:=\; \frac{1}{2}\,\mathrm{tr}\!\left(\mathbf{\Gamma}\,\bar{\mathbf{W}}\,\bar{\mathbf{\Sigma}}\,\bar{\mathbf{W}}^\top\right) \;-\; \mathrm{tr}\!\left(\mathbf{\Gamma}\,\mathbf{\Sigma}_{y\bar{z}}\,\bar{\mathbf{W}}^\top\right) \;+\; \mathrm{const}, \tag{A.10}$$

where $\bar{\mathbf{W}} \in \mathbb{R}^{K \times 2K}$, $\bar{\mathbf{\Sigma}} = \mathbb{E}[\bar{\mathbf{z}}_t \bar{\mathbf{z}}_t^\top]$, $\mathbf{\Sigma}_{y\bar{z}} = \mathbb{E}[\mathbf{y}_{t+1} \bar{\mathbf{z}}_t^\top]$, and $\mathbf{\Gamma} = \mathbb{E}[\mathbf{F}(\mathbf{p}_{t+1}^{\mathrm{PO}})]$. Then

$$\nabla_{\bar{\mathbf{W}}}\mathcal{L}(\bar{\mathbf{W}}) \;=\; \mathbf{\Gamma}\!\left(\bar{\mathbf{W}}\,\bar{\mathbf{\Sigma}} - \mathbf{\Sigma}_{y\bar{z}}\right) \;=:\; \mathbf{E}(\bar{\mathbf{W}}) \in \mathbb{R}^{K \times 2K}, \tag{A.11}$$

$$\nabla^2\mathcal{L}(\bar{\mathbf{W}})[\Delta] \;=\; \mathbf{\Gamma}\,\Delta\,\bar{\mathbf{\Sigma}} \qquad (\Delta \in \mathbb{R}^{K \times 2K}). \tag{A.12}$$

*Proof.* Write $\mathcal{L} = \mathcal{L}_1 + \mathcal{L}_2 + \mathrm{const}$ with

$$\mathcal{L}_1(\bar{\mathbf{W}}) := \tfrac{1}{2}\,\mathrm{tr}(\mathbf{\Gamma}\,\bar{\mathbf{W}}\,\bar{\mathbf{\Sigma}}\,\bar{\mathbf{W}}^\top), \qquad \mathcal{L}_2(\bar{\mathbf{W}}) := -\,\mathrm{tr}(\mathbf{\Gamma}\,\mathbf{\Sigma}_{y\bar{z}}\,\bar{\mathbf{W}}^\top).$$

Using the Frobenius inner product $\langle A, B \rangle = \mathrm{tr}(A^\top B)$ and $d(\bar{\mathbf{W}}\,\bar{\mathbf{\Sigma}}\,\bar{\mathbf{W}}^\top) = (d\bar{\mathbf{W}})\,\bar{\mathbf{\Sigma}}\,\bar{\mathbf{W}}^\top + \bar{\mathbf{W}}\,\bar{\mathbf{\Sigma}}\,(d\bar{\mathbf{W}})^\top$ with $\bar{\mathbf{\Sigma}}^\top = \bar{\mathbf{\Sigma}}$, $\mathbf{\Gamma}^\top = \mathbf{\Gamma}$,

$$d\mathcal{L}_1 = \left\langle \mathbf{\Gamma}\,\bar{\mathbf{W}}\,\bar{\mathbf{\Sigma}},\, d\bar{\mathbf{W}} \right\rangle, \qquad d\mathcal{L}_2 = \left\langle -\,\mathbf{\Gamma}\,\mathbf{\Sigma}_{y\bar{z}},\, d\bar{\mathbf{W}} \right\rangle,$$

so $d\mathcal{L} = \langle \mathbf{\Gamma}(\bar{\mathbf{W}}\bar{\mathbf{\Sigma}} - \mathbf{\Sigma}_{y\bar{z}}),\, d\bar{\mathbf{W}} \rangle$, which proves equation A.11. For the Hessian, with $\bar{\mathbf{W}}(\epsilon) = \bar{\mathbf{W}} + \epsilon\Delta$,

$$\frac{d}{d\epsilon} \nabla_{\bar{\mathbf{W}}}\mathcal{L}(\bar{\mathbf{W}}(\epsilon)) = \mathbf{\Gamma}\,\Delta\,\bar{\mathbf{\Sigma}},$$

establishing equation A.12. Equivalently, $d^2\mathcal{L}[\Delta, \Delta] = \mathrm{tr}(\Delta^\top \mathbf{\Gamma}\,\Delta\,\bar{\mathbf{\Sigma}}) = \|\mathbf{\Gamma}^{1/2}\Delta\bar{\mathbf{\Sigma}}^{1/2}\|_F^2 \geq 0$. $\quad\square$

**Lemma A.10** (PL inequality on $\mathbf{1}^\perp$)**.** Under mixture exploration (Lemma A.1) so that Lemma A.7 holds, the population Fisher–weighted quadratic

$$\mathcal{L}(\bar{\mathbf{W}}) \;=\; \tfrac{1}{2}\,\mathrm{tr}\!\left(\mathbf{\Gamma}\,\bar{\mathbf{W}}\,\bar{\mathbf{\Sigma}}\,\bar{\mathbf{W}}^\top\right) \;-\; \mathrm{tr}\!\left(\mathbf{\Gamma}\,\mathbf{\Sigma}_{y\bar{z}}\,\bar{\mathbf{W}}^\top\right) \;+\; \mathrm{const}$$

satisfies, for any global minimizer $\bar{\mathbf{W}}^\star$,

$$\mathcal{L}(\bar{\mathbf{W}}) - \mathcal{L}(\bar{\mathbf{W}}^\star) \;\leq\; \frac{1}{2\mu}\left\| \nabla_{\bar{\mathbf{W}}}\mathcal{L}(\bar{\mathbf{W}}) \right\|_F^2, \qquad \mu := \lambda_{\min}^+\!\left(\mathbf{\Gamma}\big|_{\mathbf{1}^\perp}\right) \lambda_{\min}^+\!\left(\bar{\mathbf{\Sigma}}\big|_S\right), \tag{A.13}$$

where $S := \mathbf{1}^\perp \oplus \mathbf{1}^\perp$. Moreover, by Lemma A.1 and Lemma A.7,

$$\lambda_{\min}^+\!\left(\mathbf{\Gamma}\big|_{\mathbf{1}^\perp}\right) \;\geq\; \frac{\gamma}{K}, \qquad \lambda_{\min}^+\!\left(\bar{\mathbf{\Sigma}}\big|_S\right) \;\geq\; \underline{\sigma} > 0, \qquad \Rightarrow \qquad \mu \;\geq\; \frac{\gamma}{K}\cdot\underline{\sigma}. \tag{A.14}$$

*Proof.* Let $\Delta := \bar{\mathbf{W}} - \bar{\mathbf{W}}^\star$ and define $\mathbf{X} := \mathbf{\Gamma}^{1/2}\,\Delta\,\bar{\mathbf{\Sigma}}^{1/2}$. By Lemma A.9,

$$\nabla_{\bar{\mathbf{W}}}\mathcal{L}(\bar{\mathbf{W}}) = \mathbf{\Gamma}\,\Delta\,\bar{\mathbf{\Sigma}} = \mathbf{\Gamma}^{1/2}\mathbf{X}\bar{\mathbf{\Sigma}}^{1/2}, \qquad \mathcal{L}(\bar{\mathbf{W}}) - \mathcal{L}(\bar{\mathbf{W}}^\star) = \tfrac{1}{2}\,\mathrm{tr}\!\left(\mathbf{\Gamma}\,\Delta\,\bar{\mathbf{\Sigma}}\,\Delta^\top\right) = \tfrac{1}{2}\,\|\mathbf{X}\|_F^2.$$

For any PSD $\mathbf{A}, \mathbf{B}$ and any matrix $\mathbf{X}$, $\|\mathbf{A}^{1/2}\mathbf{X}\mathbf{B}^{1/2}\|_F^2 = \mathrm{tr}(\mathbf{X}^\top \mathbf{A}\mathbf{X}\mathbf{B}) \geq \lambda_{\min}(\mathbf{A})\lambda_{\min}(\mathbf{B})\|\mathbf{X}\|_F^2$. Applying this with $\mathbf{A} = \mathbf{\Gamma}\big|_{\mathbf{1}^\perp}$ and $\mathbf{B} = \bar{\mathbf{\Sigma}}\big|_S$, we obtain

$$\left\| \nabla_{\bar{\mathbf{W}}}\mathcal{L}(\bar{\mathbf{W}}) \right\|_F^2 = \left\| \mathbf{\Gamma}^{1/2}\mathbf{X}\bar{\mathbf{\Sigma}}^{1/2} \right\|_F^2 \geq \lambda_{\min}^+\!\left(\mathbf{\Gamma}\big|_{\mathbf{1}^\perp}\right) \lambda_{\min}^+\!\left(\bar{\mathbf{\Sigma}}\big|_S\right) \|\mathbf{X}\|_F^2 = 2\,\mu\left(\mathcal{L}(\bar{\mathbf{W}}) - \mathcal{L}(\bar{\mathbf{W}}^\star)\right),$$

which is equivalent to the PL inequality equation A.13. The bound equation A.14 follows from Lemma A.1 (giving $\mathbf{\Gamma}\big|_{\mathbf{1}^\perp} \succeq (\gamma/K)I$) and Lemma A.7 (giving $\bar{\mathbf{\Sigma}}\big|_S \succeq \underline{\sigma}I$). $\quad\square$

## A.2 REWARD-SHOCK ROBUSTNESS

*Proof of Theorem 4.8.* Under the $s$-CRN coupling (Def. 4.7), the baseline and perturbed runs share the same task $\mathbf{w}$, the same uniforms $\{U_t\}$ for sampling, and the same noise sequence $\{\epsilon_t\}$; they differ only by feeding a shocked reward $\widetilde{r}_s = r_s + \delta_r$ at round $s$. Define

$$\Delta\mathbf{p}_{t+1} := \widetilde{\mathbf{p}}_{t+1} - \mathbf{p}_{t+1}, \quad \Delta\mathbf{n}_t := \widetilde{\mathbf{n}}_t - \mathbf{n}_t, \quad \Delta\mathbf{g}_t := \widetilde{\mathbf{g}}_t - \mathbf{g}_t, \quad a_{t+1} := \mathbb{E}[\|\Delta\mathbf{p}_{t+1}\| \mid \mathcal{F}_{s-1}],$$

and the normalized accumulators $\bar{\mathbf{n}}_t := \mathbf{n}_t/t$, $\bar{\mathbf{g}}_t := \mathbf{g}_t/t$ (and their deltas analogously).

Pathwise relations for counts and rewards. With $\mathbf{x}_t = \mathbf{e}_{A_t}$ and $r_t = \langle \mathbf{w}, \mathbf{x}_t \rangle + \epsilon_t$,

$$\mathbf{n}_t = \sum_{u=1}^{t} \mathbf{x}_u, \qquad \mathbf{g}_t = \sum_{u=1}^{t} r_u \mathbf{x}_u = \mathrm{Diag}(\mathbf{n}_t)\,\mathbf{w} + \sum_{u=1}^{t} \epsilon_u\,\mathbf{x}_u.$$

Let $h_t := \sum_{u=1}^{t} \epsilon_u \mathbf{x}_u$ and $\widetilde{h}_t := \sum_{u=1}^{t} \epsilon_u \widetilde{\mathbf{x}}_u$. For the perturbed run, only round $s$ changes in the fed reward, hence

$$\widetilde{\mathbf{g}}_t = \mathrm{Diag}(\widetilde{\mathbf{n}}_t)\,\mathbf{w} + \widetilde{h}_t + \mathbf{1}\{t \geq s\}\,\delta_r\,\widetilde{\mathbf{x}}_s.$$

Therefore, for $t \geq s$,

$$\Delta\bar{\mathbf{g}}_t = \frac{1}{t}\,\mathrm{Diag}(\mathbf{w})\,\Delta\mathbf{n}_t + \frac{1}{t}\,\Delta h_t + \frac{1}{t}\,\delta_r\,\widetilde{\mathbf{x}}_s, \qquad \Delta h_t := \widetilde{h}_t - h_t. \tag{A.15}$$

Two-channel linearization of projected logits. Using the (population-optimal) one-step normal form aligned with equation 3.2 and projecting away the softmax gauge,

$$\Delta\mathbf{y}_{t+1} := \mathrm{Proj}\!\left(\widehat{\mathbf{s}}_{t+1}^{(\mathrm{pert})} - \widehat{\mathbf{s}}_{t+1}^{(\mathrm{base})}\right) = \frac{c}{t}\left(\mathbf{V}\,\Delta\mathbf{n}_t + \mathbf{U}\,\Delta\mathbf{g}_t\right) = c\left(\mathbf{V}\,\Delta\bar{\mathbf{n}}_t + \mathbf{U}\,\Delta\bar{\mathbf{g}}_t\right). \tag{A.16}$$

Softmax is $1/2$-Lipschitz on $\mathbf{1}^{\perp}$, and the $\gamma$-mixture equation 3.3 scales the sensitivity by $(1-\gamma)$, hence

$$\|\Delta\mathbf{p}_{t+1}\| \leq (1-\gamma)\frac{1}{2}\|\Delta\mathbf{y}_{t+1}\|. \tag{A.17}$$

Combining equation A.15–equation A.17 and conditioning on $\mathcal{F}_{s-1}$ yields

$$\begin{aligned}
a_{t+1} \leq &\frac{c(1-\gamma)}{2}\,\mathbb{E}\!\left[\left\|\left(\mathbf{V}+\mathbf{U}\,\mathrm{Diag}(\mathbf{w})\right)\Delta\bar{\mathbf{n}}_t + \mathbf{U}\,\Delta\bar{h}_t + \mathbf{U}\,\frac{\delta_r}{t}\,\widetilde{\mathbf{x}}_s\right\| \,\Big|\, \mathcal{F}_{s-1}\right] \\
\leq &\frac{c(1-\gamma)}{2t}\left(\|\mathbf{V}+\mathbf{U}\,\mathrm{Diag}(\mathbf{w})\|_{\mathrm{op}}\,\mathbb{E}[\|\Delta\mathbf{n}_t\| \mid \mathcal{F}_{s-1}]\right. \\
&\left.+ \|\mathbf{U}\|_{\mathrm{op}}\,\mathbb{E}[\|\Delta h_t\| \mid \mathcal{F}_{s-1}] + \|\mathbf{U}\|_{\mathrm{op}}\,|\delta_r|\right).
\end{aligned} \tag{A.18}$$

**CRN coupling controls** $\mathbb{E}\|\Delta\mathbf{n}_t\|$ **and** $\mathbb{E}\|\Delta h_t\|$. By inverse-CDF coupling,

$$\mathbb{P}(\widetilde{A}_u \neq A_u \mid \mathcal{F}_{u-1}) = \tfrac{1}{2}\|\Delta\mathbf{p}_u\|_1 \leq \frac{\sqrt{K}}{2}\|\Delta\mathbf{p}_u\|.$$

Since $\|\Delta\mathbf{x}_u\| \leq \sqrt{2}$,

$$\mathbb{E}[\|\Delta\mathbf{x}_u\| \mid \mathcal{F}_{u-1}] \leq \sqrt{\tfrac{K}{2}}\,\|\Delta\mathbf{p}_u\|. \tag{A.19}$$

Summing equation A.19 from $u=s$ to $t$ and conditioning on $\mathcal{F}_{s-1}$,

$$\mathbb{E}[\|\Delta\mathbf{n}_t\| \mid \mathcal{F}_{s-1}] \leq \sqrt{\tfrac{K}{2}}\sum_{u=s}^{t} a_u. \tag{A.20}$$

For the noise accumulator, using the zero-mean $\sigma_\epsilon$-sub-Gaussian assumption (hence $\mathbb{E}[|\epsilon_u| \mid \mathcal{F}_{u-1}] \leq C_\epsilon := \sqrt{2/\pi}\,\sigma_\epsilon$) gives

$$\mathbb{E}[\|\Delta h_t\| \mid \mathcal{F}_{s-1}] \leq C_\epsilon\sqrt{\tfrac{K}{2}}\sum_{u=s}^{t} a_u. \tag{A.21}$$

**A Grönwall-type recursion and its solution.** Plugging equation A.20–equation A.21 into equation A.18, define

$$a := \frac{c(1-\gamma)}{2}\|\mathbf{U}\|_{\mathrm{op}}, \qquad b := \frac{c(1-\gamma)}{2}\sqrt{\frac{K}{2}}\left(\|\mathbf{V}+\mathbf{U}\,\mathrm{Diag}(\mathbf{w})\|_{\mathrm{op}} + C_\epsilon\|\mathbf{U}\|_{\mathrm{op}}\right),$$

and obtain, for $t \geq s$,

$$a_{t+1} \leq \frac{a}{t} |\delta_r| + \frac{b}{t} \sum_{u=s}^{t} a_u. \tag{A.22}$$

Let $S_t := \sum_{u=s}^{t} a_u$ with $S_{s-1} = 0$. Then

$$S_{t+1} \leq \left(1 + \frac{b}{t}\right) S_t + \frac{a}{t} |\delta_r|. \tag{A.23}$$

Unrolling equation A.23 and using the standard Gamma-ratio bound yields a constant $C_b > 0$ (depending only on $b$) such that

$$S_t \leq a\, C_b\, |\delta_r| \sum_{u=s}^{t-1} \frac{1}{u} \left(\frac{t}{u}\right)^b \leq \frac{a\, C_b}{b} \left(\frac{t}{s}\right)^b |\delta_r|.$$

Returning to equation A.22,

$$a_{t+1} \leq \frac{a}{t} |\delta_r| + \frac{b}{t} S_t \leq \frac{a}{t} |\delta_r| + \frac{a\, C_b}{t} \left(\frac{t}{s}\right)^b |\delta_r| = \frac{a(1 + C_b)}{s} \left(\frac{t}{s}\right)^{b-1} |\delta_r|.$$

Therefore, for any $1 \leq s \leq t$,

$$\mathbb{E}\big[\, \|\Delta \widehat{\mathbf{p}}_{t+1}^s\|_2 \mid \mathcal{F}_{s-1} \big] \leq \frac{a(1 + C_b)}{s} \left(\frac{t}{s}\right)^{b-1} |\delta_r|. \tag{A.24}$$

In particular, if the learning-rate constant $c$ is small enough so that $b < 1$, then $\left(\frac{t}{s}\right)^{b-1} \to 0$ as $t \to \infty$, and the one-shot impact decays to zero:

$$\lim_{t \to \infty} \mathbb{E}\big[\, \|\Delta \widehat{\mathbf{p}}_{t+1}^s\|_2 \mid \mathcal{F}_{s-1} \big] = 0.$$

This completes the proof. $\qquad\square$

## A.3   KL Divergence vs. Fisher-weighted Duadratic

*Proof of Theorem 4.1.* Recall that $N$ denotes the trajectory length in the dataset construction of Sec. 4, and expectations below average over $(\tau, t)$ with $t \in \{1, \ldots, N-1\}$ and $\tau \in \mathcal{D}$. Let $\widetilde{\mathbf{p}}(\mathbf{s}) := \mathrm{softmax}(\mathbf{s})$ and define the projected logit error

$$\Delta_{t+1} := \mathrm{Proj}\big(\widehat{\mathbf{s}}_{t+1} - \mathbf{s}_{t+1}^{\mathrm{PO}}\big) \in \mathbf{1}^\perp, \qquad \mathrm{Proj} := \mathbf{I} - \frac{1}{K} \mathbf{1}\mathbf{1}^\top.$$

Write the (softmax) Fisher matrix as $\mathbf{F}(\mathbf{p}) := \mathrm{Diag}(\mathbf{p}) - \mathbf{p}\mathbf{p}^\top$ and recall from Eq. equation 4.1 that

$$\mathcal{L}(\boldsymbol{\theta}) = \mathbb{E}\left[ \frac{1}{N-1} \sum_{t=1}^{N-1} \Delta_{t+1}^\top \boldsymbol{\Gamma} \Delta_{t+1} \right], \qquad \boldsymbol{\Gamma} := \frac{1}{N-1} \mathbb{E}\left[ \sum_{t=1}^{N-1} \mathbf{F}(\mathbf{p}_t^{\mathrm{PO}}) \right].$$

By Lemma A.1 (mixture curvature), along the teacher's $\gamma$-mixture policy we have the spectral sandwich on $\mathbf{1}^\perp$:

$$\frac{\gamma}{K} \mathbf{I} \preceq \mathbf{F}(\mathbf{p}_t^{\mathrm{PO}})\Big|_{\mathbf{1}^\perp} \preceq \frac{1}{2} \mathbf{I}. \tag{A.25}$$

By convexity of the PSD cone, the same bounds transfer to $\boldsymbol{\Gamma}$ on $\mathbf{1}^\perp$.

**Upper bound.** Mixing with the uniform distribution is a Markov kernel; by data processing for $f$-divergences,

$$D_{\mathrm{KL}}\big(\mathbf{p}_{t+1}^{\mathrm{PO}} \,\|\, \widehat{\mathbf{p}}_{t+1}\big) \leq D_{\mathrm{KL}}\big(\widetilde{\mathbf{p}}(\mathbf{s}_{t+1}^{\mathrm{PO}}) \,\|\, \widetilde{\mathbf{p}}(\widehat{\mathbf{s}}_{t+1})\big) = D_{\mathrm{KL}}\big(\widetilde{\mathbf{p}}(\mathbf{s}_{t+1}^{\mathrm{PO}}) \,\|\, \widetilde{\mathbf{p}}(\mathbf{s}_{t+1}^{\mathrm{PO}} + \Delta_{t+1})\big).$$

Let $\phi(\mathbf{s}) := \log \sum_i e^{s_i}$; then $\nabla \phi = \mathrm{softmax}$ and $\nabla^2 \phi(\mathbf{s}) = \mathbf{F}(\widetilde{\mathbf{p}}(\mathbf{s}))$. The Bregman integral form of KL yields, for any $\Delta \in \mathbf{1}^\perp$,

$$D_{\mathrm{KL}}\big(\widetilde{\mathbf{p}}(\mathbf{s}) \,\|\, \widetilde{\mathbf{p}}(\mathbf{s} + \Delta)\big) = \int_0^1 (1 - \tau)\, \Delta^\top \mathbf{F}\big(\widetilde{\mathbf{p}}(\mathbf{s} + \tau\Delta)\big) \Delta\, d\tau. \tag{A.26}$$

Using $\|\mathbf{F}(\cdot)\big|_{\mathbf{1}^\perp}\|_{\mathrm{op}} \leq \frac{1}{2}$ and $\int_0^1 (1 - \tau)\, d\tau = \frac{1}{2}$,

$$D_{\mathrm{KL}}\big(\mathbf{p}_{t+1}^{\mathrm{PO}} \,\|\, \widehat{\mathbf{p}}_{t+1}\big) \leq \tfrac{1}{4} \|\Delta_{t+1}\|_2^2.$$

From equation A.25, $\Delta_{t+1}^{\top} \mathbf{\Gamma} \, \Delta_{t+1} \geq \frac{\gamma}{K} \|\Delta_{t+1}\|_2^2$, hence $\|\Delta_{t+1}\|_2^2 \leq \frac{K}{\gamma} \Delta_{t+1}^{\top} \mathbf{\Gamma} \, \Delta_{t+1}$. Averaging over $t$ and taking expectation,

$$\mathbb{E}\left[ \frac{1}{N-1} \sum_{t=1}^{N-1} D_{\mathrm{KL}}\big(\mathbf{p}_{t+1}^{\mathrm{PO}} \,\|\, \widehat{\mathbf{p}}_{t+1}\big) \right] \leq \frac{K}{4\gamma} \, \mathcal{L}(\boldsymbol{\theta}).$$

**Lower bound.** Pinsker's inequality implies $D_{\mathrm{KL}}(\mathbf{p}\|\mathbf{q}) \geq \frac{1}{2}\|\mathbf{p} - \mathbf{q}\|_1^2 \geq \frac{1}{2}\|\mathbf{p} - \mathbf{q}\|_2^2$. Let $f(\mathbf{s}) := (1 - \gamma)\,\widetilde{\mathbf{p}}(\mathbf{s}) + \gamma \, \mathbf{1}/K$ so that $\mathbf{p}_{t+1}^{\mathrm{PO}} = f(\mathbf{s}_{t+1}^{\mathrm{PO}})$ and $\widehat{\mathbf{p}}_{t+1} = f(\widehat{\mathbf{s}}_{t+1})$. By the mean-value integral,

$$\mathbf{p}_{t+1}^{\mathrm{PO}} - \widehat{\mathbf{p}}_{t+1} = \int_0^1 \nabla f(\mathbf{s}_{t+1}^{\mathrm{PO}} + \tau \Delta_{t+1}) \, \Delta_{t+1} \, d\tau = (1 - \gamma) \int_0^1 \mathbf{F}\big(\widetilde{\mathbf{p}}(\mathbf{s}_{t+1}^{\mathrm{PO}} + \tau \Delta_{t+1})\big) \, \Delta_{t+1} \, d\tau,$$

whence $\|\mathbf{p}_{t+1}^{\mathrm{PO}} - \widehat{\mathbf{p}}_{t+1}\|_2 \leq (1 - \gamma) \cdot \frac{1}{2} \, \|\Delta_{t+1}\|_2$ and thus

$$\|\Delta_{t+1}\|_2^2 \geq \frac{4}{(1 - \gamma)^2} \, \|\mathbf{p}_{t+1}^{\mathrm{PO}} - \widehat{\mathbf{p}}_{t+1}\|_2^2.$$

Using the upper side of equation A.25, $\Delta_{t+1}^{\top} \mathbf{\Gamma} \, \Delta_{t+1} \leq \frac{1}{2} \, \|\Delta_{t+1}\|_2^2$, we obtain

$$\mathcal{L}(\boldsymbol{\theta}) \leq \frac{1}{2} \, \mathbb{E}\left[ \frac{1}{N-1} \sum_{t=1}^{N-1} \|\Delta_{t+1}\|_2^2 \right] \leq \frac{2}{(1 - \gamma)^2} \, \mathbb{E}\left[ \frac{1}{N-1} \sum_{t=1}^{N-1} \|\mathbf{p}_{t+1}^{\mathrm{PO}} - \widehat{\mathbf{p}}_{t+1}\|_2^2 \right].$$

Finally, combining with Pinsker yields

$$\mathbb{E}\left[ \frac{1}{N-1} \sum_{t=1}^{N-1} D_{\mathrm{KL}}\big(\mathbf{p}_{t+1}^{\mathrm{PO}} \,\|\, \widehat{\mathbf{p}}_{t+1}\big) \right] \geq \frac{1}{2} \, \mathbb{E}\left[ \frac{1}{N-1} \sum_{t=1}^{N-1} \|\mathbf{p}_{t+1}^{\mathrm{PO}} - \widehat{\mathbf{p}}_{t+1}\|_2^2 \right] \geq \frac{(1 - \gamma)^2}{4} \, \mathcal{L}(\boldsymbol{\theta}).$$

This proves the two-sided bound. $\qquad \square$

### A.4 Closed-Loop Imitation of Policy Optimization

*Proof of Theorem 4.2.* By Lemma A.4, the *projected* student logits are linear in the normalized statistics:

$$\mathrm{Proj}\,\widehat{\mathbf{s}}_{t+1} = \bar{\mathbf{W}}\,\bar{\mathbf{z}}_t, \qquad \bar{\mathbf{W}} := [\,\mathbf{W}_n \ \ \mathbf{W}_g\,] \in \mathbb{R}^{K \times 2K}, \quad \bar{\mathbf{z}}_t := \begin{pmatrix} \bar{\mathbf{n}}_t \\ \bar{\mathbf{g}}_t \end{pmatrix} \in \mathbb{R}^{2K}.$$

For the teacher generated by equation 3.2 with $\eta_t = c/t$, we likewise have

$$\mathbf{y}_{t+1} := \mathrm{Proj}\,\mathbf{s}_{t+1}^{\mathrm{PO}} = \frac{c}{t}\,\mathrm{Proj}\big(\mathbf{V}\,\mathbf{n}_t + \mathbf{U}\,\mathbf{g}_t\big) = \bar{\mathbf{W}}_\star\,\bar{\mathbf{z}}_t, \qquad \bar{\mathbf{W}}_\star := \mathrm{Proj}\,[\,\mathbf{V} \ \ \mathbf{U}\,] \in \mathbb{R}^{K \times 2K},$$

so the *labels are realizable* by the same two-channel structure.

Lemma A.5 gives the population Fisher–weighted quadratic form

$$\mathcal{L}(\boldsymbol{\theta}) = \frac{1}{2}\,\mathrm{tr}(\mathbf{\Gamma}\,\bar{\mathbf{W}}\,\bar{\mathbf{\Sigma}}\,\bar{\mathbf{W}}^{\top}) - \mathrm{tr}(\mathbf{\Gamma}\,\mathbf{\Sigma}_{y\bar{z}}\,\bar{\mathbf{W}}^{\top}) + \mathrm{const}, \qquad \mathbf{\Gamma} := \mathbb{E}\big[\mathbf{F}(\mathbf{p}_{t+1}^{\mathrm{PO}})\big], \quad \bar{\mathbf{\Sigma}} := \mathbb{E}[\bar{\mathbf{z}}_t \bar{\mathbf{z}}_t^{\top}].$$

Using realizability $\mathbf{y}_{t+1} = \bar{\mathbf{W}}_\star \bar{\mathbf{z}}_t$, we have $\mathbf{\Sigma}_{y\bar{z}} = \mathbb{E}[\mathbf{y}_{t+1}\bar{\mathbf{z}}_t^{\top}] = \mathbb{E}[\bar{\mathbf{W}}_\star \bar{\mathbf{z}}_t \bar{\mathbf{z}}_t^{\top}] = \bar{\mathbf{W}}_\star \bar{\mathbf{\Sigma}}$. Substituting gives the *completed-square* form

$$\mathcal{L}(\boldsymbol{\theta}) = \frac{1}{2}\,\mathrm{tr}\Big(\mathbf{\Gamma}\,(\bar{\mathbf{W}} - \bar{\mathbf{W}}_\star)\,\bar{\mathbf{\Sigma}}\,(\bar{\mathbf{W}} - \bar{\mathbf{W}}_\star)^{\top}\Big) + \mathrm{const}.$$

By Lemma A.1 (mixture curvature), $\mathbf{\Gamma}\big|_{\mathbf{1}^{\perp}} \succeq (\gamma/K)\,I$; by Lemma A.7, $\bar{\mathbf{\Sigma}}\big|_S \succeq \underline{\sigma}\,I$ on $S := \mathbf{1}^{\perp} \oplus \mathbf{1}^{\perp}$ under Assumption 3.1. Hence the quadratic is *strongly convex* in $\bar{\mathbf{W}}$ on $S$ and has the unique minimizer $\bar{\mathbf{W}} = \bar{\mathbf{W}}_\star$. Therefore, for any history prefix $\mathcal{H}_t$,

$$\mathrm{Proj}\,\widehat{\mathbf{s}}_{t+1}(\mathcal{H}_t; \boldsymbol{\theta}^\star) = \bar{\mathbf{W}}_\star\,\bar{\mathbf{z}}_t = \mathrm{Proj}\,\mathbf{s}_{t+1}^{\mathrm{PO}}(\mathcal{H}_t).$$

Since softmax is invariant to additive constants (the projected logits remove exactly the $\mathrm{span}\{\mathbf{1}\}$ gauge), the induced policies coincide: $\widehat{\mathbf{p}}_{t+1}(\mathcal{H}_t; \boldsymbol{\theta}^\star) = \mathbf{p}_{t+1}^{\mathrm{PO}}(\mathcal{H}_t).$ $\qquad \square$

*Proof of Theorem 4.3.* By Lemma A.4, the projected student logits are linear in the normalized statistics: $\text{Proj}\,\widehat{\mathbf{s}}_{t+1} = \bar{\mathbf{W}}\,\bar{\mathbf{z}}_t$ with $\bar{\mathbf{W}} = [\mathbf{W}_n\ \mathbf{W}_g]$ and $\bar{\mathbf{z}}_t = (\bar{\mathbf{n}}_t^\top, \bar{\mathbf{g}}_t^\top)^\top$. Let $\mathbf{y}_{t+1} := \text{Proj}\,(\mathbf{s}_{t+1}^{\text{PO}})$. Lemma A.5 and Lemma A.6 give the population and empirical Fisher–weighted quadratic forms, and Theorem 4.2 (under Assumption 3.1) guarantees realizability: there exists a unique $\bar{\mathbf{W}}_\star$ such that $\mathbf{y}_{t+1} = \bar{\mathbf{W}}_\star \bar{\mathbf{z}}_t$ for all prefixes. Moreover, Lemma A.1 implies $\mathbf{\Gamma}\big|_{\mathbf{1}^\perp} \succeq (\gamma/K)I$, and Lemma A.7 implies $\bar{\mathbf{\Sigma}}\big|_S \succeq \underline{\sigma}I$ with $\underline{\sigma} \asymp c_\lambda/t$ on $S := \mathbf{1}^\perp \oplus \mathbf{1}^\perp$, so the population quadratic in $\bar{\mathbf{W}}$ is strongly convex on $S$ with unique minimizer $\bar{\mathbf{W}}_\star$. For the empirical problem, Lemma A.8 yields (for i.i.d. rollouts)

$$\|\widehat{\bar{\mathbf{\Sigma}}} - \bar{\mathbf{\Sigma}}\|_{\text{op}} \ \leq\ C_1 L^2 \Big( \sqrt{\tfrac{2K + \log(1/\delta)}{M}} + \tfrac{2K + \log(1/\delta)}{M} \Big).$$

Taking $M \geq C\,t^2(2K + \log(1/\delta))/c_\lambda^2$ (for a large enough absolute $C$ depending on $C_1, L$) guarantees $\widehat{\bar{\mathbf{\Sigma}}}\big|_S \succeq \tfrac{1}{2}\underline{\sigma}I$, while $\widehat{\mathbf{\Gamma}}\big|_{\mathbf{1}^\perp} \succeq (\gamma/K)I$ by mixture curvature. Plugging the realizable labels $\mathbf{y}_{t+1} = \bar{\mathbf{W}}_\star \bar{\mathbf{z}}_t$ into Lemma A.6 gives

$$\widehat{\mathcal{L}}(\bar{\mathbf{W}}) = \tfrac{1}{2}\,\text{tr}\big( \widehat{\mathbf{\Gamma}}\,(\bar{\mathbf{W}} - \bar{\mathbf{W}}_\star)\,\widehat{\bar{\mathbf{\Sigma}}}\,(\bar{\mathbf{W}} - \bar{\mathbf{W}}_\star)^\top \big) + \text{const},$$

so the empirical minimizer satisfies $\bar{\mathbf{W}} = \bar{\mathbf{W}}_\star$ whenever $\widehat{\bar{\mathbf{\Sigma}}}\big|_S \succ 0$ (which holds with probability $\geq 1 - \delta$). Thus $\text{Proj}\,\widehat{\mathbf{s}}_{t+1} = \text{Proj}\,\mathbf{s}_{t+1}^{\text{PO}}$ and consequently $\widehat{\mathbf{p}}_{t+1}(\mathcal{H}_t; \widehat{\boldsymbol{\theta}}) = \mathbf{p}_{t+1}^{\text{PO}}(\mathcal{H}_t)$ for the fixed test history $\mathcal{H}_t$. For the expected mismatch, on the high-probability event the error is 0, and on its complement a crude bound together with the $(1-\gamma)$ mixture factor yields $\|\widehat{\mathbf{p}}_{t+1} - \mathbf{p}_{t+1}^{\text{PO}}\|_2^2 \leq 2(1-\gamma)^2$, hence

$$\mathbb{E}_{\text{train}}\big[\|\widehat{\mathbf{p}}_{t+1} - \mathbf{p}_{t+1}^{\text{PO}}\|_2^2\big] \ \leq\ 0 \cdot (1 - \delta) + 2(1 - \gamma)^2 \cdot \delta = 2(1 - \gamma)^2 \delta.$$

This proves the theorem. $\qquad\qquad\qquad\qquad\qquad\qquad\qquad\qquad\qquad\qquad\qquad\qquad\qquad\qquad\qquad\square$

## A.5 CONVERGENCE OF THE FISHER-TRAINED TWO-CHANNEL LSA

We analyze the continuous-time gradient flow on the two-channel operator $\bar{\mathbf{W}}$ for the Fisher-weighted quadratic objective in equation A.10.

**Theorem A.11** (Exponential convergence of the $\bar{\mathbf{W}}$–flow). Assume mixture exploration and that Assumption 3.1, Lemma A.1, and Lemma A.7 hold. Consider the gradient flow for the *population* Fisher–weighted quadratic $\mathcal{L}$:

$$\dot{\bar{\mathbf{W}}}(t) \ =\ -\nabla_{\bar{\mathbf{W}}}\mathcal{L}\big(\bar{\mathbf{W}}(t)\big) \ =\ -\mathbf{\Gamma}\big(\bar{\mathbf{W}}(t)\,\bar{\mathbf{\Sigma}} - \mathbf{\Sigma}_{y\bar{z}}\big), \tag{A.27}$$

initialized at any $\bar{\mathbf{W}}(0)$. Let $\bar{\mathbf{W}}^\star$ be a global minimizer of $\mathcal{L}$. Then $\mathcal{L}\big(\bar{\mathbf{W}}(t)\big)$ is strictly decreasing along the flow and

$$\mathcal{L}\big(\bar{\mathbf{W}}(t)\big) - \mathcal{L}(\bar{\mathbf{W}}^\star) \ \leq\ \exp(-2\mu\,t)\,\big(\mathcal{L}\big(\bar{\mathbf{W}}(0)\big) - \mathcal{L}(\bar{\mathbf{W}}^\star)\big), \tag{A.28}$$

with

$$\mu \ :=\ \lambda_{\min}^+\big(\mathbf{\Gamma}\big|_{\mathbf{1}^\perp}\big)\,\lambda_{\min}^+\big(\bar{\mathbf{\Sigma}}\big|_S\big), \qquad S := \mathbf{1}^\perp \oplus \mathbf{1}^\perp,$$

and, by Lemma A.1 and Lemma A.7,

$$\lambda_{\min}^+\big(\mathbf{\Gamma}\big|_{\mathbf{1}^\perp}\big) \ \geq\ \frac{\gamma}{K}, \qquad \lambda_{\min}^+\big(\bar{\mathbf{\Sigma}}\big|_S\big) \ \geq\ \underline{\sigma} \ >\ 0, \qquad \Rightarrow \qquad \mu \ \geq\ \frac{\gamma}{K} \cdot \underline{\sigma}.$$

*Proof of Theorem A.11.* By Lemma A.9, we have

$$\nabla_{\bar{\mathbf{W}}}\mathcal{L}(\bar{\mathbf{W}}) = \mathbf{\Gamma}\big(\bar{\mathbf{W}}\bar{\mathbf{\Sigma}} - \mathbf{\Sigma}_{y\bar{z}}\big),$$

so the gradient flow $\dot{\bar{\mathbf{W}}}(t) = -\nabla_{\bar{\mathbf{W}}}\mathcal{L}(\bar{\mathbf{W}}(t))$ is well-defined. Along the trajectory,

$$\frac{d}{dt}\,\mathcal{L}\big(\bar{\mathbf{W}}(t)\big) = \big\langle \nabla_{\bar{\mathbf{W}}}\mathcal{L}\big(\bar{\mathbf{W}}(t)\big),\ \dot{\bar{\mathbf{W}}}(t) \big\rangle = -\big\|\nabla_{\bar{\mathbf{W}}}\mathcal{L}\big(\bar{\mathbf{W}}(t)\big)\big\|_F^2 \ \leq\ 0,$$

hence $\mathcal{L}\big(\bar{\mathbf{W}}(t)\big)$ is nonincreasing and strictly decreasing whenever $\nabla_{\bar{\mathbf{W}}}\mathcal{L}(\bar{\mathbf{W}}(t)) \neq \mathbf{0}$.

Next, apply the Polyak–Łojasiewicz (PL) inequality on the restricted subspace $S := \mathbf{1}^\perp \oplus \mathbf{1}^\perp$ (Lemma A.10). Let

$$\mu \ :=\ \lambda_{\min}^+\big(\mathbf{\Gamma}\big|_{\mathbf{1}^\perp}\big)\,\lambda_{\min}^+\big(\bar{\mathbf{\Sigma}}\big|_S\big).$$

Then for any global minimizer $\bar{\mathbf{W}}^\star$,

$$\mathcal{L}(\bar{\mathbf{W}}) - \mathcal{L}(\bar{\mathbf{W}}^\star) \leq \frac{1}{2\mu} \left\| \nabla_{\bar{\mathbf{W}}} \mathcal{L}(\bar{\mathbf{W}}) \right\|_F^2.$$

Combining this with the energy decay identity gives, for all $t \geq 0$,

$$\frac{d}{dt} \left( \mathcal{L}(\bar{\mathbf{W}}(t)) - \mathcal{L}(\bar{\mathbf{W}}^\star) \right) = -\left\| \nabla_{\bar{\mathbf{W}}} \mathcal{L}(\bar{\mathbf{W}}(t)) \right\|_F^2 \leq -2\mu \left( \mathcal{L}(\bar{\mathbf{W}}(t)) - \mathcal{L}(\bar{\mathbf{W}}^\star) \right).$$

By Grönwall's inequality,

$$\mathcal{L}\big(\bar{\mathbf{W}}(t)\big) - \mathcal{L}(\bar{\mathbf{W}}^\star) \leq \exp(-2\mu\, t) \big( \mathcal{L}\big(\bar{\mathbf{W}}(0)\big) - \mathcal{L}(\bar{\mathbf{W}}^\star) \big).$$

Finally, by mixture exploration (Lemma A.1) and the lower bound on the population second moment on $S$ (Lemma A.7),

$$\lambda_{\min}^+(\mathbf{\Gamma}|_{\mathbf{1}^\perp}) \geq \frac{\gamma}{K}, \qquad \lambda_{\min}^+(\bar{\mathbf{\Sigma}}|_S) \geq \underline{\sigma} > 0,$$

which implies $\mu \geq (\gamma/K)\,\underline{\sigma}$. Strict decrease of the loss holds except at stationary points; by Lemma A.9 together with the restricted positive definiteness, these coincide with the global minimizers on $\mathbf{1}^\perp$. $\qquad\square$

# B    EXPERIMENTAL DETAILS

This section provides additional details regarding the experimental setup, including specific hyperparameters for our method and the baselines, as well as the hardware and software environment used for all experiments. We also present supplementary experimental results that complement the main text.

## B.1    HYPERPARAMETER SENSITIVITY.

Figure 4 shows the impact of varying the number of in-context optimization rounds ($n$) and candidates per round ($k$) on AIME 2024. As shown in Figure 4a, performance improves substantially as the number of candidates ($k$) increases from 2 to 64, with Maj@16 accuracy more than doubling, then saturates beyond this point. Increasing the number of rounds ($n$) is broadly beneficial up to a peak near $n = 5$. A complementary grid and the corresponding latency/VRAM measures are reported in Appendix C.2 (Tables 8).

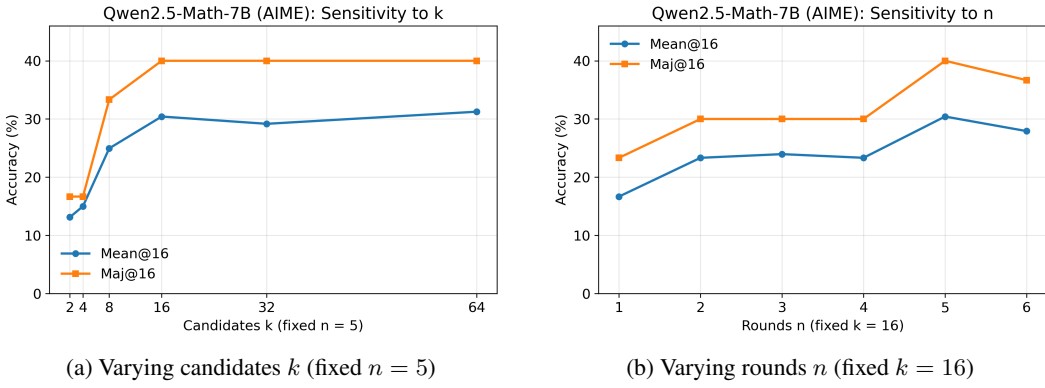

(a) Varying candidates $k$ (fixed $n = 5$)          (b) Varying rounds $n$ (fixed $k = 16$)

Figure 4: Hyperparameter sensitivity of ME-ICPO on AIME 2024 with Qwen2.5-Math-7B.

## B.2    HYPERPARAMETER AND IMPLEMENTATION DETAILS

**ME-ICPO (Our Method).** For our method, we use the primary settings described in the main text ($n = 5$ rounds of optimization, $k = 16$ candidates per round). Candidate chains-of-thought are sampled using a temperature of 0.6 and a top-p value of 0.95. The summarization step, which is a crucial component for managing context length, uses greedy decoding and is prompted to produce a concise summary of approximately 100 tokens, with a hard limit of 500 tokens, focusing on the reasoning strategy. The predictive answer distribution, used for the entropy calculation, is also estimated with a temperature of 0.6. The full text for all system and summarization prompts is provided in Appendix D.

### B.3   Hardware and Environment Configuration

All experiments were conducted on a single server node equipped with 8 NVIDIA L40S GPUs, each with 48GB of HBM2e memory. Our implementation uses PyTorch 2.2, vLLM 0.10.0, and Transformers 4.55.0. The operating system is Red Hat Enterprise Linux 9.6, and the environment is managed via Conda with Python 3.11.7 and CUDA 12.9.

### B.4   Supplementary Experimental Results

This section contains supplementary results and comparisons added during the rebuttal phase to address specific computational and scalability questions.

#### B.4.1   Computational Cost and Baseline Comparison

To assess efficiency, we compare ME-ICPO against standard prompt-style test-time search baselines—Tree of Thoughts (ToT) (Yao et al., 2023a) and Monte-Carlo Tree Refinement (MCTR) (Zhang et al., 2024a)—and the training-based test-time scaling method TTRL (Zuo et al., 2025). For fairness, ME-ICPO uses a fixed configuration of $N{=}5$ refinement rounds and $k{=}16$ samples per round, averaged over 5 seeds. We report average inference time (seconds per question) on AIME 2024. Because ToT/MCTR employ different search depths and branching factors, their runs are not strictly time-matched; for TTRL, we control the experiment to use a similar GPU hours as ME-ICPO. As shown in Table 3 and Table 4, ME-ICPO achieves top-tier accuracy and Mean@16 at a competitive compute budget, reflecting principled gains from the underlying Policy Optimization mechanism. Shallow-search prompt methods (ToT/MCTR) can be faster when search depth is limited but lag notably in accuracy, while under matched wall-clock budgets ME-ICPO attains higher Acc/Mean@16 than TTRL, demonstrating more efficient scaling; importantly, ICPO provides a mechanistic account of test-time self-refinement as policy optimization rather than a heuristic leaderboard tweak.

#### B.4.2   Frontier Model Scalability

To demonstrate the versatility of our framework, we evaluate ME-ICPO on recent frontier models, including Qwen3-4B-Instruct (Yang et al., 2024b) (a long-CoT specialized model), and the Gemini-2.5 series (Pro and Flash) (Comanici et al., 2025). Tables 5 show that ME-ICPO consistently enhances performance across these diverse architectures, confirming its scalability.

#### B.4.3   Harder Task Generalization

We further evaluate performance on exceptionally difficult, high-school/collegiate level math competition tasks, specifically the Harvard-MIT Mathematics Tournament (HMMT) (Balunović et al., 2025) and the APEX-shortlist dataset (Balunović et al., 2025). Results in Table 6 and Table 7 indicate that ME-ICPO provides robustness even on tasks with low baseline solve rates.

Table 3: Computational Cost and Performance Comparison (Mean@16, Qwen2.5-Math-7B) against Inference and Training Baselines.

| Method | AIME-2024 | AMC | MATH(Avg) | Time(s/question) |
|---|---|---|---|---|
| ToT (self eval) | 4.38 | 16.19 | 12.51 | 708 |
| ToT (Maj vote) | 19.58 | 29.37 | 35.63 | **363** |
| MCTR | 4.60 | 1.20 | 17.20 | 1758 |
| TTRL | 27.20 | 45.18 | 46.83 | 1253 |
| **ME-ICPO (ours)** | **30.42** | **47.06** | **54.71** | 1152 |

Table 4: Computational Performance Comparison (Accuracy, Qwen2.5-Math-7B).

| Method | AIME-2024 | AMC | MATH(Avg) |
|---|---|---|---|
| ToT (self-eval) | 4.40 | 18.10 | 10.74 |
| ToT (Maj-vote) | 19.30 | 29.40 | 35.91 |
| MCTR | 23.30 | 2.40 | 33.82 |
| TTRL | 30.00 | 43.37 | 45.11 |
| **ME-ICPO (ours)** | **30.05** | **47.20** | **47.30** |

Table 5: Results on frontier/long-CoT models on AIME 2024.

| Model | Method | Mean@16 (%) | Acc (%) |
|-------|--------|-------------|---------|
| Qwen3-4B-Instruct | Base | 20.62 | 20.59 |
| Qwen3-4B-Instruct | **ME-ICPO (ours)** | **57.71** | **57.67** |
| Gemini-2.5-Pro | Base | 58.54 | 56.60 |
| Gemini-2.5-Pro | **ME-ICPO (ours)** | **79.17** | **80.00** |
| Gemini-2.5-Flash | Base | 35.21 | 35.42 |
| Gemini-2.5-Flash | **ME-ICPO (ours)** | **76.46** | **76.47** |

Table 6: Results on Harder Benchmarks (HMMT / APEX) using Qwen2.5-Math-7B.

| Method | HMMT Mean@16 (%) | HMMT Acc(%) | APEX Mean@16 (%) | APEX Acc(%) |
|--------|------------------|-------------|------------------|-------------|
| Base | **1.04** | 0.67 | 2.55 | 2.61 |
| **ME-ICPO (ours)** | 0.42 | **1.33** | **4.59** | **4.57** |

Table 7: Results on Harder Benchmarks (HMMT / APEX) using Gemini-2.5-Flash.

| Method | HMMT Mean@16 (%) | HMMT Acc(%) | APEX Mean@16 (%) | APEX Acc(%) |
|--------|------------------|-------------|------------------|-------------|
| Base | 14.79 | 14.76 | 14.68 | 18.33 |
| **ME-ICPO (ours)** | **43.12** | **43.14** | **17.18** | **20.00** |

## C DETAILED COMPLEXITY ANALYSIS

### C.1 THEORETICAL TIME AND VRAM COMPLEXITY DERIVATIONS

We formalize the compute model for a decoder-only Transformer and provide exact asymptotic derivations for ME-ICPO (forward-only, summary-aware history) and TTRL (Zuo et al., 2025) (backprop-based test-time RL). The statements and proofs below match the main text analyses verbatim; we only add brief connective narration.

**Setup and primitive costs.** We analyze a decoder-only transformer with $L$ layers and width $d$ (parameter count $|\theta| \asymp Ld^2$). For a single test instance, the initial prompt length (problem statement + template) is $s_0$. Each sampled chain-of-thought (CoT) has average length $\ell$. Per round we sample $k$ candidates; the total number of rounds is $n$. Let $\beta := \ell + r$ denote the number of tokens appended per $(\mathbf{x}, r)$ pair (the reward stub $r$ is $O(1)$, so $\beta \asymp \ell$). The prompt at the beginning of round $t$ therefore has length $T_{t-1} := s_0 + \beta(t-1)$. We use $\kappa \in [2, 3]$ for the backward/forward FLOPs ratio (one backward costs $\kappa$ times one forward) and $g$ for the number of policy-optimization steps per round in TTRL. Primitive costs (suppressing constants) are:

full-sequence forward at length $T$ : $\quad C_{\text{fwd}}(T) = \Theta\big(L(T^2 d + Td^2)\big),$

autoregressive decoding of $\ell$ tokens from prefix $T$ : $\quad C_{\text{dec}}(T, \ell) = \Theta\big(L\big((T\ell + \ell^2)\, d + \ell d^2\big)\big),$

one training step (teacher-forcing fwd + bwd) at length $T$ : $\quad C_{\text{train}}(T) = (1 + \kappa)\, C_{\text{fwd}}(T).$

For ME-ICPO, the one-step lookahead score is computed *on the just-generated branch* by appending a constant number of tokens, so its cost is an incremental forward

$$C_{\text{score}}^{\text{inc}}(T, \ell) = \Theta\big(L\big((T + \ell)\, d + d^2\big)\big),$$

rather than a full recomputation at length $T + \ell$.

### C.1.1 ME-ICPO

**Theorem C.1** (Time Complexity of ME-ICPO). With one shared prefill at length $T_{t-1}$ per round, $k$ candidate decodes from that prefix, and incremental on-branch scoring for each candidate, the total time over $n$ rounds satisfies

$$T_{\text{ME}} = \Theta\Big(Ld\,\beta^2\, n^3 \;+\; k\,Ld\,\beta^2\, n^2 \;+\; Ld^2\,\beta\, n^2 \;+\; k\,Ld^2\,\beta\, n\Big),$$

and, using $\beta \asymp \ell$,

$$T_{\text{ME}} = \Theta\big(Ld\,\ell^2 n^3 + k\,Ld\,\ell^2 n^2\big) \;+\; \Theta\big(Ld^2(\ell n^2 + k\,\ell n)\big).$$

*Proof.* The total time complexity, $T_{\text{ME}}$, is the sum of the costs for prefilling the context ($S_{\text{fwd}}$), decoding the candidates ($S_{\text{dec}}$), and scoring each candidate ($S_{\text{score}}$) over all $n$ rounds.

$$T_{\text{ME}} = \sum_{t=1}^{n} C_{\text{fwd}}(T_{t-1}) + \sum_{t=1}^{n} k\,C_{\text{dec}}(T_{t-1}, \ell) + \sum_{t=1}^{n} k\,C_{\text{score}}^{\text{inc}}(T_{t-1}, \ell) =: S_{\text{fwd}} + S_{\text{dec}} + S_{\text{score}}.$$

We analyze each component by first summing over the rounds and then identifying the leading-order terms in $n$. The prompt length at round $t$ is $T_{t-1} = s_0 + \beta(t-1)$, and we use the standard sums $\sum_{r=0}^{n-1} r = \frac{n(n-1)}{2} = \Theta(n^2)$ and $\sum_{r=0}^{n-1} r^2 = \frac{(n-1)n(2n-1)}{6} = \Theta(n^3)$.

The prefill cost, $S_{\text{fwd}}$, is given by:

$$S_{\text{fwd}} = \sum_{t=1}^{n} C_{\text{fwd}}(T_{t-1})$$

$$\overset{(a)}{=} \Theta\left(Ld\sum_{t=1}^{n} T_{t-1}^2 + Ld^2 \sum_{t=1}^{n} T_{t-1}\right)$$

$$\overset{(b)}{=} \Theta\left(Ld\left(ns_0^2 + s_0\beta n(n-1) + \beta^2\frac{(n-1)n(2n-1)}{6}\right) + Ld^2\left(ns_0 + \beta\frac{n(n-1)}{2}\right)\right)$$

$$\overset{(c)}{=} \Theta\big(Ld\,\beta^2 n^3\big) + \Theta\big(Ld^2\,\beta\,n^2\big).$$

where (a) substitutes the definition of $C_{\text{fwd}}(T) = \Theta(L(T^2 d + T d^2))$ and pulls constants out of the sum; (b) substitutes the exact formulas for the sum of linear and quadratic sequences; and (c) identifies the highest-order terms in $n$ for each part of the expression.

The decoding cost, $S_{\text{dec}}$, is given by:

$$S_{\text{dec}} = \sum_{t=1}^{n} k\,C_{\text{dec}}(T_{t-1}, \ell)$$

$$\overset{(d)}{=} \Theta\left(kLd\left(\ell\sum_{t=1}^{n} T_{t-1} + n\ell^2\right) + kLd^2(n\ell)\right)$$

$$\overset{(e)}{=} \Theta\left(kLd\left(\ell\left(ns_0 + \beta\frac{n(n-1)}{2}\right) + n\ell^2\right) + kLd^2 n\ell\right)$$

$$\overset{(f)}{=} \Theta\big(kLd\,\beta\ell n^2\big) + \Theta\big(kLd^2\,\ell n\big).$$

where (d) substitutes the definition of $C_{\text{dec}}(T, \ell) = \Theta(L((T\ell + \ell^2)d + \ell d^2))$; (e) substitutes the sum for $T_{t-1}$; and (f) identifies the dominant term in $n$ for the $Ld$ component as the quadratic term $\Theta(kLd\,\beta\ell n^2)$.

The incremental scoring cost, $S_{\text{score}}$, is given by:

$$S_{\text{score}} = \sum_{t=1}^{n} k\,C_{\text{score}}^{\text{inc}}(T_{t-1}, \ell)$$

$$\overset{(g)}{=} \Theta\left(kLd\left(\sum_{t=1}^{n} T_{t-1} + n\ell\right) + kLd^2 n\right)$$

$$\overset{(h)}{=} \Theta\big(kLd\,\beta n^2\big) + \Theta\big(kLd^2 n\big).$$

where (g) substitutes the definition of $C_{\text{score}}^{\text{inc}}(T, \ell) = \Theta(L((T + \ell)d + d^2))$; and (h) identifies the dominant term as $\Theta(kLd\,\beta n^2)$.

Combining the leading-order terms for the three components, we have:

$$T_{\text{ME}} = S_{\text{fwd}} + S_{\text{dec}} + S_{\text{score}}$$
$$= \Theta\big(Ld\,\beta^2 n^3 + Ld^2\,\beta\,n^2\big) + \Theta\big(kLd\,\beta\ell n^2 + kLd^2\,\ell n\big) + \Theta\big(kLd\,\beta n^2 + kLd^2 n\big).$$

Grouping the terms by their dependence on $d$ and $d^2$, and noting that the scoring cost terms are dominated by or are asymptotically equal to the decoding cost terms (since $\ell \geq 1$), the total complexity simplifies to:

$$T_{\mathrm{ME}} = \underbrace{\Theta\big(Ld\,\beta^2 n^3\big)}_{\text{from } S_{\mathrm{fwd}}} + \underbrace{\Theta\big(kLd\,\beta\ell n^2\big)}_{\text{from } S_{\mathrm{dec}}} + \underbrace{\Theta\big(Ld^2\,\beta n^2\big)}_{\text{from } S_{\mathrm{fwd}}} + \underbrace{\Theta\big(kLd^2\,\ell n\big)}_{\text{from } S_{\mathrm{dec}}}.$$

Using the approximation $\beta = \Theta(\ell)$, we arrive at the final expression stated in the theorem. This completes the proof. $\qquad\square$

**Theorem C.2** (VRAM Complexity of ME-ICPO). *If candidates are decoded sequentially (no $k$-way parallelism), the peak memory over $n$ rounds satisfies*

$$M_{\mathrm{ME}} = \Theta(|\theta|) \;+\; \Theta\big(Ld\,(s_0 + \beta n)\big) = \Theta(|\theta|) \;+\; \Theta\big(Ld\,(s_0 + \ell n)\big).$$

*If $b \leq k$ candidates are decoded concurrently, the second term is multiplied by $b$.*

*Proof.* The peak VRAM complexity is the sum of the static memory for model parameters and the maximum dynamic memory for the attention KV cache. As ME-ICPO is a forward-only method, it requires no memory for gradients or optimizer states. The model weights occupy $\Theta(|\theta|)$ space. For a decoder-only transformer with $L$ layers and width $d$, the KV cache for a sequence of length $T$ requires $M_{\mathrm{KV}}(T) = \Theta(LdT)$ memory.

The context length at the beginning of round $t$ is $T_{t-1} = s_0 + \beta(t-1)$. During the decoding of a candidate of length $\ell$, the sequence grows to a maximum of $T_{t-1} + \ell$. Since this length is monotonically increasing with $t$, the global peak occurs during the final round ($t = n$), giving a maximum sequence length of $T_{n-1} + \ell = s_0 + \beta(n-1) + \ell = \Theta(s_0 + \beta n)$.

Therefore, the total peak memory for sequential decoding is the sum of the static and maximum dynamic components:

$$M_{\mathrm{ME}} = \Theta(|\theta|) + M_{\mathrm{KV}}\big(\Theta(s_0 + \beta n)\big) = \Theta(|\theta|) + \Theta\big(Ld(s_0 + \beta n)\big).$$

Using the approximation $\beta = \Theta(\ell)$ yields the equivalent form. If $b \leq k$ candidates are processed concurrently, each parallel branch maintains its own KV cache, so the activation memory term scales linearly with $b$. $\qquad\square$

### C.1.2 TTRL

**Theorem C.3** (Time Complexity of TTRL). *In each round, TTRL (i) samples $k$ candidates from prefix $s_0$, and (ii) performs $g$ policy-optimization steps using teacher-forcing on sequences of length $s_0 + \ell$. The total time over $n$ rounds is*

$$T_{\mathrm{TTRL}} = \Theta\Big(n\,g\,k\,(1 + \kappa)\,L\big((s_0 + \ell)^2 d + (s_0 + \ell)d^2\big)\Big)$$

*and the per-round prefill and sampling costs, $\Theta\big(L(s_0^2 d + s_0 d^2)\big)$ and $\Theta\big(kL((s_0\ell + \ell^2)d + \ell d^2)\big)$, are lower-order whenever $g \geq 1$.*

*Proof.* The total time complexity, $T_{\mathrm{TTRL}}$, is the sum of costs over $n$ rounds. In each round, the process performs one shared prefill of the prefix $s_0$, followed by $k$ candidate sampling operations, and finally $g$ training steps for each of the $k$ candidates. The cost for a single round is thus $C_{\mathrm{fwd}}(s_0) + k\,C_{\mathrm{dec}}(s_0, \ell) + gk\,C_{\mathrm{train}}(s_0 + \ell)$. The total cost over $n$ rounds is:

$$T_{\mathrm{TTRL}} = n \cdot \Big(C_{\mathrm{fwd}}(s_0) + k\,C_{\mathrm{dec}}(s_0, \ell) + gk\,C_{\mathrm{train}}(s_0 + \ell)\Big).$$

We substitute the standard complexity formulas for a decoder-only Transformer, using $C_{\mathrm{train}}(T) = (1 + \kappa)C_{\mathrm{fwd}}(T)$, where $\kappa$ is the backward/forward FLOPs ratio. This yields three terms:

$$T_{\mathrm{TTRL}} = \underbrace{\Theta\Big(nL(s_0^2 d + s_0 d^2)\Big)}_{\text{Prefill Cost}} + \underbrace{\Theta\Big(nkL((s_0\ell + \ell^2)d + \ell d^2)\Big)}_{\text{Sampling Cost}}$$

$$+ \underbrace{\Theta\Big(ngk(1 + \kappa)\,L\big((s_0 + \ell)^2 d + (s_0 + \ell)d^2\big)\Big)}_{\text{Training Cost}}.$$

To determine the tight asymptotic bound, we identify the dominant term. The training cost term scales with the number of optimization steps $g$ and the backpropagation factor $(1 + \kappa)$. Furthermore, its self-attention cost is quadratic in the longer sequence length, $s_0 + \ell$. In contrast, the sampling cost's attention component is only linear in the prefix length, $\Theta(s_0\ell)$, and the prefill cost is computed on the shorter prefix $s_0$ and lacks the multiplicative factors of $g$ and $k$.

Consequently, for any $g \geq 1$, the training cost term is of a higher order than both the prefill and sampling costs. Therefore, the lower-order terms are absorbed into the $\Theta$-notation, giving the final tight bound:

$$T_{\text{TTRL}} = \Theta\Big(n\, g\, k\, (1 + \kappa)\, L\big((s_0 + \ell)^2 d + (s_0 + \ell)d^2\big)\Big). \qquad \square$$

**Theorem C.4** (Vram Complexity of TTRL). During training at sequence length $T = s_0 + \ell$, a TTRL step must hold (at least) model weights, gradients, optimizer states (e.g., SGD momentum or Adam's first/second moments), and backward activations. Consequently, for effective batch size $\text{batch}$, the peak memory obeys

$$M_{\text{TTRL}} = \underbrace{\Theta(|\theta|)}_{\text{weights}} + \underbrace{\Theta(|\theta|)}_{\text{gradients}} + \underbrace{\Theta(|\theta|) - \Theta(2|\theta|)}_{\text{optimizer states}} + \underbrace{\Theta(L\, d\, T \cdot \text{batch})}_{\text{backward activations}}$$

$$= \Theta(|\theta|) \; + \; \Theta\big(L\, d\,(s_0 + \ell) \cdot \text{batch}\big).$$

*Proof.* The peak VRAM complexity of a TTRL training step is the sum of two primary components: parameter-resident memory and activation-resident memory. The parameter-resident portion consists of the model weights, their corresponding gradients, and the optimizer states (e.g., first and second moments for Adam). Since each of these scales linearly with the number of parameters, $|\theta|$, their combined memory requirement is compactly expressed as $M_{\text{param}} = \Theta(|\theta|)$.

The activation-resident memory is required for the backward pass. For a training sequence of length $T = s_0 + \ell$ and a decoder-only Transformer with $L$ layers and width $d$, a standard (non-checkpointed) backpropagation pass must cache activations (such as hidden states and MLP intermediates) of size $\Theta(d)$ for every token in the sequence. Aggregating this over $L$ layers and an effective batch size of $\text{batch}$, the activation memory scales as $M_{\text{act}} = \Theta(L \cdot d \cdot T \cdot \text{batch})$.

The total peak memory is the sum of these two components, $M_{\text{TTRL}} = M_{\text{param}} + M_{\text{act}}$. Substituting the derived complexities yields the final expression stated in the theorem:

$$M_{\text{TTRL}} = \Theta(|\theta|) + \Theta\big(L\, d\,(s_0 + \ell) \cdot \text{batch}\big). \qquad \square$$

### C.1.3 COMPARISON WITH TTRL (ZUO ET AL., 2025)

**Proposition C.5** (Time Complexity Threshold). ME-ICPO is computationally faster than TTRL when the number of in-context optimization rounds, $n$, is below a threshold $n^\star$. This threshold is given by:

$$n^\star = \begin{cases} \frac{s_0 + \ell}{\ell} \sqrt{gk(1 + \kappa)} & \text{if } k \leq n \\ g(1 + \kappa) \frac{(s_0 + \ell)^2}{\ell^2} & \text{if } k > n \end{cases}$$

In practical settings with a small number of rounds (e.g., $n \leq 10$), this condition is typically met, making ME-ICPO the more time-efficient approach.

*Proof.* The threshold $n^\star$ is found by equating the leading-order terms of the time complexities derived in Theorem C.1 and Theorem C.3. We consider two regimes based on the dominant term in the complexity of ME-ICPO.

Case 1 ($k \leq n$): The dominant term for ME-ICPO is the prefill cost, $T_{\text{ME}} \asymp Ld\,\ell^2 n^3$. The TTRL cost is $T_{\text{TTRL}} \asymp ngk(1 + \kappa)\, L(s_0 + \ell)^2 d$. Equating them and solving for $n$ yields:

$$Ld\,\ell^2 n^3 = ngk(1 + \kappa)\, L(s_0 + \ell)^2 d \quad \implies \quad n^2 = \frac{gk(1 + \kappa)(s_0 + \ell)^2}{\ell^2},$$

$$n^\star = \frac{s_0 + \ell}{\ell}\sqrt{gk(1 + \kappa)}.$$

Case 2 ($k > n$): The dominant term for ME-ICPO is the decoding cost, $T_{\text{ME}} \asymp kLd\,\ell^2 n^2$. Equating this with the TTRL cost gives:

$$kLd\,\ell^2 n^2 = ngk(1 + \kappa)\, L(s_0 + \ell)^2 d \quad \implies \quad \ell^2 n = g(1 + \kappa)(s_0 + \ell)^2,$$

$$n^\star = g(1+\kappa)\frac{(s_0+\ell)^2}{\ell^2}.$$

In both cases, for $n < n^\star$, the complexity of ME-ICPO is lower. □

**Proposition C.6** (Memory Complexity Comparison). Let $b_{\mathrm{ME}}$ be the number of concurrently decoded candidates in ME-ICPO, and $b$ be the TTRL training batch size. ME-ICPO achieves a lower peak VRAM than TTRL if the number of rounds $n$ is below a threshold. A sufficient condition is:

$$n \;<\; \frac{b(s_0+\ell) - b_{\mathrm{ME}}\,s_0}{b_{\mathrm{ME}}\,\ell} \;+\; \frac{\Delta_{param}\,|\theta|}{b_{\mathrm{ME}}\,c_{\mathrm{kv}}\,L\,d\,\ell},$$

where $\Delta_{param} > 0$ represents the additional parameter-resident memory (gradients, optimizer states) required by TTRL, and $c_{\mathrm{kv}}$ is a constant. Given that TTRL requires strictly more parameter-side memory and typically uses a larger batch size $b$, this condition holds for all practical values of $n$.

*Proof.* We establish the condition by comparing the upper bound on ME-ICPO memory from Theorem C.2 with the lower bound on TTRL memory from Theorem C.4. We seek the condition on $n$ for which $M_{\mathrm{ME}} < M_{\mathrm{TTRL}}$:

$$\underbrace{c_{\mathrm{w}}^{\mathrm{ME}}\,|\theta| + c_{\mathrm{kv}}\,b_{\mathrm{ME}}\,L\,d\,(s_0+\ell n)}_{M_{\mathrm{ME}}} \;<\; \underbrace{(c_{\mathrm{w}}^{\mathrm{T}} + c_{\mathrm{g}}^{\mathrm{T}} + c_{\mathrm{opt}}^{\mathrm{T}})\,|\theta| + c_{\mathrm{kv}}\,b\,L\,d\,(s_0+\ell)}_{M_{\mathrm{TTRL}}}.$$

Let $\Delta_{param} := (c_{\mathrm{w}}^{\mathrm{T}} + c_{\mathrm{g}}^{\mathrm{T}} + c_{\mathrm{opt}}^{\mathrm{T}} - c_{\mathrm{w}}^{\mathrm{ME}}) > 0$ be the constant factor for the additional parameter-sized tensors (gradients, optimizer states) that TTRL requires. Rearranging the inequality to solve for $n$:

$$c_{\mathrm{kv}}\,b_{\mathrm{ME}}\,L\,d\,\ell n < c_{\mathrm{kv}}\,b\,L\,d\,(s_0+\ell) - c_{\mathrm{kv}}\,b_{\mathrm{ME}}\,L\,d\,s_0 + \Delta_{param}\,|\theta|$$

$$n < \frac{b(s_0+\ell) - b_{\mathrm{ME}}s_0}{b_{\mathrm{ME}}\ell} + \frac{\Delta_{param}\,|\theta|}{c_{\mathrm{kv}}\,b_{\mathrm{ME}}\,L\,d\,\ell}.$$

Since $\Delta_{param} > 0$, the second term is strictly positive, providing a further margin. For typical use-cases like sequential decoding ($b_{\mathrm{ME}} = 1$) and $b \geq 1$, the first term is large and positive, making the condition true for any practical number of rounds $n$. □

## C.2 EMPIRICAL COST

We complement our derivations with a controlled latency and memory study on **AIME 2024** using **Qwen2.5-Math-7B**. We vary the number of rounds $n$ and the number of candidates per round $k$ for ME-ICPO to examine its computational characteristics and to validate the asymptotic trends predicted by Theorem C.1. We sweep $n \in \{1,3,5,7\}$ and $k \in \{4,8,16,32\}$. Unless otherwise noted, ME-ICPO uses our main protocol: temperature 0.6, top-$p = 0.95$, summary cap of 500 tokens, and entropy lookahead with $m{=}16$ short samples. All runs are performed on $2{\times}$L40S (48GB) GPUs using bf16 precision, with candidates decoded sequentially ($b_{\mathrm{ME}}{=}1$).

**metrics.** (1) **wall time** per question (s), averaged over the test set; (2) **token usage** per question (generated tokens, in thousands); (3) **peak vram** (GB) from `vllm` memory statistics.

**results.** Table 8 reports wall time, token usage, and peak VRAM across the $(n,k)$ grid. Two trends emerge. (i) For fixed $k$, wall time increases superlinearly with $n$ and is well described by a prefill-dominated scaling close to $O(n^3)$, consistent with Theorem C.1. (ii) For large $k$, the cost transitions toward $O(k\,n^2)$ as candidate evaluation/selection becomes the bottleneck. Token usage grows with both $n$ and $k$ (longer multi-round traces and more candidates), and peak VRAM remains stable in the 81–82.3 GB range across settings, indicating that memory is primarily governed by the base model residency and KV-cache size under sequential decoding.

## D PROMPT TEMPLATES AND QUALITATIVE EXAMPLES

### D.1 DATASET-SPECIFIC SYSTEM PROMPTS

These system prompts define the semantics of the reward tag and are placed once at the beginning of the context.

Table 8: Empirical cost vs. $(n, k)$ on **AIME 2024** with **Qwen2.5-Math-7B**.

| $n$ | $k$ | Wall Time (s/question) | Tokens (k/question) | Peak VRAM (GB) |
|---|---|---|---|---|
| 1 | 4 | 197.52 | 99.59 | 81.50 |
| 1 | 8 | 228.89 | 112.25 | 81.43 |
| 1 | 16 | 286.74 | 379.82 | 82.24 |
| 1 | 32 | 468.36 | 852.47 | 82.27 |
| 3 | 4 | 406.65 | 247.76 | 81.43 |
| 3 | 8 | 469.79 | 272.56 | 81.43 |
| 3 | 16 | 715.85 | 1070.98 | 82.18 |
| 3 | 32 | 1163.51 | 2111.85 | 82.30 |
| 5 | 4 | 609.75 | 372.18 | 81.42 |
| 5 | 8 | 713.25 | 506.68 | 81.43 |
| 5 | 16 | 1152.67 | 1613.49 | 82.24 |
| 5 | 32 | 1726.70 | 3412.94 | 82.30 |
| 7 | 4 | 820.60 | 559.20 | 81.43 |
| 7 | 8 | 1156.62 | 566.82 | 82.30 |
| 7 | 16 | 1371.78 | 1753.71 | 82.27 |
| 7 | 32 | 2643.74 | 5123.67 | 82.29 |

---

**System Prompt: AIME / AMC (Numeric Answer)**

You are an AI mathematician. All content you output MUST be in English.
Below are compressed solution ideas from previous attempts; each idea is tagged with reward 1 (correct) or reward 0 (incorrect). Use the question and these ideas to deduce the correct numeric answer.
**Finish all your reasoning, then on a NEW line output exactly one number (the answer) and nothing else.**
Your final output MUST be in the format boxed{<number>}, where <number> is the final numeric answer only (no expressions, variables, or additional text). The content inside boxed{<number>} must be a decimal number, not a fraction or any other form.

---

**System Prompt: GPQA (Multiple Choice)**

You are an AI mathematician. All content you output MUST be in English.
Below are compressed solution ideas from previous attempts; each idea is tagged with reward 1 (correct) or reward 0 (incorrect). Use the question and these ideas to deduce the correct choice.
**Finish all your reasoning, then on a NEW line output exactly one letter (the answer) and nothing else.**
Your final output MUST be in the format boxed{<letter>}, where <letter> is exactly one of A, B, C, D.

---

**System Prompt: MATH (Free-form Answer)**

You are an AI mathematician. All content you output MUST be in English.
Below are compressed solution ideas from previous attempts; each idea is tagged with reward 1 (correct) or reward 0 (incorrect). Use the question and these ideas to deduce the correct answer.
**Finish all your reasoning, then on a NEW line output exactly one answer (the answer) and nothing else.**
Your final output MUST be in the format boxed{<answer>}.

---

It gives an explicit, task-level meaning to the reward tags ($r \in \{0, 1\}$), telling the model to learn from high-reward ideas and discount low-reward ones—so the model can *use* feedback without any gradient updates in test time.

### D.2  SUMMARIZATION PROMPTS

These prompts compress a full CoT into a short summary that retains the high-level strategy.

---

**Summarization Prompt: AIME / AMC**

Provide a concise summary of the reasoning in the answer below. Do NOT add any introductory phrases or extra explanations. Omit all numerical calculations. The summary must be self-contained, no more than 100 tokens.

---

---

If there is a final numeric result, include it at the end in the format `boxed{<number>}` (decimal only, no fractions, variables, or extra text). If there is no numeric answer, do not output `boxed{}`.
[Answer start] {... raw model answer ...} [Answer end]
Summary:

---

**Summarization Prompt: GPQA**

Provide a concise summary of the reasoning in the answer below. Do NOT add any introductory phrases or extra explanations. Omit all numerical calculations. The summary must be self-contained, no more than 100 tokens.
If there is a final answer choice, include it at the end in the format `boxed{<letter>}` (must be exactly one of A, B, C, D, no extra text). If there is no final answer, do not output `boxed{}`.
[Answer start] {... raw model answer ...} [Answer end]
Summary:

---

**Summarization Prompt: MATH**

Provide a concise summary of the reasoning in the answer below. Do NOT add any introductory phrases or extra explanations. Omit all calculations unless essential to understanding. The summary must be self-contained, no more than 100 tokens.
If there is a final answer, include it at the end in the format `boxed{<answer>}`.
[Answer start] {... raw model answer ...} [Answer end]
Summary:

---

They replace long CoTs with short strategy summaries, keeping only decision-relevant logic while dropping arithmetic details, so the history fits in-context and remains useful across rounds.

## D.3 QUALITATIVE CASE STUDIES

---

**Total Prompt**

You are an AI mathematician. All content you output MUST be in English.
Below are compressed solution ideas from previous attempts; each idea is tagged with reward 1 (correct) or reward 0 (incorrect). Use the question and these ideas to deduce the correct numeric answer.
**Finish all your reasoning, then on a NEW line output exactly one number (the answer) and nothing else.**
Your final output MUST be in the format `boxed{<number>}`, where $<number>$ is the final numeric answer only (no expressions, variables, or additional text). The content inside `boxed{<number>}` must be a decimal number, not a fraction or any other form.
Every morning Aya goes for a 9-kilometer-long walk and stops at a coffee shop afterwards. When she walks at a constant speed of $s$ kilometers per hour, the walk takes her 4 hours, including $t$ minutes spent in the coffee shop. When she walks $s + 2$ kilometers per hour, the walk takes her 2 hours and 24 minutes, including $t$ minutes spent in the coffee shop. Suppose Aya walks at $s + \frac{1}{2}$ kilometers per hour. Find the number of minutes the walk takes her, including the $t$ minutes spent in the coffee shop.
bad ideas (reward 0):
[0]- When Aya walks at $s + \frac{1}{2}$ kilometers per hour, the total time taken, including the $t$ minutes spent in the coffee shop, is approximately 348 minutes.
Thus, the answer is 348.0
good ideas (reward 1):
[0]- To determine how long it would take Aya to walk from her house to the park and back, including the time spent in the coffee shop, we set up two equations based on the given information. By solving these equations, we found the speed $s$ and the time $t$ spent in the coffee shop. Then, we calculated the time required when Aya walks at $s + \frac{1}{2}$ km/h, which turned out to be 204 minutes.
Thus, the answer is 204.0
[1]- By analyzing the given conditions and using algebraic manipulation, we deduced that the value of $x$ is $\frac{3}{5}$, which when multiplied by 340 gives us the final answer of 204.
Thus, the answer is 204.0
[2]- We solved the system of equations to find the walking speed $s$ and the time $t$ spent in the coffee shop. Then, we calculated the total time required when Aya walks at a speed of $s + \frac{1}{2}$ km/h, including the time spent in the coffee shop. The final answer is 204.0 minutes.
Thus, the answer is 204.0

**Blue** marks the *system instructions* that enforce output schema and reward semantics. **Orange** marks the *problem statement*. **Red** marks *reward-0 (incorrect) ideas* retained as counterexamples. **Green** marks *reward-1 (correct) ideas* that ICPO prioritizes during selection.

**How ICPO appears in this example.** ICPO samples multiple CoTs, summarizes them, and assigns rewards based on self-consistency. The green items represent low-entropy agreement on the solution path that eliminates the fixed coffee time $t$ and yields the evaluation at $s + \frac{1}{2}$ as $\boxed{204.0}$ minutes. Entropy-based filtering downweights the red outlier ($\approx 348$ minutes) that improperly scales total time and ignores the fixed offset. The final evaluator, conditioned on the schema and the curated ideas, outputs the correct numeric answer, $\boxed{204.0}$.

## USE OF LARGE LANGUAGE MODELS

We used LLMs solely as a writing assistant for minor grammar and phrasing corrections during manuscript preparation. LLMs were not involved in research ideation, experiment design, data analysis, or result interpretation.

