# OpenReview forum: "Provable and Practical In-Context Policy Optimization for Self-Improvement"
_ICLR.cc/2026/Conference — ICLR 2026 Poster_

### Official Review · Reviewer_jkwj · 2025-10-26

**Soundness:** 3
**Presentation:** 4
**Contribution:** 4
**Rating:** 6
**Confidence:** 5

**Summary:**

This paper introduces the In-Context Policy Optimization (ICPO) framework for understanding the phenomenon of test-time scaling in LLMs, i.e., when models improve their responses through multi-round self-reflection without parameter updates. The main idea is to model this process as an agent that optimizes its response (action $x$), based on a history of in-context action-reward pairs $(x_t, r_t)$, effectively performing policy optimization within the context window. The main theoretical contribution proves that a single-layer Linear Self-Attention (LSA) transformer can imitate a specific policy optimization algorithm: a variant of FTRL for linear bandits. This imitation is achieved when the LSA model is pre-trained using a Fisher-weighted logit-matching objective. The authors present this as a foundational proof of how an attention-based architecture can learn to perform in-context optimization. Building on this theoretical result, the authors propose a practical, gradient-free inference-time algorithm called Minimum-Entropy ICPO (ME-ICPO), which they leverage for complex mathematical reasoning tasks. At each round, ME-ICPO generates multiple candidate solutions, assigns self-assessed rewards using majority voting on the final answers, summarizes the reasoning paths (Chain-of-Thought) to manage context length, and selects the next reasoning step to add to the context based on a minimum-entropy criterion. The experiments show that ME-ICPO achieves competitive, and in some cases state-of-the-art, performance on benchmarks like AIME, AMC, and MATH.

**Strengths:**

1. To best of my knowledge, the paper introduces a first framework to formally model in-context self-improvement as a policy optimization problem. The authors provide a new and principled perspective on the mechanisms underlying test-time scaling.

2. The main theorems establish population-level equivalence to an FTRL-like algorithm and provide finite-sample guarantees for learning this algorithm from data. They also analyze the stability of the learned policy to reward perturbations.

3. The proposed ME-ICPO algorithm is an effective method. It demonstrates substantial and consistent performance improvements over specialized base models on multiple mathematical reasoning benchmarks.

4. ME-ICPO is a gradient-free, inference-time-only algorithm. This makes it significantly more computationally efficient (particularly in terms of VRAM) than methods that require test-time backpropagation, such as TTRL. This practicality makes it a more accessible method for improving LLM performance.

**Weaknesses:**

Please respond to weaknesses, I will consider raising my score from 6 to 8 if all weaknesses are addressed -- this is a good paper!

1. The most significant weakness is the large abstraction gap between the theoretical model and the practical application. The theory is built on a single-layer Linear Self-Attention model solving a linear bandit problem, whereas the experiments are run on deep, multi-layer, non-linear transformers performing complex, structured reasoning. The paper's claim to "explain" the mechanism of self-reflection is an overstatement. The theory provides an elegant proof-of-concept that the attention mechanism can implement a form of optimization, but it does not and cannot prove that this specific linear mechanism is what underlies the sophisticated self-correction abilities observed in models like Qwen2.5-Math. The paper would be stronger if it framed the theory more cautiously as an inspirational, minimal model that demonstrates a core computational capability, rather than a direct explanation of an emergent phenomenon.

2. The ablation study shows that the minimum-entropy selection criterion is the most critical component of ME-ICPO. However, the paper's justification for this heuristic is purely intuitive, suggesting it avoids "corrupted" responses and encourages "diversified" ones. This justification is somewhat vague and potentially self-contradictory (low entropy implies low diversity). The success of this heuristic may be domain-specific. For mathematical problems with a single correct reasoning path, low entropy (high agreement among future sampled paths) is likely a strong proxy for correctness. However, for more open-ended or creative tasks, the optimal path might be one that leads to a rich and diverse set of possibilities (high entropy). The paper lacks a more formal justification for this algorithmic choice and does not compare it against more standard selection criteria from RL, such as simply selecting the candidate with the highest self-assessed reward.

3. Modeling a multi-round reasoning process as a sequence of K-armed bandit pulls is a major simplification. This abstraction ignores the stateful and compositional nature of logical deduction. Each step in a mathematical proof is not an independent choice from a fixed set of K options; rather, it generates a new logical state that constrains all subsequent steps. A more faithful, albeit likely intractable, model would involve a contextual bandit or a full Markov Decision Process (MDP). The paper should explicitly acknowledge and discuss the limitations of this memoryless abstraction and how it impacts the interpretation of the theoretical results.

4. (minor) There is clear over-abuse of spacing in the paper in terms of vspaces. While I realize all authors use this, the authors should not abuse it. Please remove these if your paper is accepted.

Typos and grammatical errors:
- ...the model's ability to digest the in-context information to improve their response. $\rightarrow$ ...to improve its response.
- Such an in-context information can be... $\rightarrow$ Such in-context information can be...
- ...without answering why these ability emerge... $\rightarrow$ ...without answering why this ability emerges... (or ...why these abilities emerge...)
- ...learn to optimize it's behavior x by optimizing it's policy... $\rightarrow$ ...learn to optimize its behavior x by optimizing its policy...
- ...how LLM leverage the in-context actions... $\rightarrow$ ...how LLMs leverage the in-context actions...
- ...to improve it's response \$x\_\{t+1\}\$... $\rightarrow$ ...to improve its response \$x\_\{t+1\}\$...
- ...generating it's response \$x\_t\$ and receives... and then improve it's response... $\rightarrow$ ...generating its response \$x\_t\$ and receives... and then improves its response...
- ...into it's policy optimization process and to gradually improves its response. $\rightarrow$ ...into its policy optimization process and to gradually improve its response.
-  ...where the agent generates and improve it's response... $\rightarrow$ ...where the agent generates and improves its response...
- ...denotes its norm \$l\_2\$ For a matrix A. $\rightarrow$ ...denotes its norm \$l\_2\$. For a matrix A.
-...during the test-time can improve... $\rightarrow$ ...during test-time can improve...
- ...including the Monte-Carol Tree Search... $\rightarrow$ ...including the Monte-Carlo Tree Search...
- ...where the LLM evaluate their own response... $\rightarrow$ ...where the LLM evaluates its own response... (or ...LLMs evaluate their own...)
-  ...by directly assume the LLM's ability... $\rightarrow$ ...by directly assuming the LLM's ability...
-  ...trained linear self attention can implement... $\rightarrow$ ...trained linear self-attention can implement...
- ...multi head constructions... $\rightarrow$ ...multi-head constructions...
-  ...in which first layer heads preprocess... $\rightarrow$ ...in which first-layer heads preprocess...
-  ...rare recent literature have covered... $\rightarrow$ ...rare recent literature has covered...
- ...optimize it's policy \$x\_t\$... $\rightarrow$ ...optimize its policy \$x\_t\$...
- ...dataset is generating from the policy... $\rightarrow$ ...dataset is generated from the policy...
- ...similar with the Follow-the-Regularized Leader... $\rightarrow$ ...similar to the Follow-the-Regularized Leader...
- ...defined by \$s\propto log~p\$ In the following... $\rightarrow$ ...defined by \$s\propto log~p\$. In the following...
-  ...prefix of trajectory up to... $\rightarrow$ ...prefix of trajectory \$\tau\$ up to...
-...exploration parametery is wide... $\rightarrow$ ...exploration parameter \$\gamma\$ is wide...
-...and p is a normalization factor... $\rightarrow$ ...and \$\rho\$ is a normalization factor...
-  The LSA model parameterized by starts with... $\rightarrow$ The LSA model parameterized by \$\theta\$ starts with...
- ...the LSA model updates it's policy... $\rightarrow$ ...the LSA model updates its policy...
-  ...corresponding K dimension... $\rightarrow$ ...corresponding \$K\$ dimensions...
- The expected matrix I is inspired by... $\rightarrow$ The expected matrix \$\Gamma\$ is inspired by...
- The Fisher-weighted loss provide new loss... $\rightarrow$ The Fisher-weighted loss provides a new loss...
- ...that common KL loss between... $\rightarrow$ ...that the common KL loss between...
- ...using the KL loss enable the transformers... $\rightarrow$ ...using the KL loss enables the transformers...

**Questions:**

See above.

---

> ### Author Response · Authors · 2025-11-21
> **Response to Reviewer jkwj**
>
> We sincerely thank the reviewer for the encouraging assessment and the constructive comments that have helped us improve the paper revision.
>
> **Q1.** Why the theoretical analysis is built on a linear model
>
> **A1.** We would like to emphasis that the adoptation of a simplified linear model is to deliver the framework for in-context policy optimization to understand the self-refinement process in LLMs. Therefore, we follow the LSA structure as a common analytic proxy in recent ICL theory [4]. We appreciate your suggestions that the theoretical foundation in Section 4 is a 'inspiration model' to understand the emergent of self-reflection. We have reflected this into the introduction of Section 4 (marked in red)
>
> ---
> **Q2.**. Why we need the  Minimum-Entropy (ME) Heuristic
>
> **A2.** We clarify that ME is **not** computed over the final-answer vote; it is the predictive entropy of the next-step policy after roll-in, i.e., $H[\pi(\cdot \mid \mathcal{H}\oplus c)]$ where $c$ is a candidate inserted into the context. *Intuitively*, a in-context candidate that diversifies the history will give the agent more information thus improves the certainty of next response (with lower entropy). *Theoretically*, this aligned with the analysis in linear bandits in [2]: if a representation yields lower uncertainty from the observated history, this representation should be selected for an improved performance (lower regret).
>
> ---
>
>
>
>
> **Q3.** Why choose bandit simplification instead of the MDP/Stateful Nature
>
> **A3.** We agree that viewing the reasoning process, especially the multi-step reasoning process as a MDP / stateful structure would be promising. However, we also would like to note that due to the next state for next-token prediction framework is determinstic since $s' = s \oplus a$, current literature still be more focused on the bandit structure viewing the whole response as an action. Given the fruitful results of policy optimization in RL community [1, 2, 3], we believe extending the ICPO framework to MDP setting is a feasible future work.
>
> In addition, we have carefully proofread the paper and fixed all the typos we could find and formatting issues listed in your review. Thank you again for helping us polish this work!
>
> ---
> **References**
>
> [1] Cai, Qi, et al. "Provably efficient exploration in policy optimization." International Conference on Machine Learning. PMLR, 2020.
>
> [2] Zhong, Han, and Tong Zhang. "A theoretical analysis of optimistic proximal policy optimization in linear markov decision processes." Advances in Neural Information Processing Systems 36 (2023): 73666-73690.
>
> [3] Shani, Lior, et al. "Optimistic policy optimization with bandit feedback." International Conference on Machine Learning. PMLR, 2020.
>
> [4] Zhang, R., Frei, S., & Bartlett, P. L. (2024). Trained transformers learn linear models in-context. Journal of Machine Learning Research, 25(49), 1-55.

---

> ### Author Response · Authors · 2025-11-24
> **Follow up with Reviewer jkwj**
>
> Thank you for your review and supportive feedback in raising the score. We would like to follow up to see if our previous response has resolved your concerns and if there are any new questions and suggestions.
>
> In our previous response, we have addressed your questions regarding (1) why the theory is developed under a linear model by positioning Section 4 as an “inspiration model” that uses LSA as a minimal sandbox to mechanistically explain emergent self-reflection as in-context policy optimization; (2) the role of the Minimum-Entropy (ME) heuristic, clarifying that ME is defined on the predictive entropy of the next-step policy after rolling in a candidate (rather than on final-answer votes) and connecting this to linear bandit theory where lower uncertainty indicates better representations; and (3) the choice of a bandit formulation instead of a full MDP/stateful model, by explaining current practice that treats each full response as an action under deterministic next-token dynamics, and by highlighting that extending ICPO to MDP-style multi-step reasoning is a natural and promising direction for future work (which we now explicitly note in the conclusion), along with fixing the typos and formatting issues you pointed out.

---

### Official Review · Reviewer_4bex · 2025-10-27

**Soundness:** 3
**Presentation:** 2
**Contribution:** 3
**Rating:** 4
**Confidence:** 2

**Summary:**

- The paper introduces ICPO, framing multi‑round self‑reflection at inference as **in‑context policy optimization** that uses self‑assessed or external rewards without parameter updates. The authors prove that, under **Fisher‑weighted logit‑matching** pretraining, a **single‑layer linear self‑attention (LSA)** model can imitate a policy‑optimization algorithm for linear bandits. Building on this, they propose **Minimum‑Entropy ICPO (ME‑ICPO)**, a practical test‑time algorithm that iteratively samples candidates, assigns self‑assessed rewards, and selects low‑entropy, high‑confidence responses; **majority voting** is used to robustify the reward signal.
- Experiments use **Qwen2.5‑Math‑7B** and **Qwen2.5‑Math‑1.5B** across **AIME‑2024, AMC, and MATH L1–L5**, etc., demonstrating the power of ME-ICPO.

**Strengths:**

1) **Clear mechanistic link:** a theoretically grounded account connecting pretraining under a Fisher‑weighted objective to in‑context policy‑optimization behavior in an LSA.
2) **Practicality:** ME‑ICPO yields strong math‑reasoning gains with gradient‑free test‑time optimization; **Mean@16 can surpass the base model’s majority‑vote upper bound**, and adding majority vote on ME‑ICPO output brings further gains.

**Weaknesses:**

- **No variability reported in Table 1.** Table 1 reports only point estimates (Accuracy and Mean@16) with no variability across multiple runs; please add mean±std over, e.g., 5 seeds.
- **Theory scope.** Guarantees apply to a **single‑layer LSA** and **linear bandits**; practical models may not be LSA, so the theoretical guarantees do not directly cover the standard non-LSA archetictures.

**Questions:**

1) Although the proofs target single‑layer LSA, **can ME‑ICPO be safely applied to general (non‑LSA, multi‑layer) Transformers in practice**?
2) The paper should clearly articulate the scope and the required setup for ICPO vs ICPO with LSA. In Section 4, it seems ICPO is only defined with LSA, is this correct? If so, is ME-ICPO only defined for LSA models?
3) "that with sufficient pretraining under a novel Fisher-weighted logit-matching objective, a single-layer linear self-attention model can provably
imitate policy-optimization algorithm for linear bandits", does ME-ICPO described in Algorithm 1 **require** such pretraining to work?
It is not clear whether ME-ICPO can be used as a test-time only method OR it has to be bundled with the specific pretraining procedures.

---

> ### Author Response · Authors · 2025-11-21
> **Response to Reviewer 4bex**
>
> Thank you for your feedback. Here we address your questions.
>
> **Q1.** Can you add the variance for the results reported in Table 1
>
> **A1.** Thanks for the suggestion, we have revised Table 1 to now explicitly report the standard deviation.
>
> ---
>
> **Q2.** Why the theoretical analysis is built on simplified linear model?
>
> **A2.** We would like to first highlight that the theoretical work built on the linear model is to provide a clear and inspirational understanding of this self-reflection phenomenon as in-context policy optimization. To the best of our knowledge, we present the first work to theoretically model in-context self-refinement as Policy Optimization. Therefore, we start from the linear model for clearer analysis. Second, the LSA structure is used as a common analytic proxy in recent ICL theory [1]. We would also like to note that extending the analysis in the linear model to non-linear, general function approximations is possible via statistical complexity measurements (e.g., Eluder dimension [2]) and/or modern representation learning techniques [3]. However, the key message delivered in understanding the self-refinement as a in-context policy optimization framework remains unchanged.
>
> ---
>
> **Q3.** Can the author clarity the relationship and connection between ICPO and ICPO-with-LSA setup?
>
> **A3.** We would like to note that ICPO is a general policy-optimization framework that operates in context and does not assume a specific architecture. "ICPO with LSA" in Section 4 is an analysis lens used to obtain verifiable guarantees under minimal assumptions; it does not define the framework. Built on ICPO, ME-ICPO is a test time instantiation for settings like math reasoning where external rewards may be unavailable; it uses self or weak rewards with entropy based selection and remains training free. Both ICPO and ME-ICPO run on standard multi layer Transformers and are not defined only for LSA.
>
> ---
>
> **Q4.** Does ME-ICPO require Fisher-weighted pretraining?
>
> **A4.** No. In all our experiments ME-ICPO runs at test time on ordinarily pretrained LLMs with standard autoregressive pretraining under cross-entropy (reverse KL) loss. We do not need to introduce any special Fisher-weighted pretraining. The Fisher-weighted logit-matching in our theory is a sufficient analytical condition, not a practical prerequisite. Moreover, Theorem 4.1 shows that a mixed-policy KL term is both upper and lower bounded by a Fisher-projected quadratic loss used in our analysis, so standard KL or cross-entropy alignment already provides the needed surrogate. In practice this means ME-ICPO is training-free and plug-and-play on off-the-shelf models; any model trained with general logit-matching suffices.
>
> ---
> **References**
>
> [1] Zhang, R., Frei, S., & Bartlett, P. L. (2024). Trained transformers learn linear models in-context. Journal of Machine Learning Research, 25(49), 1-55.
>
> [2] Russo, D., & Van Roy, B. (2013). Eluder dimension and the sample complexity of optimistic exploration. Advances in Neural Information Processing Systems, 26.
>
> [3] Nishikawa, Naoki, et al. "Nonlinear transformers can perform inference-time feature learning." Forty-second International Conference on Machine Learning.

---

> > ### Comment · Reviewer_4bex · 2025-11-25
> >
> > > Thanks for the suggestion, we have revised Table 1 to now explicitly report the standard deviation.
> >
> > Thanks for the 5-seed results in Table 1, the lift is clear.
> >
> > >We would like to first highlight that the theoretical work built on the linear model is to provide a clear and inspirational understanding of this self-reflection phenomenon as in-context policy optimization. To the best of our knowledge, we present the first work to theoretically model in-context self-refinement as Policy Optimization. Therefore, we start from the linear model for clearer analysis. Second, the LSA structure is used as a common analytic proxy in recent ICL theory [1]. We would also like to note that extending the analysis in the linear model to non-linear, general function approximations is possible via statistical complexity measurements (e.g., Eluder dimension [2]) and/or modern representation learning techniques [3]. However, the key message delivered in understanding the self-refinement as a in-context policy optimization framework remains unchanged.
> >
> > We understand that extending the analysis to the non-LSA family may be beyond the scope of the current paper. Could you also clarify whether the insights derived from LSA hold in practice, and to what extent the corresponding assumptions are actually satisfied in real-world settings?
> >
> > Overall, it is unclear what specific theoretical insights motivate the design of ME-ICPO, or to what extent the current theorems help explain its strong empirical performance.

---

> ### Author Response · Authors · 2025-11-24
> **Follow up with Reviewer 4bex**
>
> Thank you for your review. We would like to follow up to see if our response has resolved your questions.
>
> In our previous response, we have addressed your questions regarding (1) adding variance in Table 1 by explicitly reporting mean ± standard deviation over multiple seeds; (2) the use of a linear model in the theory, clarifying that it is intended as an inspirational, analytically tractable model to understand in-context self-refinement as policy optimization, with LSA serving as a standard proxy in ICL theory and clear paths to nonlinear extensions; (3) the distinction between the general ICPO framework and the specific “ICPO-with-LSA” setup, emphasizing that ICPO itself is architecture-agnostic and that LSA is used purely as an analysis lens; and (4) whether ME-ICPO requires Fisher-weighted pretraining, explaining that our method is fully training-free at test time on normally pretrained Transformers and that Fisher-weighted logit matching appears only as a sufficient analytical condition rather than a practical requirement.
>
> We would be more than happy to have future discussions.

---

> ### Author Response · Authors · 2025-11-25
> **Further response to Reviewer 4bex**
>
> We sincerely appreciate these insightful follow-up questions and we are happy to see our response has resolved your question regarding the variance in experiments. We also appreciate your support that the current study on LSA aligns with the scope of this paper. We are happy to clarify the following two follow-up questions.
>
> **Q1.** Could you clarify whether the insights derived from LSA hold in practice?
>
> **A1.** First and most importantly, our LSA analysis provides an insightful theoretical analysis showing that LLM **can indeed** optimize its output using in-context feedback. This serves as a foundation for the design of the empirical ME-ICPO version and **inspires** us to put the past experiences in-context for LLM's self-improvement. As the experimental success suggests, the aforementioned insight **holds in practice** so that LLM can improve its response over test-time scaling. Second, another important insight derived from LSA is that this **policy optimization** framework is robust to reward error (Theorem 4.8). This insight directly enables ME-ICPO to collect the reward through majority voting without being overwhelmed by the reward error. In conclusion, we believe these two important insights both hold in practice to ensure the success of ME-ICPO.
>
> ---
>
> **Q2.** To what extent are the corresponding assumptions actually satisfied in real-world settings?
>
> **A2.** We would like to highlight that, despite the linear model assumed for analytical ease, the major assumption we make is that the model is pretrained on a dataset that contains policy-improvement or self-reflection patterns, so that the model can acquire a self-reflection ability. We respectfully note that this assumption is also made in many theoretical analyses [1, 2] and empirical studies [3, 4]. We believe this assumption is commonly satisfied, since there are many real-world corpora that contain such *reflection* patterns.
>
> Another assumption concerns the design of the data structure used for in-context policy optimization. In the theory, each step is treated as an action–reward pair $(x_t, r_t)$ that is appended to the context. This is made explicit in our construction of the in-context trajectory
> $$
> E^{(t)} = \\begin{pmatrix}
> x_1 & \\cdots & x_t & q_x\\\\
> r_1 & \\cdots & r_t & q_r
> \\end{pmatrix}
> $$
> (see lines 214–215 in the main text). In ME-ICPO, we mirror this idea directly in Algorithm 1: at each round we update the in-context list via $H_t \leftarrow H_{t-1} \cup (x^t_{j*}, r^t_{j*})$ (see Algorithm 1, line 7). This design is a direct implementation of the theoretical ICPO mechanism: the model is explicitly fed a growing history of $(x_t, r_t)$ pairs so that the in-context policy-optimization behavior identified in the LSA analysis can be activated in practice.
>
> In addition, the policy-optimization perspective and Theorem 4.8 (“Stability to One-step Reward Perturbations”) jointly emphasize that ICPO-style updates are robust to noisy or locally perturbed rewards. This directly motivates using self- or weak supervision plus majority voting as the reward signal in ME-ICPO (Algorithm 1, lines 3–4): although the bandit-style rewards are noisy, they are aggregated from multiple samples, and Theorem 4.8 suggests that occasional mis-evaluations will not break the in-context optimization loop. Therefore, the assumptions about having noisy rewards and aggregating them through majority vote are also naturally satisfied in our real-world setting, and they help explain why ME-ICPO remains stable and effective in practice.
>
> ---
> **In conclusion**: The theoretical insights motivated us to put the past trials into in-context for policy optimization, accountably following the same phenomenon we verified theoretically. The design of the policy optimization framework inspires us to design a mechanism to select diverse responses in ME-ICPO. The theorem on reward perturbation explains the success of ME-ICPO since the policy improvement result is robust to the errors from the reward assignment.
>
> ---
> **References**
>
> [1] Lin, Licong, Yu Bai, and Song Mei. "Transformers as Decision Makers: Provable In-Context Reinforcement Learning via Supervised Pretraining." The Twelfth International Conference on Learning Representations.
>
> [2] Park, Chanwoo, et al. "Do LLM Agents Have Regret? A Case Study in Online Learning and Games." ICML 2024 Workshop on Theoretical Foundations of Foundation Models.
>
> [3] Renze, Matthew, and Erhan Guven. "Self-reflection in llm agents: Effects on problem-solving performance." arXiv preprint arXiv:2405.06682 (2024).
>
> [4] Yuan, Weizhe, et al. "Self-rewarding language models." Forty-first International Conference on Machine Learning. 2024.

---

### Official Review · Reviewer_XA9N · 2025-11-01

**Soundness:** 3
**Presentation:** 2
**Contribution:** 2
**Rating:** 4
**Confidence:** 2

**Summary:**

This paper introduces ICPO (In-Context Policy Optimization), a framework showing that a one-layer linear self-attention model can imitate a policy-optimization algorithm under a Fisher-weighted training objective. Motivated by insights from the theoretical results, the paper also proposes a practical algorithm, ME-ICPO, which performs multi-round generation, self-assessment with majority voting, chain-of-thought summarization, and minimum-entropy response selection to enable reward-aware prompting and principled feedback selection without further training. The empirical results show improvements on mathematical reasoning benchmarks.

**Strengths:**

1. It is interesting to formulate ICPO as a bandit-style policy optimization approach. The theoretical grounding for in-context self-refinement is potentially impactful if the claims hold in more realistic settings.

2. The framework and algorithm diagrams are well-organized, and the writing is mostly easy to follow.

**Weaknesses:**

1. The theoretical framework in Section 4 uses a linear bandit abstraction and a simplified linear self-attention model, whereas ME-ICPO is demonstrated with models like Qwen2.5-Math-7B. It is not clear how these theoretical assumptions connect to the practical model choices.

2. ICPO requires iterative sampling, which implicitly increases inference compute. The paper only compares with the base model; since this is technically a prompting technique, it is unclear how this improvement differs from test-time scaling methods such as Tree-of-Thoughts, ReAct, and Monte-Carlo Tree Refinement, or from lightweight training methods such as GRPO and TTRL.

3. The ME-ICPO also seems limited. Majority voting requires that (1) the model has sufficient capability to solve the task, (2) reasoning verification is cheap and easier than generation, and (3) the majority answer correlates with correctness. These appear to be strong assumptions that many real tasks may not satisfy.

**Questions:**

1. How does the method perform on recent long-CoT models, for example Qwen3-4B-Instruct? And since this is a training-free method, how does it perform even on frontier models, such as GPT-5 or Gemini-2.5-Pro?

2. How does ICPO extend to harder tasks—for example HMMT, APEX-shortlist tasks—or tasks without final-answer executability, or where the final answer is not discrete?

---

> ### Author Response · Authors · 2025-11-21
> **Response to Reviewer XA9N**
>
> Thank you for your constructive feedback, we address your questions here.
>
> **Q1.** Why the theoretical analysis is built on simplified linear model?
>
> **A1.** We would like to first highlight that the theoretical work built on the linear model is to provide a clear and inspirational understanding of this self-reflection phenomenon as in-context policy optimization. To the best of our knowledge, we present the first work to theoretically model in-context self-refinement as Policy Optimization. Therefore, we start from the linear model for clearer analysis. Second, the LSA structure is used as a common analytic proxy in recent ICL theory [1]. We would also like to note that extending the analysis in the linear model to non-linear, general function approximations is possible via statistical complexity measurements (e.g., Eluder dimension [2]) and/or modern representation learning techniques [3]. However, the key message delivered in understanding the self-refinement as a in-context policy optimization framework remains unchanged.
>
> ---
> **Q2.** How is the computational cost compared with ToT, ReAct, MCTR, GRPO, TTRL methods?
>
> **A2.** Per your suggestion, we add comparisons on Qwen2.5-Math-7B across AIME-2024, AMC, MATH. We compare against Tree-of-Thoughts (ToT), Monte-Carlo Tree Refinement (MCTR), and TTRL. We do not include GRPO since it requires additional training data, and we do not include ReAct because it targets interactive tool-use rather than static math QA under our setup. Due to the varying computational architectures of each method (e.g., MCTR's deep search vs. ToT's shallow search), this comparison is not strictly time-matched across all baselines. Instead, we compared ME-ICPO at its optimal configuration against the standard runs of ToT/MCTR. For the training-based method TTRL, we controlled the experiment to utilize a similar GPU hours as ME-ICPO. Metrics are Acc and Mean@16. These results affirm that ME-ICPO is highly competitive, achieving top-tier performance. In summary, prompt-style test-time scaling methods (e.g., ToT/MCTR) can be faster under shallow search but lag substantially behind our approach in accuracy, and compared to the test-time-scaling baseline TTRL, under matched wall-clock budgets ME-ICPO attains higher Acc/Mean@16, demonstrating more efficient scaling. The result is also presented in Appendix B.4.1 in our revision.
>
>
>
>
>
> **Table 1 — Mean@16 (Qwen2.5-Math-7B)**
>
> | Method | AIME-2024 | AMC | MATH(Avg) |Time(s/question)|
> |---|---|---|---|---|
> | ToT（self-eval） |  4.38 |16.19 |12.51 |708|
> | ToT（Maj_vote） | 19.58 | 29.37 | 35.63|**363**|
> | MCTR |   4.60 |1.20 |17.20  |1758
> | TTRL  | 27.20 |45.18  | 46.83 | 1253 |
> | **ME-ICPO (ours)** | **30.42** | **47.06** |  **54.71**| 1152|
>
> **Table 2 — Acc (Qwen2.5-Math-7B)**
>
> | Method | AIME-2024 | AMC | MATH(Avg) |
> |---|---|---|---|
> | ToT（self-eval） | 4.40 | 18.10 | 10.74|
> | ToT（Maj-vote） |   19.30 | 29.40 | 35.91|
> | MCTR | 23.30 | 2.40 | 33.82 |
> | TTRL | 30.00 | 43.37 | 45.11 |
> | **ME-ICPO (ours)** |**30.05**  | **47.20** |**47.30**  |
>
> Importantly, ICPO is a mechanistic account of test-time self-refinement as policy optimization (PO), with ME-ICPO as a minimal instantiation rather than a leaderboard-style test-time scaling tweak.

---

> ### Author Response · Authors · 2025-11-21
> **Response to Reviewer XA9N**
>
> **Q3.** How are the validity of the major assumptions made in ICPO?
>
> **A3.** We will demonstrate the validity of the major assumptions from the following three aspects.
> 1. **Model capability:** While we agree that the model for ICPO should be capable enough for reasonable improvement, we would like to highlight that this premise is shared by RL-style LLM methods: TTRL also needs the model to produce some workable rollouts to improve at test time, and GRPO-style training likewise fails to get signal if a group never yields a viable solution. Furturemore, we would like to highlight that in the worst case where the model suffers to deliver a good judgement, using a DPO-style reward [4] (e.g., $r_\theta \propto \log \pi_\theta - \log \pi_{\text{ref}}$) can also be an alternative plan.
> 2. **Verification is cheaper:** This assumption is reasonable in our setting so as to the general RL with verifiable reward setting. In addition, we would like to show that recent evidence shows a generation–verification gap: LLMs are often stronger as verifiers than as generators, which enables reliable selection and reranking at modest cost [5]. In competition math, checking a normalized final answer, running brief step checks, or using an LLM judge with strict token caps is typically far cheaper than producing full multi step solutions.
> 3. **The correctness of majority voting**. While we agree that if the base model never produce any correct answer within a dataset through majority voting, the model should be first finetuned instead of directly considered under test-time computing. However, we would like to highlight that in a more practical senerio where the model has the chance to output the correct answer, ICPO will further improve the response accuracy through in-context reflection process. Specially, we have proved that this in-context reflection process is robust to reward error generated through majority voiting (see Theorem 4.8). These claims are also verified by our ablation study (see Figure 6.2, in Sec. 6.1) and our empirical success.
> ---
>
> **Q4.** Additional experiments on recent long-CoT and frontier models
>
> **A4.** Per your suggestion, we evaluate Qwen3-4B-Instruct, Gemini-2.5-Pro and Gemini-2.5-Flash on AIME. Metrics: Mean@16 (%) and Acc (%). We would like to highlight that ME-ICPO provides substantial improvement on these models. The result is also presented in Appendix B.4.2 in our revision.
>
> **Table 4 — Frontier/long-CoT models on AIME**
>
> | Model                 | Method              | Mean@16 (%) | Acc (%) |
> |-----------------------|---------------------|-------------|---------|
> | Qwen3-4B-Instruct     | Base                | 20.62       | 20.59   |
> | Qwen3-4B-Instruct     | **ME-ICPO (ours)**  | **57.71**     | **57.67**   |
> | Gemini-2.5-Pro        | Base                | 58.54       | 56.60   |
> | Gemini-2.5-Pro        | **ME-ICPO (ours)**  | **79.17**       | **80.00**   |
> | Gemini-2.5-Flash      | Base                | 35.21       | 35.42   |
> | Gemini-2.5-Flash      | **ME-ICPO (ours)**  | **76.46**       | **76.47**   |
>
> ---
> **Q5.** Additional experiments on harder tasks.
>
> **A5.** Per your suggestion, we have added the harder benchmarks.
>
> We add results on HMMT and APEX-shortlist using Qwen2.5-Math-7B and Gemini-2.5-Flash. Metrics: Mean@16 (%) and Acc (%). For Qwen2.5-Math-7B, we hypothesize that these tasks are fundamentally hard so that the 7B model lacks enough capacity to solve them. However, with Gemini-2.5-Flash, ME-ICPO significantly improves the accuracy as presented in Table 5.2. The result is also presented in Appendix B.4.3 in our revision.
>
> **Table 5.1 — Qwen2.5-Math-7B on HMMT / APEX-shortlist**
>
> | Method                | HMMT Mean@16 (%)| HMMT Acc(%) | APEX Mean@16(%) | APEX Acc(%) |
> |-----------------------|--------------|----------|---------------|----------|
> | Base     |    **1.04**          |    0.67      |         2.55      |      2.61    |
> | **ME-ICPO (ours)**    |        0.42      |      **1.33**    |    **4.59**           |   **4.57**       |
>
>
> **Table 5.2 — Gemini-2.5-flash on HMMT / APEX-shortlist**
>
> | Method                | HMMT Mean@16 (%)| HMMT Acc(%) | APEX Mean@16(%) | APEX Acc(%) |
> |-----------------------|--------------|----------|---------------|----------|
> | Base     |        14.79      |     14.76     |           14.68 |   18.33     |
> | **ME-ICPO (ours)**    |     **43.12**         |     **43.14**     |     **17.18**        |   **20.00**     |

---

> ### Author Response · Authors · 2025-11-21
> **Response to Reviewer XA9N**
>
> **Q6.** How will ICPO work without final-answer executability or with non-discrete outputs.
>
> **A6.** We would like to highlight that ICPO is a framework, not tied to a particular reward. For open-ended or non-discrete tasks (free-form answers, proofs, summaries), we simply swap the reward module: use LLM-as-Judge scoring, or preference signals (e.g., DPO-based rewards [4] or preference ranking) computed inside the ICPO loop. ME-ICPO is tailored to math reasoning through verifiable majority voiting, but the same ICPO design pattern yields new, task-appropriate algorithms once the reward is redefined.
>
> ---
> **References**
>
> [1] Zhang, R., Frei, S., & Bartlett, P. L. (2024). Trained transformers learn linear models in-context. Journal of Machine Learning Research, 25(49), 1-55.
>
> [2] Russo, D., & Van Roy, B. (2013). Eluder dimension and the sample complexity of optimistic exploration. Advances in Neural Information Processing Systems, 26.
>
> [3] Nishikawa, Naoki, et al. "Nonlinear transformers can perform inference-time feature learning." Forty-second International Conference on Machine Learning.
>
> [4] Wang, Z., He, W., Liang, Z., Zhang, X., Bansal, C., Wei, Y., Fu, Y., & Yao, H. (2024). Cream: Consistency regularized self-rewarding language models. arXiv preprint arXiv:2410.12735.
>
> [5] Song, Y., Zhang, H., Eisenach, C., Kakade, S., Foster, D., & Ghai, U. (2024). Mind the gap: Examining the self-improvement capabilities of large language models. arXiv preprint arXiv:2412.02674.

---

> ### Author Response · Authors · 2025-11-24
> **Follow up with Reviewer XA9N**
>
> Thank you again for your review. We would like to follow up to see if there are any remaining questions.
>
> In our previous response, we have addressed your questions regarding (1) why the theoretical analysis is built on a simplified linear self-attention model, emphasizing that LSA serves as a minimal analytic proxy and that the main ICPO insights extend to more general function classes via tools like eluder dimension and representation learning; (2) the computational cost comparison with ToT, MCTR, and TTRL, by adding wall-clock and performance comparisons on AIME-2024/AMC/MATH showing that ME-ICPO achieves stronger Acc/Mean@16 under comparable or lower compute; (3) the validity of the ICPO assumptions, including model capability, the “verification is cheaper” assumption, and the robustness of majority voting with theoretical and empirical support; (4) additional experiments on frontier and long-CoT models (Qwen3-4B-Instruct, Gemini-2.5-Pro, Gemini-2.5-Flash) showing consistent gains from ME-ICPO; (5) additional experiments on harder benchmarks (HMMT and APEX-shortlist) with both Qwen2.5-Math-7B and Gemini-2.5-Flash; and (6) how ICPO can handle tasks without verifiable final answers or non-discrete outputs by swapping in LLM-as-judge or preference-based rewards (e.g., DPO-style signals) inside the same framework.
>
> We would be more than happy to have further discussions!

---

### Official Review · Reviewer_gZmq · 2025-11-01

**Soundness:** 3
**Presentation:** 3
**Contribution:** 3
**Rating:** 6
**Confidence:** 4

**Summary:**

This paper introduces In-Context Policy Optimization (ICPO), a theoretical framework that explains how large language models can self-improve during test-time by iteratively refining their responses without parameter updates.The paper formulates multi-round self-refinement in LLMs as a form of in-context policy optimization, where the model treats its previous responses and associated rewards as contextual experience to adjust future outputs. This extends existing in-context learning theory from supervised prediction to policy optimization with bandit feedback. One the theoretical side, the authors prove that a single-layer linear self-attention transformer, when pretrained using a Fisher-weighted logit-matching objective, can provably imitate a policy optimization algorithm for linear bandits, thereby establishing a mechanistic explanation for the emergence of self-reflection in LLMs. Based on the theory, the paper proposes ME-ICPO, a practical inference-time algorithm that performs iterative response refinement using self-assessed rewards and entropy-based selection to ensure robustness to reward noise. Across standard mathematical reasoning benchmarks, ME-ICPO achieves competitive and often state-of-the-art test-time performance while maintaining affordable inference cost, demonstrating that test-time scaling can be improved without parameter fine-tuning.

**Strengths:**

The authors derive provable guarantees showing that a linear self-attention transformer, when trained under a Fisher-weighted objective, can imitate the behavior of a policy optimization algorithm in a linear bandit setting. This is a novel result from the theoretical perspective.

The paper proposed Minimum-Entropy ICPO (ME-ICPO) algorithm which demonstrates a practical and implementable version of in-context policy optimization. It integrates entropy-regularized response selection and self-assessed rewards, leading to consistent empirical improvements in mathematical reasoning tasks. The experimental results are strong and align with the theoretical insights.

By modeling the self-reflection and iterative response refinement as In-Context policy optimization problem, the paper offers a clear mechanistic and mathematically grounded explanation for self-improvement phenomena observed in LLMs.

**Weaknesses:**

The effectiveness of ME-ICPO depends on choices such as number of refinement rounds, sample count per round, and entropy thresholds. Tuning those hyperparameters are non-trivial and might heavily depend on model sizes and datasets.

**Questions:**

How can we handle the situation that the model itself cannot score or rank its own responses? How can we handle mis-aligned reward heuristics that the incorrect reasonings are being rewarded or reinforced? How to mitigate such caveats?

---

> ### Author Response · Authors · 2025-11-21
> **Response to Reviewer gZmq**
>
> We appreciate the reviewer's recognition of our work's theoretical novelty and practical value, and we thank you for the insightful questions.
>
> **Q1.** How is ME-ICPO sensitive to its hyperparameter tuning?
>
>
> **A1.** We would like to first clarify that the design of ME-ICPO does not rely on the entropy threshold but instead selects the single candidate that yields the *minimum relative entropy*. This design choice intentionally avoids non-trivial hyperparameter tuning across different datasets. We have conducted a hyperparameter sensitivity analysis in Appendix B.1 (Figure 4) in our original manuscript. We list the detailed results below (evaluated on AIME 2024 with Qwen2.5-Math-7B) for your convenience.
>
>
>
> * **Table 1 — Sensitivity to Sample Count (k)** (fixed Rounds=5):
>     | k | Mean@16 (%) | Maj@16 (%) |
>     | :--- | :--- | :--- |
>     | 2 | 13.13 | 16.67 |
>     | 4 | 15.00 | 16.67 |
>     | 8 | 24.95 | 33.33 |
>     | 16 | 30.42 | 40.00 |
>     | 32 | 29.17 | 40.00 |
>     | 64 | **31.25** | **40.00** |
>
> * **Table 2 — Sensitivity to Rounds** (fixed $k=16$):
>     | Round | Mean@16 (%) | Maj@16 (%) |
>     | :--- | :--- | :--- |
>     | 1 | 16.67 | 23.33 |
>     | 2 | 23.33 | 30.00 |
>     | 3 | 23.96 | 30.00 |
>     | 4 | 23.33 | 30.00 |
>     | 5 | **30.42** | **40.00** |
>     | 6 | 27.92 | 36.67 |
>
> As shown in the tables, performance consistently improves as the computational budget ($k$ and rounds) increases and then stabilizes. For example, performance plateaus after $k=16$, and increasing $k$ to 32 or 64 maintains the high performance without degradation. Similarly, the method benefits from iterative refinement up to round 5. This indicates that ME-ICPO is stable and not brittle to hyperparameter changes; sufficient compute budget reliably yields performance gains.
>
> ---
>
> **Q2.** What if the model is bad at self-scoring or self-ranking?
>
> **A2.** We would like to first clarify that ME-ICPO uses Majority Judgment for the final outcomes to avoid explicit self-scoring or self-ranking. In particular, the reward $r_t$ is assigned based on whether a response's answer matches this majority-voted outcome. This outcome-based signal is more reliable than subjective reasoning-path scoring. We would also like to note that in the case when the majority judgement cannot be easily applied (e.g., when the answers cannot be easy verfied), applying the DPO-based intrinsic reward [1] (e.g., $r_\theta \propto \log \pi_\theta - \log \pi_\text{ref}$), as explored in recent self-rewarding literature [2]
>
>
> ---
>
> **Q3.** How to handle the mis-aligned rewards?
>
> We thank the reviewer for this critical question. This is a known challenge in reward-based learning, and our framework is equipped with robust defenses against mis-aligned rewards at multiple levels:
> 1.  Foundational Theoretical Robustness: the ICPO framework is grounded in Policy Optimization, which is fundamentally designed to handle noisy feedback. This approach is rooted in the non-stochastic (adversarial) bandit literature with robust algorithms like EXP4 [3] and the FTRL [4]. These algorithms are provably robust under corrupted or adversarial rewards.
> 2. We have provide a theoretical guarantee as presented in Theorem 4.8 analyzes the impact of a single reward shock (a misaligned reward). The theorem proves that the impact of this perturbation decays over time (as $t/s \to 0$) and does not get amplified, preventing the policy from destabilizing. This theoretical guarantee is further validated empirically in Figure 2 (Bottom), which clearly shows the "Purturbation Stability" of the ICPO framework when applied in LSAs.
> 3. Finally, we would like to highlight that the empirical success of these methods is built upon small-size language models like 7B models. These small models will unavoidably have misaligned rewards, but our empirical perforamcne have demonstrated that ICPO frameworks are remarkably stable against this type of feedback noise.
>
>
>
>
> ---
> **References**
>
> [1] Rafailov, Rafael, et al. "Direct preference optimization: Your language model is secretly a reward model." Advances in neural information processing systems 36 (2023): 53728-53741.
>
> [2] Wang, Z., He, W., Liang, Z., Zhang, X., Bansal, C., Wei, Y., Fu, Y., & Yao, H. (2024). Cream: Consistency regularized self-rewarding language models. arXiv preprint arXiv:2410.12735.
>
> [3] Auer, P., Cesa-Bianchi, N., Freund, Y., & Schapire, R. E. (2002). The nonstochastic multiarmed bandit problem. *SIAM Journal on Computing*, *32*(1), 48–77.
>
> [4] Shalev-Shwartz, S. (2012). Online learning and online convex optimization. *Foundations and Trends® in Machine Learning*, *4*(2), 107–194.

---

> ### Author Response · Authors · 2025-11-24
> **Follow up with Reviewer gZmq**
>
> Thank you again for your positive review. We would like to follow up with our previous review to see if there are any further concerns.
>
> In particular, in our previous response, we have addressed your questions regarding (1) the sensitivity of ME-ICPO to its hyperparameters \(k\) and the number of rounds, showing that performance improves and then plateaus without being brittle to these settings; (2) the concern that base models may be bad at self-scoring, by clarifying that ME-ICPO relies on Majority Judgment over final answers and can incorporate DPO-style intrinsic rewards when explicit majority voting is hard to apply; and (3) the impact of mis-aligned rewards, by highlighting the robustness guarantees of the ICPO framework (grounded in adversarial bandits and FTRL), our perturbation-stability theorem (Theorem 4.8) showing that reward shocks do not destabilize the policy, and empirical evidence that ICPO remains stable even with noisy rewards from 7B-scale models.
>
> We would be more than happy to continue the discussion!

---

### Author Response · Authors · 2025-11-21
**Response to all reviewers.**

We sincerely thank all reviewers for their insightful and constructive feedback. We appreciate the recognition of our mathematically grounded explanation (Reviewer gZmq, XA9N, 4bex, jkmj), practical algorithm (Reviewer gZmq, 4bex, jkmj). According to these comments, we have improved the paper (new pdf uploaded) and highlighted the main changes with red text. Below, we summarize all changes besides carefully correcting the typos.

1. Table 1 now reports *mean ± std over 5 seeds* to address variance reporting. (Sec. 6.3, [L418–L419]; addresses **Reviewer gZmq Q1**)
2. we highlight in Section 4 that the theoretical results serve as an inspirational analysis for in-context self-reflection via a minimal model (linear self-attention).(Sec. 4, [L196–L199]; addresses **Reviewer jkwj Q1**)
3. As requested by Reviewer jkwj, we note in the Conclusion that modeling multi-round reasoning as an MDP and analyzing multi-layer, nonlinear Transformer structure are left for future work. (Sec. 7, Conclusion, [L482–L485]; addresses **Reviewer jkwj Q3**)
4. Added compute comparisons vs. ToT, MCTR, and TTRL on AIME-2024/AMC/MATH, including time per question. ( Appx. B.4.1, [L1359–L1371]; addresses **Reviewer XA9N Q2**)
5. Added frontier/long-CoT models: Qwen3-4B-Instruct, Gemini-2.5-Pro, Gemini-2.5-Flash on AIME (Mean@16 & Acc). (Appx. B.4.2, [L1373–L1377]; addresses **Reviewer XA9N Q4**)
6. Added harder benchmarks HMMT and APEX-shortlist with Qwen2.5-Math-7B and Gemini-2.5-Flash. (Appx. B.4.3, [L1378–L1382]; addresses **Reviewer XA9N Q5**)

---

### Meta-Review · Area_Chair_ogpz · 2026-01-07

**Summary:**

The reviewers were split about this paper and did not come to a consensus. On one hand they appreciated the performance improvements of the method and the computational efficiency of the approach. On the other they had issues with (a) not enough experiments, (b) missing computational cost computations, (c) the abstraction gap between the theory and practice, (d) a lack of formal justification for the minimum-entropy selection criterion, (e) the simplification of k-armed bandit modelling. The authors responded convincingly to (a) and (b). Their response to (c)-(e) was limited: apart from fixing typos, two sentences were added to section 4, I don’t think the reviewer would have raised their score. However, I am confident that the authors can make further changes to the paper to respond to these points. After this is done, the authors have responded to all substantial reviewer concerns. For this reason, I vote to accept. Authors please carefully update the paper to respond to (c)-(e) in the final version. Once you have done this, the paper will make a nice contribution to the conference!

**Reviewer Concerns:**

Please see above.

**Reviewer Scores:**

I believe reviewers would have kept or increased their scores.

---

### Decision · Program_Chairs · 2026-01-26

Accept (Poster)